# Variance-Gated Ensembles: An Epistemic-Aware Framework for Uncertainty Estimation

**H. Martin Gillis**[*]                                    *martin.gillis@dal.ca*
*Faculty of Computer Science*
*Dalhousie University, Halifax, NS*

**Isaac Xu**[*]                                             *isaac.xu@dal.ca*
*Faculty of Computer Science*
*Dalhousie University, Halifax, NS*

**Thomas Trappenberg**[†]                                  *tt@cs.dal.ca*
*Faculty of Computer Science*
*Dalhousie University, Halifax, NS*

**Reviewed on OpenReview:** *https://openreview.net/forum?id=fNMZjV1gje*

## Abstract

Machine learning applications require fast and reliable per-sample uncertainty estimation. A common approach is to use predictive distributions from Bayesian or approximation methods and additively decompose uncertainty into aleatoric (data-related) and epistemic (model-related) components. However, additive decomposition has recently been questioned, with evidence that it breaks down when using finite-ensemble sampling and/or mismatched predictive distributions. This paper introduces variance-gated ensembles (VGE), a differentiable framework that injects epistemic sensitivity *via* a signal-to-noise gate computed from ensemble statistics. VGE provides: (i) a variance-gated margin uncertainty (VGMU) score that couples decision margins with ensemble predictive variance; and (ii) a variance-gated normalization (VGN) layer that generalizes the variance-gated uncertainty mechanism to training *via* per-class, learnable normalization of ensemble member probabilities. We derive closed-form vector-Jacobian products enabling end-to-end training through ensemble sample mean and variance. VGE is positioned as a compute-efficient alternative to pairwise-divergence methods. It delivers competitive uncertainty quality relative to state-of-the-art information-theoretic baselines, while reducing per-sample cost from quadratic to linear in ensemble size. As a result, VGE provides a practical and scalable approach to epistemic-aware uncertainty estimation in ensemble models.[1]

## 1 Introduction

Machine learning models achieve high accuracy on average yet still fail on individual predictions in ways that are obvious on inspection. Flagging such failures requires per-sample uncertainty estimates that indicate when a prediction should not be trusted. A common approach to estimate uncertainty is Bayesian model averaging (BMA) in Bayesian neural networks (BNNs) or through approximations such as Monte Carlo dropout (MCD) (Gal & Ghahramani, 2016), deep ensembles (DE) (Lakshminarayanan et al., 2017), and last-layer ensembles (LLEs) (Harrison et al., 2024; Sensoy et al., 2018). These methods approximate the predictive distribution by averaging predictions from an ensemble of models, and the standard approach decomposes the resulting predictive entropy into data- and model-related components (Houlsby et al., 2011).

---

[*]Co-first authors.
[†]Corresponding author.
[1]Code available at: https://github.com/nextdevai/vge.

However, existing approaches face a trade-off between efficiency and reliability. Entropy-based decompositions are efficient but unreliable, and finite ensembles introduce sampling error into the decomposition. Methods such as MCD and DE sample from approximate posteriors (*i.e.*, the dropout distribution or an implicit distribution over independently-trained networks) rather than the true posterior (Wimmer et al., 2023; Schweighofer et al., 2023). Pairwise alternatives such as the expected pairwise Kullback–Leibler (EPKL) divergence (Schweighofer et al., 2023) recover epistemic sensitivity under finite ensembles but incur an $O(M^2C)$ cost that scales poorly with ensemble size and class count.

Here, we introduce variance-gated ensembles (VGE), a framework that recovers epistemic sensitivity at linear cost through an exponential gating mechanism. We propose a variance-gated margin uncertainty (VGMU) score using a margin-based signal-to-noise of the top-2 ranked classes. For decisions about whether to trust a prediction or defer to review, only the separability of the leading candidates relative to ensemble agreement is relevant, not disagreement among distant classes. This concept is extended to an end-to-end variance-gated normalization (VGN) layer using a per-class-based signal-to-noise, where parameters are used to adaptively scale a gating mechanism during training. The VGN layer suppresses high-variance, low-consensus predictions and re-shapes member distributions. VGE achieves $O(MC)$ complexity for uncertainty decomposition and $O(C)$ for VGMU evaluation at inference, enabling real-time deployment without sacrificing epistemic sensitivity.

In summary, this research provides the following contributions:

1. **Variance-Gated Ensembles.** A framework that introduces epistemic sensitivity *via* a signal-to-noise gate computed from ensemble statistics.

2. **Variance-Gated Margin Uncertainty.** An inference-time score combining decision margin with ensemble predictive variance to quantify class separability.

3. **Variance-Gated Normalization.** A differentiable, epistemic-aware normalization layer that modulates member probabilities through a per-class learnable parameter, enabling efficient end-to-end optimization with $O(MC)$ complexity.

Apart from this section, the remainder of the paper is organized as follows: Section 2 provides the information-theoretic background for uncertainty decomposition and positions this work within the existing literature. Section 3 introduces the variance-gated ensemble framework, including the VGMU scoring metric and the formulation of VGN. This section also presents a geometric-based interpretation of ensemble predictions. Section 4 introduces the analytical gradients required for end-to-end training using VGN. Section 5 presents experimental set-up and results, comparing the proposed approach with common uncertainty estimation methods in the machine learning literature. Section 6 provides an axiomatic comparison to prior uncertainty decomposition frameworks, and clarifies the intended scope and limitations of VGN and VGMU. Finally, Section 7 concludes the paper with a summary of findings and recommendations.

## 2 Background and Related Work

### 2.1 Predictive Uncertainty

Predictive uncertainty under a probabilistic model is typically formalized through the Bayesian predictive distribution, which averages model predictions over a posterior on parameters. Let $\mathbf{w}$ denote the parameters of a probabilistic model and $\mathcal{D} = \{(\mathbf{x}_n, y_n)\}_{n=1}^N$ the observed data. Bayesian inference maintains a posterior $p(\mathbf{w} \mid \mathcal{D})$ over parameters. Predictions for a new input $\mathbf{x}$ are obtained *via* BMA, which integrates over the posterior to yield the predictive distribution over labels $y$

$$p(y \mid \mathbf{x}, \mathcal{D}) = \int_{\mathbf{w}} p(y \mid \mathbf{x}, \mathbf{w}) \, p(\mathbf{w} \mid \mathcal{D}) \, d\mathbf{w}. \tag{1}$$

This integral relies on the assumption that $y$ is *conditionally independent of $\mathcal{D}$ given* $\mathbf{w}$

$$p(y \mid \mathbf{x}, \mathbf{w}, \mathcal{D}) = p(y \mid \mathbf{x}, \mathbf{w}). \tag{2}$$

This means that once $\mathbf{w}$ is known, the training data provides no additional information for predicting $y$. To quantify this uncertainty, we measure the Shannon entropy of the predictive distribution

$$H(y \mid \mathbf{x}, \mathcal{D}) = -\sum_y p(y \mid \mathbf{x}, \mathcal{D}) \log p(y \mid \mathbf{x}, \mathcal{D}) \tag{3}$$

which captures the predictive uncertainty for a new input $\mathbf{x}$.

## 2.2 Additive Decomposition

Total uncertainty (TU) as measured by entropy conflates two distinct sources, aleatoric uncertainty (AU) and epistemic uncertainty (EU). By definition, the mutual information ($I$) between the label ($y$) and parameters $\mathbf{w}$ is the reduction in predictive entropy from knowing $\mathbf{w}$, where $I(y; \mathbf{w} \mid \mathbf{x}, \mathcal{D}) = H(y \mid \mathbf{x}, \mathcal{D}) - H(y \mid \mathbf{w}, \mathbf{x}, \mathcal{D})$. Applying the conditional independence assumption (Equation 2), we can decompose $H(y \mid \mathbf{x}, \mathcal{D})$. Since $p(y \mid \mathbf{x}, \mathbf{w}, \mathcal{D}) = p(y \mid \mathbf{x}, \mathbf{w})$, the conditional entropy of $y$ given $\mathbf{w}$ can be estimated from model sampling (*i.e.*, an ensemble)

$$H(y \mid \mathbf{w}, \mathbf{x}, \mathcal{D}) = \mathbb{E}_{\mathbf{w} \sim p(\mathbf{w}|\mathcal{D})}\big[H(y \mid \mathbf{x}, \mathbf{w})\big]. \tag{4}$$

Replacing the second term with Equation 4 results in the additive decomposition.

$$\underbrace{H(y \mid \mathbf{x}, \mathcal{D})}_{\text{TU}} = \underbrace{\mathbb{E}_{\mathbf{w} \sim p(\mathbf{w}|\mathcal{D})}\big[H(y \mid \mathbf{x}, \mathbf{w})\big]}_{\text{AU}} + \underbrace{I(y; \mathbf{w} \mid \mathbf{x}, \mathcal{D})}_{\text{EU}}. \tag{5}$$

The two terms have qualitatively different implications for downstream tasks such as active learning, selective prediction, out-of-distribution (OOD) detection, and human-in-the-loop systems. The aleatoric term $\mathbb{E}_{\mathbf{w} \sim p(\mathbf{w}|\mathcal{D})}[H(y \mid \mathbf{x}, \mathbf{w})]$ averages the entropy of the individual model predictions over the posterior. It captures noise intrinsic to the data-generating process, uncertainty that persists even if the posterior were known exactly and therefore cannot be reduced by collecting more data. The epistemic term $I(y; \mathbf{w} \mid \mathbf{x}, \mathcal{D})$ measures how much the predictions of the model would change if we knew which parameters $\mathbf{w}$ were correct. When ensemble members disagree, this term is large; when they agree, it vanishes. Unlike aleatoric uncertainty, epistemic uncertainty is in principle reducible. Observing additional data updates the posterior $p(\mathbf{w} \mid \mathcal{D})$ and reduces the disagreement among model hypotheses. The additive decomposition (Equation 5) enables downstream decisions for active learning (Houlsby et al., 2011), selective prediction (Geifman & El-Yaniv, 2017), and OOD detection (Lakshminarayanan et al., 2017) that selectively target only the reducible component of predictive uncertainty (Gawlikowski et al., 2023).

## 2.3 Expected Pairwise Kullback–Leibler Divergence

In practice, the posterior $p(\mathbf{w} \mid \mathcal{D})$ is approximated by a finite ensemble $\{\mathbf{w}_m\}_{m=1}^M$, with the posterior predictive distribution approximated as the mixture

$$p(y \mid \mathbf{x}, \mathcal{D}) \approx \frac{1}{M} \sum_{m=1}^M p(y \mid \mathbf{x}, \mathbf{w}_m). \tag{6}$$

When a finite ensemble produces individual member distributions that are non-Gaussian, multimodal, or heavily skewed, the averaged prediction may not represent the output of any individual member, and its entropy fails to capture the true spread of the ensemble (Wimmer et al., 2023). Given this finite ensemble approximation (Equation 6), an alternative for the epistemic component is the EPKL divergence (Schweighofer et al., 2023)

$$\text{EPKL}(\mathbf{x}) = \frac{1}{M(M-1)} \sum_{i=1}^M \sum_{\substack{j=1 \\ j \neq i}}^M \mathrm{D}_{\mathrm{KL}}(p(y \mid \mathbf{x}, \mathbf{w}_i) \,\|\, p(y \mid \mathbf{x}, \mathbf{w}_j)). \tag{7}$$

EPKL is non-negative, equals zero if and only if all ensemble members agree, and captures pairwise disagreement directly in predictive distribution space without requiring estimation of the mixture entropy.

However, a practical limitation of EPKL is quadratic scaling. Evaluating Equation 7 requires $O(M^2 C)$ pairwise divergence computations, compared with the $O(MC)$ for entropy-based estimators. This cost becomes prohibitive as ensemble size and number of classes increases, motivating more efficient alternatives that still capture ensemble disagreement.

### 2.4 Ensemble Uncertainty Estimation

**Ensemble approximations.**   Uncertainty estimation (Gawlikowski et al., 2023; Hüllermeier & Waegeman, 2021) in classification tasks is often framed through Bayesian predictive distributions; however, an exact Bayesian solution is intractable for modern neural networks. Practical approximations include variational Bayesian neural networks (Radford, 1995), MCD (Gal & Ghahramani, 2016; Gal et al., 2017), DE (Lakshminarayanan et al., 2017), and LLE approaches such as evidential deep learning (Sensoy et al., 2018), variational inference (Harrison et al., 2024; Steger et al., 2024), multihead (Lee et al., 2015), and other ensemble strategies (Huang et al., 2017; Kushibar et al., 2022; Wen et al., 2020). Among these approaches, DE have demonstrated strong performance in calibration and OOD detection, often outperforming approximate Bayesian methods under dataset shift (Ovadia et al., 2019). As a result, many recent uncertainty estimation methods operate on ensemble predictive distributions.

**Moment-based methods.**   An alternative line of work derives uncertainty signals from ensemble statistics, such as the mean and variance of predicted class probabilities (Depeweg et al., 2018; Smith & Gal, 2018). Moment-based approaches are attractive in large-scale or real-time settings. VGE follows this approach by deriving measures from ensemble variance, enabling linear-time computation while retaining sensitivity to predictive disagreement.

**Calibration methods.**   Calibration methods, such as temperature scaling, operate post hoc and adjust predictive confidence without modifying uncertainty structure during training (Guo et al., 2017; Kumar et al., 2022). In contrast, recent work explores integrating uncertainty-aware mechanisms into the training process itself, including diversity-promoting losses and uncertainty-regularized objectives (Fort et al., 2020; Ashukha et al., 2021). VGN contributes to this area by introducing a differentiable normalization layer that modulates ensemble predictions based on epistemic variance, allowing uncertainty re-shaping to be learned end-to-end.

**Margin-based criteria.**   In many applications, risk-based decisions depend on ambiguity between top-ranked classes rather than uncertainty of the full-simplex distribution. Margin-based criteria, such as Best-versus-Second Best (BvSB) scores, are used in active learning and selective prediction (Joshi et al., 2009; Geifman & El-Yaniv, 2017). VGMU extends this method by incorporating ensemble variance into the decision margin, offering a decision-focused epistemic uncertainty measure that emphasizes disagreement among top-ranked classes while remaining computationally efficient.

The variance-gated ensemble framework, introduced in the next section, derives uncertainty directly from ensemble moments, achieving linear-time epistemic sensitivity without requiring pairwise comparisons.

## 3 Framework Definition and Setup

This section describes the variance-gated ensemble framework for epistemic-aware ensemble modeling, defining normalization and analytical properties. We first formalize variance-gated normalization, where member probabilities are modulated by a signal-to-noise gate, and analyze its sensitivity to sample mean confidence, predictive spread, and per-class learnable parameter **k**. We then provide a geometric-based interpretation, followed by a variance-gated uncertainty decomposition for total uncertainty into aleatoric and epistemic components, and introduce the variance-gated margin uncertainty score to capture class separability. Figure 1 provides a high-level overview of the VGE framework, illustrating how ensemble predictions flow through variance-gated normalization to produce epistemic-aware distributions and uncertainty scores. See Table S1 in the supporting information for a listing of symbols and abbreviations used throughout the variance-gated ensemble framework.

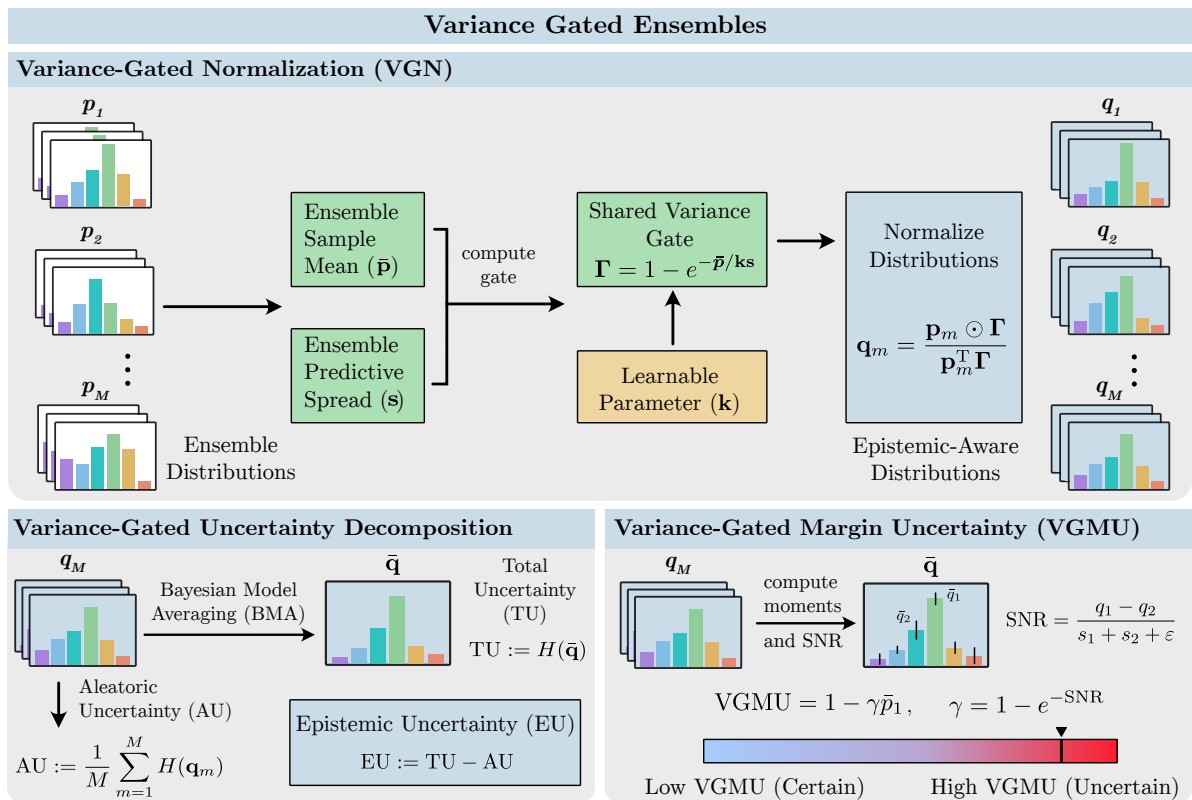

Figure 1: Overview of the variance-gated ensemble framework. Top (training-time): Variance-gated normalization (VGN) computes a shared gate $\boldsymbol{\Gamma}$ from the ensemble mean $\bar{\mathbf{p}}$, predictive spread $\mathbf{s}$, and learnable parameter $\mathbf{k}$, producing epistemic-aware distributions $\mathbf{q}_m$; since it reshapes separately trainable members, VGN applies to deep ensembles (DE-VGN) and last-layer ensembles (LLE-VGN), but is undefined for MC-dropout and the hybrid MCD-LLE, whose members arise from stochastic dropout passes rather than separately trainable sub-models. Bottom (left, inference-time): Variance-gated additive decomposition of total uncertainty into aleatoric and epistemic components. Bottom (right, inference-time): Variance-gated margin uncertainty (VGMU), a decision-focused score computed from the top-2 class margins and their signal-to-noise ratio. The two inference-time paths apply to any ensemble (DE, LLE, MCD, MCD-LLE); in particular, VGMU does not require the variance-gated prediction $q$ and can be evaluated on any ensemble output $p$ given access to per-member means and standard deviations.

**Notation convention.** Throughout this paper, boldface lowercase letters denote $C$-dimensional vectors (*e.g.*, $\mathbf{p}_m$, $\bar{\mathbf{p}}$, $\mathbf{s}$, $\boldsymbol{\Gamma}$) and all operations on such vectors, including squaring, division, exponentiation, multiplication, and inequalities are applied element-wise unless explicitly stated otherwise.

**Problem setup.** Consider a $C$-class classification task in which an input $\mathbf{x}$ is mapped to a label $y \in \{1, \ldots, C\}$. An ensemble of $M$ models with parameters $\{\mathbf{w}_m\}_{m=1}^{M}$ is available, each producing a predictive categorical distribution over classes. The goal is per-sample uncertainty estimation. Given a new input $\mathbf{x}$, quantify both the predictive confidence of the ensemble and the degree of inter-member disagreement, decomposing predictive uncertainty into aleatoric and epistemic components.

## 3.1 Ensemble Statistics and Variance Gate Definition

We first define the variance gate and analyze its local sensitivities. Let each ensemble member produce a predictive categorical distribution

$$\mathbf{p}_m = p(y \mid \mathbf{x}, \mathbf{w}_m) \in \Delta^{C-1}, \qquad m \in \{1, \ldots, M\}, \tag{8}$$

where $\mathbf{p}_m = [p_m(1), \ldots, p_m(C)]^\top$ lies in a compact convex region known as the $(C-1)$-simplex. The per-class ensemble sample mean and standard deviation are defined as

$$\bar{\mathbf{p}} = \frac{1}{M} \sum_{m=1}^{M} \mathbf{p}_m, \qquad \mathbf{s} = \sqrt{\frac{1}{M-1} \sum_{m=1}^{M} (\mathbf{p}_m - \bar{\mathbf{p}})^2 + \varepsilon}, \quad M > 1. \tag{9}$$

Let $\bar{\mathbf{p}} \geq 0$, $\mathbf{s} \geq 0$, $\mathbf{k} > 0$, and $\varepsilon > 0$ (*e.g.*, $1.0 \times 10^{-8}$).

**Design rationale.** The variance gate requires a mapping from the signal-to-noise ratio $\bar{\mathbf{p}}/\mathbf{ks}$ to a gating weight in $[0, 1)$. We adopt the exponential form $1 - e^{-(\cdot)}$ because it satisfies four desirable properties: (i) smoothness and differentiability everywhere, enabling end-to-end gradient-based training; (ii) monotonic increase with mean confidence and monotonic decrease with predictive spread; (iii) bounded output in $[0, 1)$, ensuring numerical stability during normalization; and (iv) saturation that prevents excessive attenuation of well-supported predictions. The exponential form is one choice among several possible gating families (*e.g.*, sigmoid, rational, piecewise-linear). Alternative parameterizations that trade-off sensitivity and saturation are discussed as an open direction in [Section 6](#).

The variance gate is defined as

$$\mathbf{\Gamma} = 1 - e^{-\bar{\mathbf{p}}/\mathbf{ks}}, \qquad \mathbf{\Gamma} \in \mathbb{R}^C, \quad 0 \leq \mathbf{\Gamma} < 1 \quad \text{(element-wise)} \tag{10}$$

and the normalized variance-gated member distribution is defined as

$$\mathbf{q}_m = \frac{\mathbf{p}_m \odot \mathbf{\Gamma}}{Z_m}, \quad Z_m = \mathbf{p}_m^\top \mathbf{\Gamma} \qquad \mathbf{q}_m \in \mathbb{R}^C, \quad 0 \leq \mathbf{q}_m \leq 1 \quad \text{(element-wise)} \tag{11}$$

where the scalar normalization $Z_m$ ensures $\mathbf{1}^\top \mathbf{q}_m = 1$ and $\mathbf{1} \in \mathbb{R}^C$ denotes the all-ones vector. The per-class $\mathbf{k} > 0$ controls the sensitivity of the gate.

The variance gate modulates ensemble predictions based on the scaled signal-to-noise ratio $\text{SNR} = \bar{\mathbf{p}} / \mathbf{ks}$ and acts as a smooth reliability correction. This means that classes with high mean confidence and low predictive spread receive larger gate values, while classes that are uncertain or highly variable are suppressed before normalization. In other words, $\mathbf{\Gamma}$ returns a per-class reliability weight close to 1 when the ensemble agrees confidently and close to 0 when the ensemble is unreliable; multiplying each member distribution $\mathbf{p}_m$ by $\mathbf{\Gamma}$ and re-normalizing therefore preserves trusted classes while attenuating untrusted predictions before downstream uncertainty computations. We now examine the sensitivity of the variance gate and gated member distributions to sample mean confidence $\bar{\mathbf{p}}$, predictive spread $s$, scaling factor $\mathbf{k}$, and its effects on gated distributions $\mathbf{q}_m$.

**Proposition 3.1** (Sensitivity to sample mean confidence $\bar{\mathbf{p}}$). *For the exponential gate $\mathbf{\Gamma} = 1 - e^{-\bar{\mathbf{p}}/\mathbf{ks}}$, the per-class derivative with respect to mean confidence $\partial \mathbf{\Gamma}/\partial \bar{\mathbf{p}} > 0$ and is modulated by predictive spread $\mathbf{s}$ through both an explicit inverse prefactor $1/(k_c s_c)$ and the saturation term $(1 - \Gamma_c)$, which itself depends on $s_c$ through the exponent.*

*Proof.* Consider a single class $c \in \{1, \ldots, C\}$, for which the gate is

$$\Gamma_c = 1 - e^{-\bar{p}_c/k_c s_c}. \tag{12}$$

Differentiating with respect to the mean confidence $\bar{p}_c$ yields

$$\frac{\partial \Gamma_c}{\partial \bar{p}_c} = \frac{1}{k_c s_c} e^{-\bar{p}_c/k_c s_c} = \frac{1 - \Gamma_c}{k_c s_c} > 0. \tag{13}$$

The derivative is strictly positive, showing that increasing the mean confidence for class $c$ always increases its gate value. The dependence on $s_c$ enters through two mechanisms: the explicit prefactor $1/(k_c s_c)$ provides linear suppression, while the saturation term $(1 - \Gamma_c) = e^{-\bar{p}_c/k_c s_c}$ provides nonlinear modulation. As ensemble disagreement $s_c$ increases, both factors contribute to reducing the derivative, meaning that the gate becomes less responsive to changes in mean confidence. Thus, for classes with high predictive variance, increases in $\bar{p}_c$ are suppressed. $\square$

**Remark 3.1.1.** *The factor $(1 - \Gamma_c)$ acts as a saturation term. As $\bar{p}_c/(k_c s_c) \to \infty$, we have $\Gamma_c \to 1$ and therefore $\partial \Gamma_c/\partial \bar{p}_c \to 0$. Consequently, once a class is deemed sufficiently reliable, further increases in mean confidence produce diminishing effects. This ensures that the variance-gate is most sensitive in intermediate regions and becomes increasingly insensitive as confidence saturates.*

**Proposition 3.2** (Sensitivity to predictive spread **s**). *For the exponential gate $\mathbf{\Gamma} = 1 - e^{-\bar{\mathbf{p}}/\mathbf{ks}}$, the per-class derivative with respect to predictive spread is $\partial \mathbf{\Gamma}/\partial \mathbf{s} < 0$ and is modulated by mean confidence $\bar{\mathbf{p}}$ through both an explicit linear prefactor $\bar{p}_c$ and the saturation term $(1 - \Gamma_c)$, which itself depends on $\bar{p}_c$ through the exponent.*

*Proof.* Consider a single class $c \in \{1, \ldots, C\}$, for which

$$\Gamma_c = 1 - e^{-\bar{p}_c/k_c s_c}. \tag{14}$$

Differentiating with respect to the predictive spread $s_c$ yields

$$\frac{\partial \Gamma_c}{\partial s_c} = -\frac{\bar{p}_c}{k_c s_c^2} \, e^{-\bar{p}_c/k_c s_c} = -\frac{(1 - \Gamma_c) \, \bar{p}_c}{k_c s_c^2} < 0. \tag{15}$$

The derivative is strictly negative, indicating that increasing predictive spread decreases the gate value. The factor $1/k_c s_c^2$ shows that the magnitude of this effect decays rapidly as $s_c$ grows. Thus, classes with high ensemble disagreement experience strong suppression, while further increases in large spreads have diminishing influence. $\square$

**Remark 3.2.1.** *Increasing the predictive spread $s_c$ while holding $\bar{p}_c$ and $k_c$ fixed decreases the gate through two mechanisms: i) a linear dependence on $\bar{p}_c$ and a quadratic decay in $s_c$; and ii) the saturation factor $(1 - \Gamma_c)$ that ensures once the gate is already small, additional increases in spread have limited effect. This behavior enforces strong suppression for uncertain classes, while the quadratic decay in $s_c$ ensures diminishing sensitivity as predictive spread grows.*

**Proposition 3.3** (Sensitivity to scalar **k**). *For the exponential gate $\mathbf{\Gamma} = 1 - e^{-\bar{\mathbf{p}}/\mathbf{ks}}$, where $k_c$ may be user-defined or learned, the derivative with respect to **k** is $\partial \mathbf{\Gamma}/\partial \mathbf{k} < 0$ and is modulated by mean confidence $\bar{\mathbf{p}}$ through both an explicit linear prefactor $\bar{p}_c$ and the saturation term $(1 - \Gamma_c)$, which itself depends on $\bar{p}_c$ through the exponent.*

*Proof.* Consider a single class $c \in \{1, \ldots, C\}$, for which

$$\Gamma_c = 1 - e^{-\bar{p}_c/k_c s_c}. \tag{16}$$

Differentiating with respect to the scaling parameter $k_c$ yields

$$\frac{\partial \Gamma_c}{\partial k_c} = -\frac{\bar{p}_c}{k_c^2 s_c} \, e^{-\bar{p}_c/k_c s_c} = -\frac{(1 - \Gamma_c) \, \bar{p}_c}{k_c^2 s_c} < 0. \tag{17}$$

Thus, increasing $k_c$ decreases the gate value. The inverse quadratic dependence on $k_c$ shows that the influence of the scaling parameter rapidly diminishes as $k_c$ grows, resulting in controlled and saturating sensitivity. $\square$

**Remark 3.3.1.** *Increasing $k_c$ while holding $\bar{p}_c$ and $s_c$ fixed decreases the gate by increasing the effective predictive spread. The quadratic decay in $k_c$ ensures that sensitivity to further increases rapidly diminishes, while the saturation factor $(1 - \Gamma_c)$ prevents excessive attenuation once the gate is already small. Under mild distributional assumptions on ensemble dispersion, the product $k_c s_c$ may be interpreted as a classwise risk-tolerance threshold reflecting typical deviations from the ensemble mean.*

### 3.1.1 Geometric Interpretation

We now reinterpret variance-gating geometrically in the probability simplex. Each pair $(\bar{\mathbf{p}}, \{\mathbf{p}_m\})$ defines a configuration in the simplex $\Delta^{C-1}$. The geometry of this configuration is defined by: (i) Confidence (or ambiguity): How close (or far) the ensemble sample mean $\bar{\mathbf{p}}$ is positioned near a simplex vertex; and (ii)

Certainty (or uncertainty): How close (or far) the set of members cluster around the ensemble sample mean $\bar{\mathbf{p}}$ (*i.e.*, ensemble member agreement/disagreement). These geometric effects are modulated by the local sensitivities of the variance gate, which increases with mean confidence and is progressively attenuated by predictive spread and the classwise risk-tolerance scale $k_c s_c$. The qualitative combinations of these two dimensions correspond to four simplex regions that define the space of all ensemble behaviors.

**Confident–Certain.** This region is characterized by a near-deterministic ensemble mean and low variance. The ensemble members form a cluster around a single vertex of the simplex, indicating high confidence and high agreement:

$$\bar{p}_c \uparrow, \; s_c \downarrow \Longrightarrow \Gamma_c = 1 - e^{-\bar{p}_c/k_c s_c} \approx 1. \tag{18}$$

**Ambiguous–Certain.** This region occurs when the ensemble mean is near-uniform but the variance is low. Members concentrate near the simplex barycenter, showing ambiguity but mutual certainty:

$$\bar{p}_c \downarrow, \; s_c \downarrow \Longrightarrow \Gamma_c = 1 - e^{-\bar{p}_c/k_c s_c}, \quad \text{depends on } \bar{p}_c/k_c s_c. \tag{19}$$

**Confident–Uncertain.** This region is defined by a near-deterministic mean but high variance. Members radiate outward from a vertex, reflecting confidence but disagreement among ensemble members:

$$\bar{p}_c \uparrow, \; s_c \uparrow \Longrightarrow \Gamma_c = 1 - e^{-\bar{p}_c/k_c s_c} \ll 1, \quad \text{with sensitivity suppressed by large } s_c. \tag{20}$$

**Ambiguous–Uncertain.** This region is defined by both a near-uniform mean and high variance. Members are diffused around the barycenter, indicating ambiguity and high disagreement:

$$\bar{p}_c \downarrow, \; s_c \uparrow \Longrightarrow \Gamma_c = 1 - e^{-\bar{p}_c/k_c s_c} \approx 0. \tag{21}$$

The variance gate functions as a continuous geometric adjustment within the probability simplex. It maintains the geometry in confident–certain regions, selectively attenuates confident–uncertain areas, and contracts diffuse ensembles' predictions in ambiguous–uncertain cases. In contrast, ambiguous–certain regions remain unaffected, as their geometry already reflects agreement in uncertainty. These four simplex spaces define the set of ensemble behaviors through which variance-gating is applied.

Collectively, the sensitivity properties of the variance gate define a controlled signal-to-noise mechanism. The gate increases with mean confidence $\bar{p}_c$, but this effect is progressively attenuated by predictive spread $s_c$ and the class-wise parameter $k_c$, with sensitivities that decay quadratically in both quantities. As a result, confident and consistent ensemble predictions are preserved, while high-confidence but high-variance predictions are suppressed in a stable and saturating manner (for an alternative risk-based interpretation, see SI S5).

### 3.2 Variance-Gated Margin Uncertainty

Entropy is known to overestimate uncertainty when probability values are spread across many classes and can underestimate it when a model is highly confident about a few options. Therefore, entropy alone is not sufficient to provide adequate information for decision-making. We posit the following question: *Can we identify a measure that is sensitive to class separation, while incorporating uncertainty awareness?* We want to identify a scoring metric that (i) maintains epistemic awareness and (ii) ideally, provides a user-defined threshold that reflects the degree of acceptable risk.

We propose using the margins of the top-2 mean predictions and their corresponding standard deviations. The motivation is decision-focused. For the practical question of whether to trust a prediction or defer to human review, uncertainty about distant classes is irrelevant, and what matters is the reliability of the top-ranked prediction relative to ensemble agreement. A sample whose top prediction carries 90% probability with strong ensemble agreement is trustworthy, whereas the same 90% prediction with high inter-member variance at the decision boundary is not. This perspective motivates a measure operating only on the margin between the top-ranked classes, modulated by ensemble agreement; we defer the full decision-theoretic,

computational, and empirical justification of the top-2 restriction to the discussion below (see "Justification for the top-2 restriction").

This approach extends the BvSB score used by Joshi et al. (2009), incorporating a sensitivity scalar $k$ (applied to all classes). Let $\bar{p}_1$ and $\bar{p}_2$ denote the top-1 and top-2 ranked ensemble mean probabilities of $\bar{\mathbf{p}}$, with $s_1$ and $s_2$, their corresponding standard deviations from $\mathbf{s}$. Let $i$ denote the class index of the top-1 ranked prediction. We define a prediction rule $\hat{y}$ and derive a margin-based SNR as

$$\hat{y} = \begin{cases} i & \text{if } \bar{p}_1 - ks_1 > \bar{p}_2 + ks_2 \\ \text{abstain} & \text{otherwise} \end{cases}; \qquad \text{SNR} = \frac{\bar{p}_1 - \bar{p}_2}{s_1 + s_2 + \varepsilon} > k \tag{22}$$

where $\bar{p}_1 - \bar{p}_2$ is the probability margin between the two most likely classes, $s_1 + s_2 + \varepsilon$ is the combined standard deviations, and "abstain" denotes that the model declines to predict due to insufficient margin. In principle, the SNR can be interpreted as a binary decision boundary between classes $\bar{p}_1$ and $\bar{p}_2$, restricted by $k$. Under mild distributional assumptions on ensemble dispersion, threshold $k$ can be set to reflect the fraction of samples that require abstentions or human intervention (i.e., by sorting the margin-based SNR in increasing values). For example, when $k = 1$, only samples with SNR $> 1$ will be considered; all others are abstained. However, this criterion fails to capture cases where a model outputs ambiguous and uncertain predictions. Such outputs artificially inflate the SNR values, leading to misleading classifications. To address this limitation, we introduce VGMU, using a gating function that rescales the top-ranked model prediction by incorporating both confidence and variance (i.e., epistemic) information

$$\text{VGMU} = 1 - \gamma \bar{p}_1, \qquad \gamma = 1 - e^{-\text{SNR}}. \tag{23}$$

The VGMU functions as an uncertainty metric, where small values correspond to confident, well-separated predictions (low uncertainty), while large values capture ambiguous or uncertain cases (low separation or high variance). The "variance-gated" designation refers to the role of ensemble standard deviations $s_1, s_2$ within the SNR-based gate $\gamma$. VGMU operates on the ensemble distributions $\mathbf{p}$ or the VGN-transformed distributions $\mathbf{q}$, making it a lightweight, post hoc metric applicable at inference time without retraining.

**Justification for the top-2 restriction.** The restriction to the top-2 classes is motivated by three considerations. First, from a decision-theoretic perspective, the practical question in selective prediction and human-in-the-loop systems is whether the model can distinguish its best prediction from the runner-up; disagreement about distant classes is irrelevant to the decision at hand. This aligns with the Best-versus-Second-Best framework of Joshi et al. (2009). Second, restricting to the top-2 classes yields $O(C)$ complexity compared to $O(M^2 C)$ for pairwise divergence measures, enabling the computational speedups demonstrated in our experiments. Third, the empirical validation in Table 3 shows that on CIFAR-10, VGMU achieves Spearman $\rho > 0.985$ with full-simplex measures, confirming that the top-2 restriction is consistent in ranking uncertainty for moderate class counts. On CIFAR-100, the lower correlation with pairwise measures reflects intentional insensitivity to tail-class disagreement. A property that we view as desirable for decision-focused uncertainty estimation, where the relevant question is separability of the leading candidates rather than full-simplex characterization.

## 3.3 Variance-Gated Uncertainty Decomposition

Using the variance-gated distributions, we define the ensemble mixture $\bar{\mathbf{q}} = \frac{1}{M} \sum_{m=1}^{M} \mathbf{q}_m$, where $\bar{\mathbf{q}}$ denotes the mean of the variance-gated member distributions. The total uncertainty is then measured by

$$\text{TU} := H(\bar{\mathbf{q}}) = -\bar{\mathbf{q}}^\top \log \bar{\mathbf{q}}. \tag{24}$$

Following standard ensemble decomposition (Houlsby et al., 2011), we define the gated aleatoric and epistemic components as

$$\text{AU} := \frac{1}{M} \sum_{m=1}^{M} H(\mathbf{q}_m) = -\frac{1}{M} \sum_{m=1}^{M} \mathbf{q}_m^\top \log \mathbf{q}_m, \qquad \text{EU} := \text{TU} - \text{AU} = \frac{1}{M} \sum_{m=1}^{M} D_{\text{KL}}(\mathbf{q}_m \,\|\, \bar{\mathbf{q}}). \tag{25}$$

Table 1: Components of the variance-gated ensemble (VGE) framework. The variance gate $\boldsymbol{\Gamma}$ is the shared primitive; VGN applies it during training, while the decomposition and VGMU consume ensemble statistics at inference. VGMU requires only the per-member means and standard deviations of any ensemble output $\mathbf{p}$ and does not depend on the gated prediction $\mathbf{q}$.

| Component | Stage | Input $\to$ Output | Intended use |
|---|---|---|---|
| Variance gate $\boldsymbol{\Gamma}$ | shared primitive | $(\bar{\mathbf{p}}, \mathbf{s}, \mathbf{k}) \to \boldsymbol{\Gamma} \in [0,1)^C$ | per-class reliability weight |
| VGN layer | training | $(\mathbf{p}_m, \boldsymbol{\Gamma}) \to \mathbf{q}_m$ | end-to-end epistemic-aware normalization |
| VG decomposition | inference | $\{\mathbf{q}_m\} \to (\text{TU}, \text{AU}, \text{EU})$ | additive aleatoric/epistemic split |
| VGMU score | inference | top-2 $(\bar{\mathbf{p}}, \mathbf{s}) \to$ scalar | decision-focused selective prediction |

To compare our variance-gated decomposition, we computed the corresponding standard decomposition without variance-gating (*i.e.* distributions $\mathbf{p}_m$) and inter-member disagreements as unbounded EPKL (Schweighofer et al., 2023), and bounded Expected Pairwise Jensen-Shannon (EPJS) divergences using each ensemble member pair $(i, j)$, the midpoint $\mathbf{m}_{ij} = \frac{1}{2}(\mathbf{p}_i + \mathbf{p}_j)$, where pairwise measures are calculated as

$$\text{EPKL} = \frac{1}{M(M-1)} \sum_{i=1}^{M} \sum_{\substack{j=1 \\ j \neq i}}^{M} \text{D}_{\text{KL}}(\mathbf{p}_i \,\|\, \mathbf{p}_j), \qquad \text{D}_{\text{KL}}(\mathbf{p}_i \,\|\, \mathbf{p}_j) = \mathbf{p}_i^\top (\log \mathbf{p}_i - \log \mathbf{p}_j), \tag{26}$$

$$\text{EPJS} = \frac{1}{M(M-1)} \sum_{i=1}^{M} \sum_{\substack{j=1 \\ j \neq i}}^{M} \text{D}_{\text{JS}}(\mathbf{p}_i \,\|\, \mathbf{p}_j), \qquad \text{D}_{\text{JS}}(\mathbf{p}_i \,\|\, \mathbf{p}_j) = \tfrac{1}{2}\text{D}_{\text{KL}}(\mathbf{p}_i \,\|\, \mathbf{m}_{ij}) + \tfrac{1}{2}\text{D}_{\text{KL}}(\mathbf{p}_j \,\|\, \mathbf{m}_{ij}). \tag{27}$$

**Summary of components.** The framework comprises a single shared primitive and three components built on it. The variance gate $\boldsymbol{\Gamma}$ converts ensemble statistics into a per-class reliability weight; the variance-gated normalization (VGN) layer applies this gate during training to produce epistemic-aware member distributions $\mathbf{q}_m$; the variance-gated decomposition then decomposes total uncertainty into aleatoric and epistemic components from $\{\mathbf{q}_m\}$; and the variance-gated margin uncertainty (VGMU) score provides a decision-focused, inference-time measure computed from any ensemble output $\mathbf{p}$ without requiring the gated prediction $\mathbf{q}$. Table 1 summarizes the input, output, stage, and intended use of each.

## 4 Analytical Gradients for Variance-Gated Normalization

In this section, we introduce reverse-mode differentiation in the ensemble setting, using VGN to capture epistemic signals during training. We discuss vector-Jacobian products through the variance-gated normalized layer, gradients of the gate $\boldsymbol{\Gamma}$ with respect to $\bar{\mathbf{p}}, \mathbf{s}$ and a learnable per-class parameter $\mathbf{k}$. By optimizing a negative log-likelihood objective (*e.g.*, cross-entropy), the model implicitly minimizes predictive uncertainty, with a particular emphasis on reducing the epistemic component (Equation 5). The propagation of these gradients back to ensemble members provides a practical approach for variance-aware training within modern automatic differentiation frameworks.

Recall that each ensemble member produces $\mathbf{p}_m \in \Delta^{C-1}$ and all members share the same gating function $\boldsymbol{\Gamma} = 1 - e^{-\bar{\mathbf{p}}/\mathbf{ks}}$, with ensemble statistics $\bar{\mathbf{p}}, \mathbf{s}$ and $\mathbf{k}$ defined in Section 3. The gated member distribution and its normalization constant are

$$\mathbf{q}_m = \frac{\mathbf{p}_m \odot \boldsymbol{\Gamma}}{Z_m}, \qquad Z_m = \mathbf{p}_m^\top \boldsymbol{\Gamma}. \tag{28}$$

When the training objective is applied to the predicted ensemble mixture distribution (averaged across models), the loss depends only on $\bar{\mathbf{q}}$, where $\mathcal{L} = \mathcal{L}(\bar{\mathbf{q}})$ such that

$$\bar{\mathbf{q}} = \frac{1}{M} \sum_{m=1}^{M} \mathbf{q}_m \implies \frac{\partial \bar{\mathbf{q}}}{\partial \mathbf{q}_m} = \frac{1}{M}\mathbf{I}. \tag{29}$$

Then by the chain rule:

$$\frac{\partial \mathcal{L}}{\partial \mathbf{q}_m} = \frac{\partial \bar{\mathbf{q}}}{\partial \mathbf{q}_m} \frac{\partial \mathcal{L}}{\partial \bar{\mathbf{q}}} \implies \frac{1}{M}\mathbf{u}, \quad \mathbf{u} = \frac{\partial \mathcal{L}}{\partial \bar{\mathbf{q}}} \quad \text{(upstream gradient).} \tag{30}$$

This quantity is obtained by differentiating the loss objective with respect to the variance-gated mixture distribution $\bar{\mathbf{q}}$ and represents the gradient signal backpropagated through the variance-gated normalization layer.

In the following analysis we decompose the total gradient of the mixture objective into its independent paths and describe gradients that flow through the ensemble statistics (mean and variance) that parameterize the gate. These local gradients are combined to provide a general reverse-mode differentiation rule for shared ensemble gates. This defines a complete analytical foundation for backpropagating epistemic-aware gradients, in which the supervised cross-entropy loss depends only on $\bar{\mathbf{q}}$. This can be interpreted as optimizing a model by minimizing predictive uncertainty, subject to epistemic constraints (complete analytical derivations and gradient expressions available in SI S2).

### 4.1 Full Gradient Decomposition

**Proposition 4.1.** *When the objective depends on the ensemble mixture $\bar{\mathbf{q}} = \frac{1}{M}\sum_{m=1}^{M}\mathbf{q}_m$, the total gradient received by each ensemble member is*

$$\frac{\partial \mathcal{L}}{\partial \mathbf{p}_m} = \frac{1}{M}\left(\left.\frac{\partial \mathcal{L}}{\partial \mathbf{p}_m}\right|_{\mathbf{\Gamma}} + \left.\frac{\partial \mathcal{L}}{\partial \mathbf{p}_m}\right|_{\bar{\mathbf{p}}} + \left.\frac{\partial \mathcal{L}}{\partial \mathbf{p}_m}\right|_{\mathbf{s}}\right). \tag{31}$$

*Proof.* Each path represents an independent dependency of $\mathcal{L}$ on $\mathbf{p}_m$:

- **Direct normalization** (with $\mathbf{\Gamma}$ and $\mathbf{k}$ fixed): how $\mathbf{q}_m$ changes when $\mathbf{p}_m$ changes. This captures the effect of normalization on the simplex space;

- **Indirect gating *via* the mean** (with $\mathbf{s}$ and $\mathbf{k}$ fixed): how $\mathbf{q}_m$ changes when the gate $\mathbf{\Gamma}$ changes through the mean $\bar{\mathbf{p}}$; and

- **Indirect gating *via* the variance** (with $\bar{\mathbf{p}}$ and $\mathbf{k}$ fixed): how $\mathbf{q}_m$ changes when the gate $\mathbf{\Gamma}$ changes through the predictive spread $\mathbf{s}$.

By the multivariate chain rule, these contributions combine additively to provide the total gradient. $\square$

**Remark 4.1.1.** *The term "fixed" in the parameters definition denotes that the corresponding variable is held constant during partial differentiation.*

We instantiate these formulations for our variance-gate introduced in Section 3, providing analytic gradients with respect to gating statistics, $(\bar{\mathbf{p}}, \mathbf{s})$ and per-class learnable parameter $\mathbf{k}$. For the exponential variance-gate $\mathbf{\Gamma} = 1 - e^{-\bar{\mathbf{p}}/\mathbf{ks}}$, gradients with respect to the ensemble mean $\bar{\mathbf{p}}$, predictive spread $\mathbf{s}$, and sensitivity parameter $\mathbf{k}$ follow directly by the chain rule. Collectively, these derivations establish a complete analytic framework for backpropagating epistemic-aware gradients through ensemble networks, enabling end-to-end optimization under epistemic-sensitive normalization (Figure 2). The variance-gated normalization layer propagates gradients across ensemble members $\mathbf{p}_m$ through a shared gating mechanism parameterized by ensemble statistics $(\bar{\mathbf{p}}, \mathbf{s})$ and a learnable sensitivity parameter $\mathbf{k}$. Forward activations from each member contribute to the ensemble mean and variance, which adjust the shared gate $\mathbf{\Gamma}$ during backpropagation. By including a learnable parameter $\mathbf{k} = \text{softplus}(\boldsymbol{\ell})$, models can adaptively modulate the strength of epistemic signals for optimizing predictive uncertainty. See Table S2 for a summary of analytical gradients used for the variance-gated normalization framework.

We summarize the forward and backward passes of VGN in Figure 2 and describe them step-by-step below:

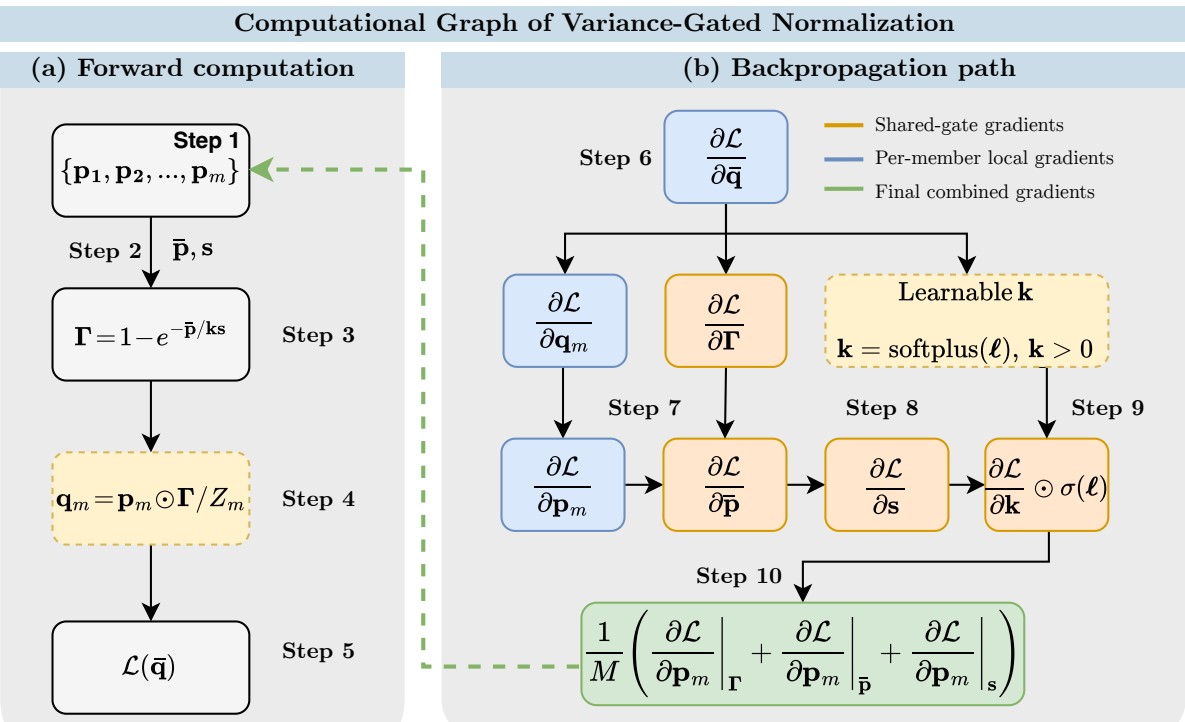

Figure 2: Computational graph of variance-gated normalization. Panel (a): Forward computation in which ensemble predictions are modulated by a shared variance gate and combined into a mixture distribution. Panel (b): Backpropagation path showing how gradients propagate through the normalization layer and shared gate *via* ensemble mean and predictive spread. See below for further step-by-step discussion.

**(a) Forward computation.** Each ensemble member $m \in \{1, \ldots, M\}$ produces a categorical predictive distribution $\mathbf{p}_m \in \Delta^{C-1}$ (Step 1). The ensemble sample mean $\bar{\mathbf{p}}$ and predictive spread $\mathbf{s}$ are then computed across members, summarizing ensemble consensus and disagreement (Step 2). Using these statistics, a shared variance gate is constructed as $\boldsymbol{\Gamma} = 1 - e^{-\bar{\mathbf{p}}/\mathbf{ks}}$, where $\mathbf{k} > 0$ controls the gate sensitivity (Step 3). Each member distribution is modulated by the shared gate and re-normalized as $\mathbf{q}_m = \mathbf{p}_m \odot \boldsymbol{\Gamma}/Z_m$, with $Z_m = \mathbf{p}_m^\top \boldsymbol{\Gamma}$ (Step 4). Finally, the ensemble mixture $\bar{\mathbf{q}} = \frac{1}{M} \sum_{m=1}^{M} \mathbf{q}_m$ is formed, and the loss $\mathcal{L}$ is applied to this mixture distribution (Step 5).

**(b) Backpropagation path.** Gradients with respect to the mixture are distributed equally to ensemble members as $\frac{\partial \mathcal{L}}{\partial \mathbf{q}_m} = \frac{1}{M} \frac{\partial \mathcal{L}}{\partial \bar{\mathbf{q}}}$ (Step 6). The total gradient with respect to each member prediction $\mathbf{p}_m$ then decomposes into three additive contributions (Proposition 4.1): (i) a direct path through the normalization with the gate held fixed, $\partial \mathcal{L}/\partial \mathbf{p}_m|_{\boldsymbol{\Gamma}}$; (ii) an indirect path through the ensemble mean $\bar{\mathbf{p}}$; and (iii) an indirect path through the predictive spread $\mathbf{s}$ (Step 7). Both indirect paths propagate through the shared variance gate *via* $\partial \boldsymbol{\Gamma}/\partial \bar{\mathbf{p}}$ and $\partial \boldsymbol{\Gamma}/\partial \mathbf{s}$, coupling gradients across ensemble members (Step 8). The sensitivity parameter $\mathbf{k}$, reparameterized as $\mathbf{k} = \text{softplus}(\boldsymbol{\ell})$, receives gradients through its influence on the gate $\boldsymbol{\Gamma}$ (Step 9). All three gradient contributions are summed and scaled by $1/M$, yielding the final per-member gradient update (Step 10).

For completeness, all Jacobians, vector–Jacobian products, and gradients with respect to ensemble statistics and learnable gating parameters are derived in full in SI S2.

## 5 Experiments

We evaluate the proposed variance-gated ensemble framework on MNIST, SVHN, CIFAR-10, and CIFAR-100 using convolutional backbones and ensemble configurations commonly employed in uncertainty estimation.

Table 2: Applicability of the variance-gated components to each evaluated ensemble framework. VGN requires separately trainable members and is therefore defined for DE and LLE only. MCD and the hybrid MCD-LLE derive their ensemble diversity from stochastic dropout passes rather than separately trainable ensemble-models, so no VGN variant is defined for them. VGMU and the variance-gated decomposition are inference-time and apply to any ensemble output.

| Framework | Separate trainable members | VGN | VGMU / decomposition | $M$ evaluated |
|---|---|---|---|---|
| DE | yes (independent networks) | DE-VGN | ✓ | 5 |
| LLE | yes (last-layer heads) | LLE-VGN | ✓ | $5, 10, 100$ |
| MCD | no (stochastic passes) | — | ✓ | $5, 10, 100$ |
| MCD-LLE | no (diversity from stochastic passes) | — | ✓ | $5, 10, 100$ |

For MNIST we use a LeNet-5 style network, while for SVHN and CIFAR-10/100 we use WideResNet-28-10. Experiments consider DE, MCD, LLE, and the hybrid variant MCD-LLE, with VGN applied where it is well-defined. VGN is a training-time normalization layer that requires distinct, trainable ensemble members. It applies to DE-VGN and LLE-VGN, but is not defined for MCD or the hybrid MCD-LLE, whose ensemble diversity is produced by stochastic dropout passes rather than separately trainable members. Since DE/DE-VGN require training $M$ independent networks, we evaluate them at $M = 5$, consistent with Lakshminarayanan et al. (2017); LLE/LLE-VGN and MCD/MCD-LLE are evaluated across $M \in \{5, 10, 100\}$. Table 2 maps each component to the ensemble frameworks it applies to, together with the ensemble sizes evaluated.

Models are trained using the Adam optimizer with early stopping, and all results are averaged over three trials with different random seeds. We compare VGMU against entropy-based EU (mutual information) and recent information-theoretic baselines, including EPKL and EPJS pairwise divergence measures. For variance-gated variants, the classwise gating parameter **k** is learned end-to-end. Uncertainty scores were assessed *via* rank-based agreement with baseline methods (Spearman's $\rho$ and Kendall's $\tau$), Cumulative Area Under the Curve ($\text{AUC}_c$) for uncertainty mass concentration, and margin–variance geometry visualizations. Predictive performance and calibration are reported using accuracy, F1-score, Expected Calibration Error (ECE), and ensemble diversity.

To quantify practical differences between methods, we report an effect size $(\bar{x}_1 - \bar{x}_2) / \max(\sigma_1, \sigma_2) \geq 1$, where bold values indicate a single best-performing value over the next nearest competitor. Complete network specifications, training protocols, evaluation definitions, and implementation details are provided in SI S3, and additional results for MNIST and SVHN appear in SI S4. MNIST and SVHN are included as complementary benchmarks to provide uncertainty behavior in low-noise (domain saturated) image settings. The influence of the principal hyperparameters $M$ (ensemble size), **k** (learnable per-class sensitivity), and ensemble type is analyzed in the sections that follow: **k** in Section 5.4 (with its theoretical sensitivity established in Proposition 3.3), $M$ in Section 5.5 and Section 5.6, and the ensemble types throughout.

The aim of these experiments is not to demonstrate state-of-the-art predictive accuracy, but to study how the uncertainty measures behave on a common set of backbones as task difficulty varies. Uncertainty measures are therefore compared under identical training conditions, and accuracy is reported as context for interpreting metric behavior rather than as the target of comparison. All claims of competitiveness are relative to the information-theoretic baselines evaluated in this controlled setting. The central finding is that VGMU and VGN match information-theoretic baselines at a fraction of their computational cost, rather than uniformly outperforming them. VGMU is therefore best understood as complementary to information-theoretic measures, capturing decision-relevant epistemic uncertainty, rather than as a replacement for them.

## 5.1 Rank Consistency with Existing Measures

To validate that VGMU captures uncertainty structure consistent with established measures, we compare its sample-level rankings against information-theoretic baselines. High correlation demonstrates consistency of uncertainty rankings, confirming that the computationally efficient margin-based VGMU score preserves the

Table 3: Spearman rank correlation ($\rho$) between VGMU and epistemic uncertainty baselines.[1,2]

| Dataset | $M$ | Method | VGMU *vs.* EPJS | VGMU *vs.* EPKL | VGMU *vs.* EU |
|---|---|---|---|---|---|
| CIFAR-10 | 5 | DE | $0.975 \pm 0.011$ | $0.928 \pm 0.032$ | $0.962 \pm 0.015$ |
| | | DE-VGN | $0.991 \pm 0.002$ | $0.977 \pm 0.004$ | $0.986 \pm 0.003$ |
| | | LLE | $0.992 \pm 0.002$ | $\mathbf{0.987 \pm 0.003}$ | $\mathbf{0.990 \pm 0.002}$ |
| | | LLE-VGN | $0.988 \pm 0.002$ | $0.981 \pm 0.004$ | $0.985 \pm 0.003$ |
| | | MCD | $0.985 \pm 0.003$ | $0.981 \pm 0.003$ | $0.982 \pm 0.003$ |
| | | MCD-LLE | $0.980 \pm 0.004$ | $0.979 \pm 0.004$ | $0.979 \pm 0.004$ |
| | 10 | LLE | $0.992 \pm 0.002$ | $0.985 \pm 0.004$ | $0.988 \pm 0.004$ |
| | | LLE-VGN | $0.993 \pm 0.002$ | $0.987 \pm 0.003$ | $\mathbf{0.990 \pm 0.002}$ |
| | | MCD | $0.989 \pm 0.002$ | $0.985 \pm 0.002$ | $0.986 \pm 0.002$ |
| | | MCD-LLE | $0.987 \pm 0.002$ | $0.986 \pm 0.002$ | $0.986 \pm 0.002$ |
| | 100 | LLE | $0.997 \pm 0.000$ | $0.974 \pm 0.004$ | $0.993 \pm 0.001$ |
| | | LLE-VGN | $0.997 \pm 0.001$ | $0.986 \pm 0.006$ | $0.993 \pm 0.002$ |
| | | MCD | $0.992 \pm 0.002$ | $0.990 \pm 0.002$ | $0.990 \pm 0.002$ |
| | | MCD-LLE | $0.990 \pm 0.002$ | $0.989 \pm 0.002$ | $0.989 \pm 0.002$ |
| CIFAR-100 | 5 | DE | $0.783 \pm 0.008$ | $0.530 \pm 0.029$ | $0.774 \pm 0.008$ |
| | | DE-VGN | $0.791 \pm 0.021$ | $0.557 \pm 0.027$ | $0.781 \pm 0.018$ |
| | | LLE | $0.870 \pm 0.003$ | $0.763 \pm 0.020$ | $0.856 \pm 0.004$ |
| | | LLE-VGN | $\mathbf{0.881 \pm 0.009}$ | $0.809 \pm 0.023$ | $\mathbf{0.867 \pm 0.010}$ |
| | | MCD | $0.868 \pm 0.012$ | $0.850 \pm 0.011$ | $0.860 \pm 0.012$ |
| | | MCD-LLE | $0.863 \pm 0.015$ | $0.849 \pm 0.015$ | $0.856 \pm 0.015$ |
| | 10 | LLE | $0.854 \pm 0.001$ | $0.566 \pm 0.035$ | $0.814 \pm 0.004$ |
| | | LLE-VGN | $0.875 \pm 0.022$ | $0.697 \pm 0.017$ | $0.843 \pm 0.023$ |
| | | MCD | $0.874 \pm 0.012$ | $0.856 \pm 0.011$ | $0.862 \pm 0.011$ |
| | | MCD-LLE | $0.874 \pm 0.015$ | $0.860 \pm 0.016$ | $0.864 \pm 0.016$ |
| | 100 | LLE | $0.724 \pm 0.012$ | $0.276 \pm 0.050$ | $0.587 \pm 0.023$ |
| | | LLE-VGN | $0.790 \pm 0.012$ | $0.279 \pm 0.033$ | $0.588 \pm 0.014$ |
| | | MCD | $0.885 \pm 0.015$ | $0.869 \pm 0.015$ | $0.872 \pm 0.015$ |
| | | MCD-LLE | $0.880 \pm 0.016$ | $0.867 \pm 0.016$ | $0.869 \pm 0.016$ |

[1] Higher values indicate better alignment with information-theoretic measures. [2] Bold values indicate a single best-performing method with an effect size $(\bar{x}_1 - \bar{x}_2) / \max(\sigma_1, \sigma_2) \geq 1$ over the next nearest competitor for each $M \in \{5, 10, 100\}$ and method per VGMU comparison.

ordering produced by more expensive methods. Table 3 reports Spearman correlations for CIFAR-10 and CIFAR-100 (additional results in SI S4.1) computed for each testing dataset and ensemble configuration.

On CIFAR-10, VGMU align with all baselines, indicating that the margin-based score captures similar uncertainty structure as pairwise divergence measures and mutual information. For the CIFAR-100 dataset, correlations remain strong for MCD variants but diverged for LLE models (*e.g.*, EPKL, $\rho = 0.566$). This reflects a deliberate design choice rather than a limitation. Pairwise measures capture distributional disagreement across all 100 classes, while VGMU focuses exclusively on the decision-relevant margin between the top-2 predictions. When ensemble members agree on the most likely classes but disagree about the tail distribution, pairwise measures increase substantially while VGMU remains low. For practical decision-making (*e.g.*, whether to trust a prediction or defer to a human for review), disagreement about distant classes is irrelevant; the insensitivity of VGMU to such disagreement is a feature, moving away from information-theoretic approaches (Wimmer et al., 2023).

The addition of VGN improves the correlation (LLE, EPKL) with $\rho = 0.566$ to $\rho = 0.697$, indicating that normalization can partially recover sensitivity to distributional disagreement. Figure 3 illustrates this behavior through rank-rank scatter plots. Points cluster along the diagonal for CIFAR-10, while CIFAR-100 displays off-diagonal deviations for LLE models. These deviations correspond to samples where ensemble members agree on the top predictions but disagree about lower-ranked classes, the cases where decision-focused design of VGMU diverges from entropy-based scores.

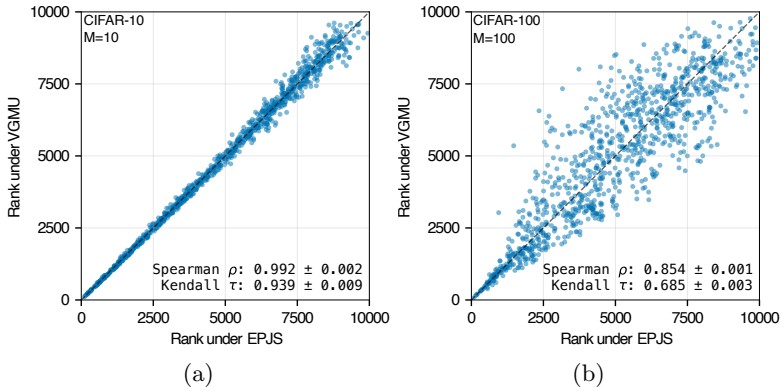

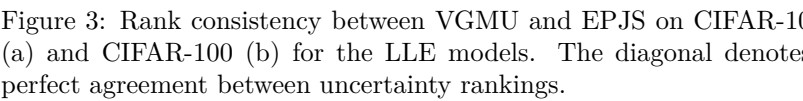

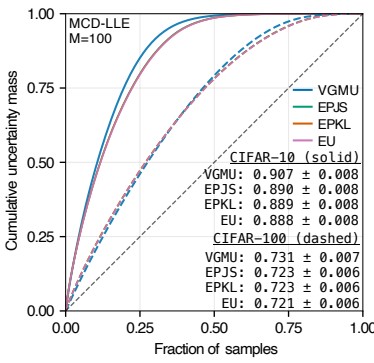

Figure 3: Rank consistency between VGMU and EPJS on CIFAR-10 (a) and CIFAR-100 (b) for the LLE models. The diagonal denotes perfect agreement between uncertainty rankings.

Figure 4: AUCc curves for CIFAR-10/100. The diagonal ($\mathrm{AUC}_c = 0.5$), corresponds to no concentration on difficult samples.

## 5.2 Uncertainty Mass Concentration

For practical deployment in selective prediction or human-in-the-loop systems, uncertainty estimates should concentrate on difficult samples; otherwise they become non-informative for decision-making under uncertainty. We quantify this through $\mathrm{AUC}_c$, calculated by sorting samples by descending uncertainty and measuring the area under the cumulative distribution. Higher $\mathrm{AUC}_c$ values indicate more efficient concentration of uncertainty on difficult samples, while $\mathrm{AUC}_c = 0.5$ no concentration. Table 4 summarizes $\mathrm{AUC}_c$ across datasets (additional information provided in SI S4.2).

On CIFAR-10, VGMU is comparable across all evaluated ensemble configurations. In several cases, the separation in mean values relative to information-theoretic baselines is larger than the observed run-to-run variability, indicating a tendency toward stronger concentration of uncertainty on difficult samples. From a practical perspective, this behavior implies that VGMU is competitive at prioritizing difficult samples. This trend is slightly less pronounced for the more challenging CIFAR-100 task. Information-theoretic measures show better performing $\mathrm{AUC}_c$ values than VGMU for most settings, reflecting the insensitivity of VGMU to disagreement across the full class simplex. This observation is consistent with the rank-correlation analysis, where information-theoretic measures capture forms of uncertainty that VGMU intentionally de-emphasizes. However, $\mathrm{AUC}_c$ can be improved using the hybrid MCD-LLE configuration (although difference within observed variability). This pattern suggests that increasing ensemble diversity through the combination of stochastic sampling and multiple classifier heads may help recover decision-relevant uncertainty structure, allowing margin-based estimation to remain competitive on more challenging tasks.

Figure 4 shows cumulative uncertainty mass concentration curves for CIFAR-10 and CIFAR-100 using MCD-LLE models ($M = 100$). For CIFAR-10, uncertainty mass is concentrated among a relatively small fraction of sample (*ca.* 25%), VGMU shows a small but consistent upward shift relative to the information-theoretic baselines. In this case of CIFAR-100, the signals are noticeably closer to the diagonal ($\mathrm{AUC}_c \approx 0.5$), indicating substantially weaker concentration. This collapse in concentration is consistent with the increased difficulty of CIFAR-100, where uncertainty is spread across a broader set of samples and full-simplex disagreement becomes more prominent. SI S4.2 results indicate a general trend where VGMU adopts a conservative uncertainty assessment. In these cases, uncertainty is distributed more broadly across samples (*i.e.*, $\mathrm{AUC}_c \approx 0.5$) rather than being concentrated, consistent with the margin-based design of VGMU.

Overall, VGMU matches the information-theoretic measures in concentration quality, while incurring lower computational cost, $O(C)$ *vs.* $O(MC)$ for mutual information and $O(M^2C)$ for the pairwise divergences (Section 5.5). The practical advantage of VGMU in these settings is efficiency rather than sharper concentration.

Table 4: Cumulative area of the curve ($\text{AUC}_c$) scores for uncertainty mass concentration. Higher values indicate sharper concentration on difficult samples.[1]

| Dataset | $M$ | Method | VGMU | EPJS | EPKL | EU |
|---------|-----|--------|------|------|------|-----|
| CIFAR-10 | 5 | DE | $0.759 \pm 0.032$ | $0.773 \pm 0.030$ | $0.781 \pm 0.029$ | $0.775 \pm 0.030$ |
| | | DE-VGN | $0.829 \pm 0.008$ | $0.838 \pm 0.007$ | $0.841 \pm 0.007$ | $0.837 \pm 0.007$ |
| | | LLE | $0.902 \pm 0.010$ | $0.898 \pm 0.011$ | $0.902 \pm 0.001$ | $0.896 \pm 0.011$ |
| | | LLE-VGN | $0.888 \pm 0.009$ | $0.887 \pm 0.007$ | $0.892 \pm 0.007$ | $0.885 \pm 0.007$ |
| | | MCD | $0.897 \pm 0.007$ | $0.895 \pm 0.007$ | $0.895 \pm 0.007$ | $0.893 \pm 0.007$ |
| | | MCD-LLE | $0.907 \pm 0.008$ | $0.901 \pm 0.007$ | $0.901 \pm 0.007$ | $0.901 \pm 0.007$ |
| | 10 | LLE | $0.885 \pm 0.015$ | $0.873 \pm 0.016$ | $0.876 \pm 0.015$ | $0.868 \pm 0.016$ |
| | | LLE-VGN | $0.885 \pm 0.009$ | $0.874 \pm 0.010$ | $0.874 \pm 0.011$ | $0.870 \pm 0.011$ |
| | | MCD | $0.895 \pm 0.007$ | $0.883 \pm 0.007$ | $0.883 \pm 0.007$ | $0.881 \pm 0.007$ |
| | | MCD-LLE | $\mathbf{0.907 \pm 0.008}$ | $0.890 \pm 0.008$ | $0.889 \pm 0.008$ | $0.888 \pm 0.008$ |
| | 100 | LLE | $0.881 \pm 0.012$ | $0.856 \pm 0.009$ | $0.829 \pm 0.001$ | $0.834 \pm 0.006$ |
| | | LLE-VGN | $0.872 \pm 0.016$ | $0.853 \pm 0.018$ | $0.838 \pm 0.024$ | $0.836 \pm 0.020$ |
| | | MCD | $0.893 \pm 0.007$ | $0.873 \pm 0.008$ | $0.871 \pm 0.008$ | $0.868 \pm 0.008$ |
| | | MCD-LLE | $\mathbf{0.907 \pm 0.008}$ | $\mathbf{0.890 \pm 0.008}$ | $\mathbf{0.889 \pm 0.008}$ | $\mathbf{0.888 \pm 0.008}$ |
| CIFAR-100 | 5 | DE | $0.549 \pm 0.002$ | $0.575 \pm 0.003$ | $0.600 \pm 0.003$ | $0.591 \pm 0.003$ |
| | | DE-VGN | $0.554 \pm 0.003$ | $0.581 \pm 0.004$ | $0.606 \pm 0.003$ | $0.596 \pm 0.004$ |
| | | LLE | $0.666 \pm 0.002$ | $0.691 \pm 0.003$ | $0.721 \pm 0.002$ | $0.696 \pm 0.003$ |
| | | LLE-VGN | $0.677 \pm 0.008$ | $0.702 \pm 0.009$ | $0.729 \pm 0.008$ | $0.706 \pm 0.009$ |
| | | MCD | $0.722 \pm 0.006$ | $0.739 \pm 0.006$ | $0.743 \pm 0.006$ | $0.740 \pm 0.006$ |
| | | MCD-LLE | $0.733 \pm 0.007$ | $\mathbf{0.745 \pm 0.006}$ | $0.747 \pm 0.006$ | $0.745 \pm 0.006$ |
| | 10 | LLE | $0.628 \pm 0.006$ | $0.643 \pm 0.008$ | $0.668 \pm 0.013$ | $0.651 \pm 0.007$ |
| | | LLE-VGN | $0.647 \pm 0.021$ | $0.666 \pm 0.020$ | $0.684 \pm 0.019$ | $0.673 \pm 0.019$ |
| | | MCD | $0.719 \pm 0.006$ | $0.726 \pm 0.005$ | $0.728 \pm 0.005$ | $0.725 \pm 0.005$ |
| | | MCD-LLE | $\mathbf{0.731 \pm 0.007}$ | $0.723 \pm 0.006$ | $0.723 \pm 0.006$ | $0.721 \pm 0.006$ |
| | 100 | LLE | $0.552 \pm 0.006$ | $0.556 \pm 0.007$ | $0.592 \pm 0.008$ | $0.565 \pm 0.006$ |
| | | LLE-VGN | $0.555 \pm 0.002$ | $0.558 \pm 0.002$ | $0.587 \pm 0.003$ | $0.566 \pm 0.002$ |
| | | MCD | $0.718 \pm 0.006$ | $0.713 \pm 0.006$ | $0.713 \pm 0.006$ | $0.711 \pm 0.006$ |
| | | MCD-LLE | $\mathbf{0.731 \pm 0.007}$ | $\mathbf{0.723 \pm 0.006}$ | $\mathbf{0.723 \pm 0.006}$ | $\mathbf{0.721 \pm 0.006}$ |

[1] Bold values indicate a single best-performing measure with an effect size $(\bar{x}_1 - \bar{x}_2) / \max(\sigma_1, \sigma_2) \geq 1$ over the next nearest competitor for each $M \in \{5, 10, 100\}$ and method.

## 5.3 Margin-Variance Geometry

The VGMU score explicitly couples the predictive margin $(\bar{p}_1 - \bar{p}_2)$ with ensemble variance $(s_1 + s_2)$, providing a two-dimensional uncertainty landscape. This geometric perspective provides insight into how different ensemble methods populate the margin-variance space and how VGMU responds to these configurations. Figure 5 illustrates the margin-variance landscape for CIFAR-100 (additional visualizations provided in SI S4.3). Several consistent patterns emerge.

**Ensemble method signatures.** Different ensemble strategies show characteristic distributions in margin-variance space. DE and DE-VGN scatter broadly, showing substantial variance even at high margins. LLE and LLE-VGN occupy regions with moderate variance and moderate-to-high margins, while MCD and MCD-LLE produce more compact clusters with lower overall variance.

**VGMU response.** The behavior of VGMU across this space confirms its sensitivity to both margin and variance. Low VGMU values (blue) occur predominantly when the predictive margin is large and variance is low, whereas high values (yellow) arise from either small margins or elevated variance. The variance-gated formulation induces smooth transitions rather than hard decision boundaries, enabling graded confidence assessments.

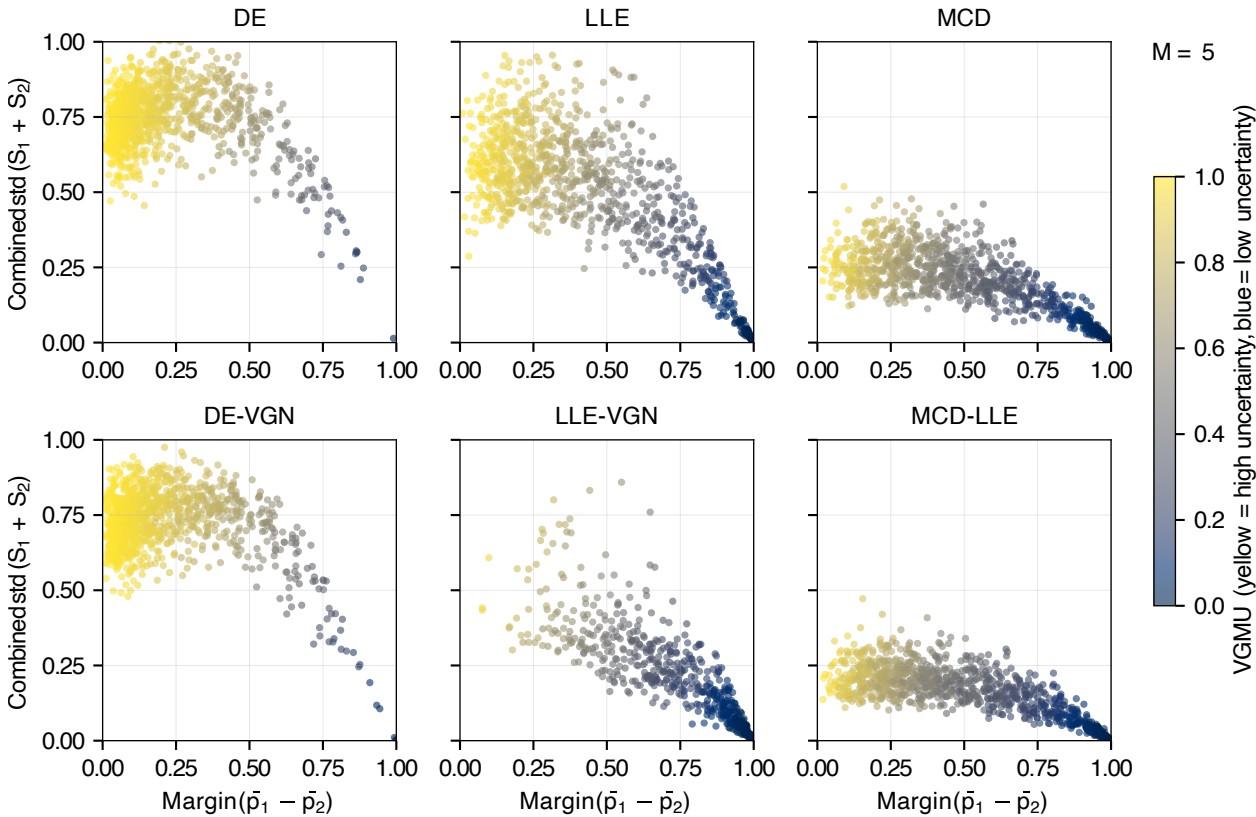

Figure 5: Margin-variance geometry for CIFAR-100 with $M = 5$. Each point represents a test sample; color indicates VGMU value (yellow = high uncertainty, blue = low). DE and DE-VGN show high variance even at large margins, LLE and LLE-VGN show moderate variance, while MCD and MCD-LLE concentrates samples in the high-margin (confident), low-variance (certain) region.

**Effect of VGN.** Comparing LLE and LLE-VGN reveals a shift toward lower VGMU values, with samples concentrating in the high-margin, low-variance region while high-uncertainty regions remain populated. This reduction in variance is achieved through VGN, which suppresses high-variance class predictions during training. Similar trends are observed across other datasets and configurations in SI S4.3, although in some cases the effects are subtle. The margin-variance geometry provides an intuitive interpretation of how, predictive margin and epistemic disagreement interact. By coupling margin and variance, VGMU assigns low uncertainty only when predictions are well-separated and consistent across ensemble members, while remaining conservative in regions characterized by ambiguity or disagreement. The effect of VGN is to reshape this geometry during training by suppressing high-variance class predictions, leading to a greater concentration of samples in the high-margin, low-variance region (Figure 5: LLE *vs.* LLE-VGN). This geometric behavior is consistent with the rank-based and concentration analyses, and shows how VGN complements VGMU by promoting decision-relevant uncertainty with conservative assessments in challenging settings.

## 5.4 Calibration and Performance

Table 5 reports calibration metrics for CIFAR-10 and CIFAR-100 (additional results in SI S4.5). For DE on CIFAR-10, VGN reduces mean ECE by a margin that exceeds the observed run-to-run variability, while also improving classification accuracy, indicating improved alignment with predictive confidence. In contrast, for CIFAR-100 and for other ensemble settings, calibration effects are smaller and less consistent, with differences falling within the variability across runs.

Table 5: Performance and calibration metrics.[1]

| Dataset | $M$ | Method | Accuracy | F1-Score | ECE | Diversity[2] |
|---|---|---|---|---|---|---|
| CIFAR-10 | 5 | DE | $0.836 \pm 0.012$ | $0.836 \pm 0.012$ | $0.049 \pm 0.014$ | $1.9 \times 10^{-2} \pm 0.2$ |
| | | DE-VGN | $\mathbf{0.875 \pm 0.005}$ | $\mathbf{0.875 \pm 0.006}$ | $\mathbf{0.023 \pm 0.003}$ | $9.8 \times 10^{-3} \pm 0.5$ |
| | | LLE | $0.855 \pm 0.002$ | $0.855 \pm 0.002$ | $0.062 \pm 0.012$ | $2.4 \times 10^{-3} \pm 0.3$ |
| | | LLE-VGN | $0.846 \pm 0.008$ | $0.845 \pm 0.008$ | $0.067 \pm 0.005$ | $2.3 \times 10^{-3} \pm 0.5$ |
| | | MCD | $0.852 \pm 0.007$ | $0.851 \pm 0.007$ | $0.057 \pm 0.002$ | $1.4 \times 10^{-3} \pm 0.1$ |
| | | MCD-LLE | $0.852 \pm 0.009$ | $0.852 \pm 0.008$ | $0.067 \pm 0.004$ | $3.4 \times 10^{-4} \pm 0.1$ |
| | 10 | LLE | $0.848 \pm 0.003$ | $0.848 \pm 0.002$ | $0.058 \pm 0.010$ | $3.6 \times 10^{-3} \pm 0.6$ |
| | | LLE-VGN | $0.849 \pm 0.004$ | $0.849 \pm 0.003$ | $0.066 \pm 0.003$ | $3.2 \times 10^{-3} \pm 0.2$ |
| | | MCD | $0.852 \pm 0.007$ | $0.851 \pm 0.006$ | $0.055 \pm 0.003$ | $1.5 \times 10^{-3} \pm 0.1$ |
| | | MCD-LLE | $0.853 \pm 0.009$ | $0.852 \pm 0.008$ | $0.067 \pm 0.004$ | $3.4 \times 10^{-4} \pm 0.2$ |
| | 100 | LLE | $0.851 \pm 0.004$ | $0.850 \pm 0.005$ | $0.065 \pm 0.006$ | $5.8 \times 10^{-3} \pm 0.8$ |
| | | LLE-VGN | $0.847 \pm 0.005$ | $0.848 \pm 0.004$ | $0.071 \pm 0.006$ | $5.4 \times 10^{-3} \pm 0.8$ |
| | | MCD | $0.853 \pm 0.008$ | $0.853 \pm 0.007$ | $0.053 \pm 0.004$ | $1.7 \times 10^{-3} \pm 0.1$ |
| | | MCD-LLE | $0.853 \pm 0.008$ | $0.852 \pm 0.008$ | $0.067 \pm 0.004$ | $3.6 \times 10^{-4} \pm 0.2$ |
| CIFAR-100 | 5 | DE | $0.487 \pm 0.002$ | $0.481 \pm 0.003$ | $0.095 \pm 0.006$ | $3.5 \times 10^{-3} \pm 0.1$ |
| | | DE-VGN | $0.507 \pm 0.007$ | $0.504 \pm 0.007$ | $0.086 \pm 0.006$ | $3.1 \times 10^{-3} \pm 0.2$ |
| | | LLE | $0.545 \pm 0.002$ | $0.541 \pm 0.003$ | $\mathbf{0.074 \pm 0.002}$ | $1.6 \times 10^{-3} \pm 0.0$ |
| | | LLE-VGN | $0.548 \pm 0.005$ | $0.545 \pm 0.007$ | $0.111 \pm 0.007$ | $1.3 \times 10^{-3} \pm 0.1$ |
| | | MCD | $0.564 \pm 0.012$ | $0.562 \pm 0.012$ | $0.092 \pm 0.001$ | $3.0 \times 10^{-4} \pm 0.2$ |
| | | MCD-LLE | $0.563 \pm 0.013$ | $0.562 \pm 0.013$ | $0.106 \pm 0.001$ | $1.9 \times 10^{-4} \pm 0.1$ |
| | 10 | LLE | $0.543 \pm 0.005$ | $0.538 \pm 0.005$ | $\mathbf{0.062 \pm 0.004}$ | $2.4 \times 10^{-3} \pm 0.1$ |
| | | LLE-VGN | $0.548 \pm 0.014$ | $0.542 \pm 0.014$ | $0.095 \pm 0.026$ | $2.0 \times 10^{-3} \pm 0.2$ |
| | | MCD | $0.567 \pm 0.012$ | $0.565 \pm 0.012$ | $0.086 \pm 0.001$ | $3.4 \times 10^{-4} \pm 0.2$ |
| | | MCD-LLE | $0.564 \pm 0.014$ | $0.563 \pm 0.014$ | $0.103 \pm 0.003$ | $2.2 \times 10^{-4} \pm 0.1$ |
| | 100 | LLE | $0.537 \pm 0.010$ | $0.533 \pm 0.012$ | $0.059 \pm 0.019$ | $4.3 \times 10^{-3} \pm 0.2$ |
| | | LLE-VGN | $0.544 \pm 0.012$ | $0.542 \pm 0.011$ | $0.059 \pm 0.009$ | $4.2 \times 10^{-3} \pm 0.1$ |
| | | MCD | $0.568 \pm 0.012$ | $0.566 \pm 0.012$ | $0.082 \pm 0.002$ | $3.7 \times 10^{-4} \pm 0.2$ |
| | | MCD-LLE | $0.565 \pm 0.013$ | $0.563 \pm 0.013$ | $0.102 \pm 0.001$ | $2.3 \times 10^{-4} \pm 0.1$ |

[1] Bold values indicate a single best-performing measure with an effect size $(\bar{x}_1 - \bar{x}_2) / \max(\sigma_1, \sigma_2) \geq 1$ over the next nearest competitor for each $M \in \{5, 10, 100\}$ and method. [2] Defined as the ensemble variance averaged across samples and classes, $\mathbb{E}_{i,c}[\mathrm{Var}_M]$.

Overall, the results indicate that incorporating VGN does not adversely affect calibration and, in several configurations, provides small but consistent improvements. Similar trends are observed for the MCD-LLE setting, where VGN tends to improve or preserve calibration rather than degrade it.

**Learned k values.** Figure 6 displays the per-class **k** parameters learned by VGN models (see Figure S5 for CIFAR-100). DE-VGN learns higher values ($\bar{k} \approx 2.5$–$4.0$) than LLE-VGN ($\bar{k} \approx 0.75$–$1.1$), reflecting the diversity in deep ensembles. Higher **k** reduces gate sensitivity (see Proposition 3.3 for additional details), allowing DE-VGN to tolerate the disagreement among independently trained members. In these examples, the relative consistency of **k** across classes suggests that it adapts to ensemble-level diversity rather than class-specific difficulty. However, results on SVHN (see Figure S4) indicate that **k** captures both effects, varying across classes while also reflecting overall ensemble diversity.

### 5.5 Computational Efficiency

A key advantage of the VGE framework is computational efficiency. Table 6 summarizes the asymptotic complexity of each uncertainty measure. VGMU requires only $O(C)$ operations after ensemble moments have been computed, compared to $O(M^2C)$ for pairwise divergence measures such as EPKL and EPJS. The VGN uncertainty decomposition (TU, AU, EU) operates at $O(MC)$, matching the cost of standard entropy-based decompositions. The quadratic scaling of pairwise methods becomes prohibitive for large ensembles. For

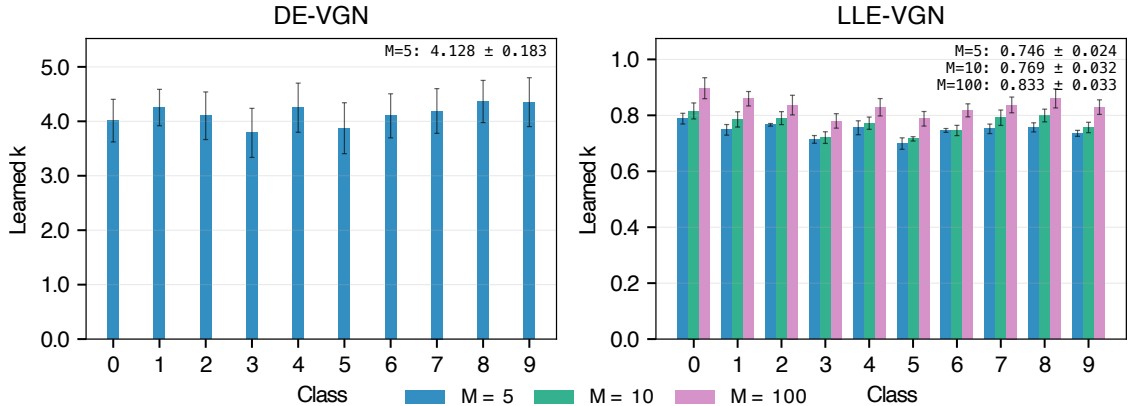

Figure 6: Learned per-class **k** values for VGN models on CIFAR-10. DE-VGN learns higher values ($\bar{k} \approx 4.1$) than LLE-VGN ($\bar{k} \approx 0.8$), reflecting adaptation to ensemble diversity.

Table 6: Asymptotic complexity of uncertainty measures for $M$ ensemble members and $C$ classes.

| Measure | Complexity | Stage |
|---|---|---|
| VGMU | $O(C)$ | Inference (post-hoc) |
| VGN decomposition (TU, AU, EU) | $O(MC)$ | Training / Inference |
| Standard entropy decomposition | $O(MC)$ | Inference |
| EPKL / EPJS | $O(M^2C)$ | Inference |

example, with $M = 100$ members and $C = 100$ classes, pairwise measures require $10^6$ operations per sample, compared to $10^4$ for VGN decomposition and $10^2$ for VGMU.

**Empirical scaling with $M$ and $C$.** To validate these asymptotic predictions empirically, we measured wall-clock inference times on synthetic ensemble outputs across a range of ensemble sizes $M \in \{2, 5, 10, 20, 50, 100\}$ and class counts $C \in \{10, 100, 1000\}$ on an NVIDIA RTX 4090 GPU (Intel Core i9-13900KF; 32 GB RAM), averaging over 10 trials of 128 samples per configuration. Figure 7 displays the log-log scaling behavior, and Table 7 reports representative timings for $C = 1000$. VGMU remains near-constant at approximately 0.5 $\mu$s per sample regardless of $M$, confirming its $O(C)$ complexity. In contrast, EPKL and EPJS exhibit the expected quadratic scaling. At $M = 100$ with $C = 100$, EPKL requires 19 $\mu$s and EPJS requires 75 $\mu$s, representing 38$\times$ and 150$\times$ speedup for VGMU, respectively. At $M = 100$ with $C = 1000$, VGMU achieves a 320$\times$ speedup over EPKL and a 1,342$\times$ speedup over EPJS. EU scales linearly in $M$ but remains within an order of magnitude of VGMU for moderate ensemble sizes; however, it diverges substantially at $M = 100$, $C = 1000$ (5.2$\times$ slower). These results confirm that the computational advantages of VGMU are not merely asymptotic but provide practical speedups of several orders of magnitude in realistic configurations, enabling real-time uncertainty estimation in deployment scenarios where pairwise methods would be prohibitive. Full results across all $M$ and $C$ combinations are provided in Table S7.

## 5.6 Out-of-Distribution Detection

To evaluate whether VGN and/or the VGMU score provides competitive OOD detection, we train models on SVHN (in-distribution, ID) and evaluate against CIFAR-10 (OOD). Detection performance is measured using the area under the ROC curve (AUC) and the false positive rate at 95% true positive rate (FPR@95). All LLE and LLE-VGN models for SVHN attained *ca.* 95% accuracy with strong calibration (ECE $\approx 0.014$). As in prior sections, we interpret differences conservatively, distinguishing trends whose magnitude exceeds run-to-run variability.

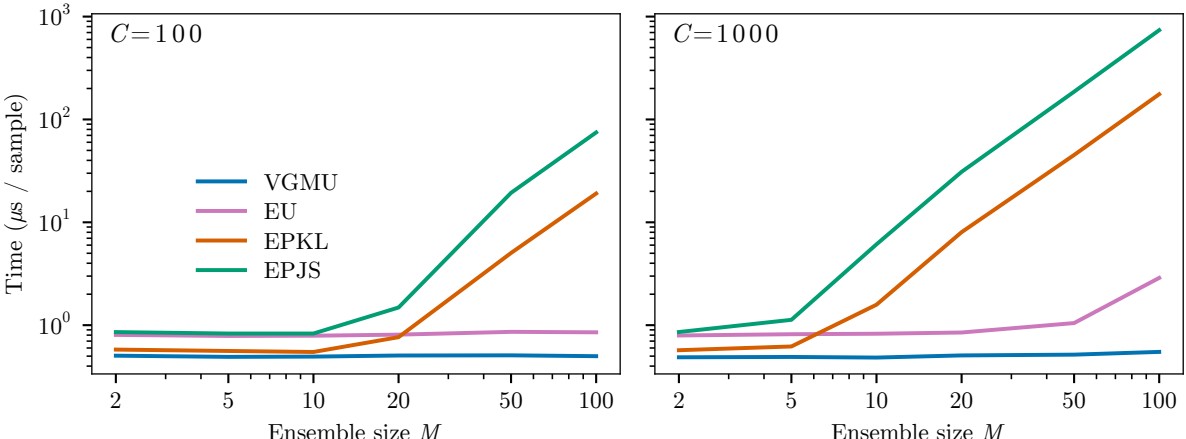

Figure 7: Wall-clock inference time per sample as a function of ensemble size $M$ for $C = 100$ classes and $C = 1000$ classes. VGMU remains constant regardless of $M$, while pairwise measures (EPKL, EPJS) result in quadratic growth consistent with their $O(M^2C)$ complexity. At $M = 100$, $C = 1000$, VGMU is over $320\times$ faster than EPKL and $1{,}343\times$ faster than EPJS.

Table 7: Wall-clock inference time ($\mu$s / sample) and VGMU speedup over baselines for representative ensemble sizes $M$ with $C = 1000$ classes.

| $M$ | Time ($\mu$s / sample) | | | | VGMU Speedup | | |
|---|---|---|---|---|---|---|---|
| | **VGMU** | **EPKL** | **EPJS** | **EU** | **EPKL** | **EPJS** | **EU** |
| 2 | 0.5 | 0.6 | 0.9 | 0.8 | 1.2 | 1.8 | 1.6 |
| 5 | 0.5 | 0.6 | 1.1 | 0.8 | 1.3 | 2.3 | 1.7 |
| 10 | 0.5 | 1.6 | 6.1 | 0.8 | 3.3 | 13 | 1.7 |
| 20 | 0.5 | 8.0 | 31 | 0.8 | 16 | 61 | 1.7 |
| 50 | 0.5 | 45 | 186 | 1.0 | 87 | 360 | 2.0 |
| 100 | 0.5 | 176 | 738 | 2.9 | 320 | 1343 | 5.2 |

**LLE and LLE-VGN across ensemble sizes.** Figure 8 reports OOD detection results for LLE and LLE-VGN across $M \in \{5, 10, 100\}$. At $M = 5$, VGMU achieves the highest mean AUC ($0.941 \pm 0.006$ for LLE; $0.942 \pm 0.006$ for LLE-VGN) and lowest mean FPR@95 ($0.173 \pm 0.027$; $0.167 \pm 0.023$, respectively). The AUC differences between VGMU and EPJS or EU fall within overlapping variability, indicating comparable detection performance among these measures. However, the separation from EPKL is more pronounced. For LLE ($M = 5$), the FPR@95 gap between VGMU ($0.173 \pm 0.027$) and EPKL ($0.256 \pm 0.049$) exceeds the observed run-to-run variability.

At larger $M$, the ID distribution over EPKL shifts into a range also occupied by a low-uncertainty subset of OOD samples, increasing ID–OOD overlap in that region (SI Figure S8). With $M = 100$, VGMU and EPJS perform comparably in AUC (*ca.*, 0.919–0.923), while EPKL degrades more substantially (AUC $= 0.891 \pm 0.004$ for LLE; $0.875 \pm 0.007$ for LLE-VGN), consistent with the unbounded range of pairwise KL amplifying tail-disagreement. Incorporating VGN provides a consistent but modest improvement in FPR@95 at small-to-moderate ensemble sizes. At $M = 5$, LLE-VGN reduces mean FPR@95 from 0.173 to 0.167; at $M = 10$, from 0.238 to 0.227. While these improvements fall within seed-level variability, the direction is consistent across both ensemble sizes.

Across all configurations, VGMU achieves comparable detection performance while avoiding the $O(M^2C)$ computational cost of pairwise divergence measures, relying instead on ensemble moments with $O(MC)$ complexity.

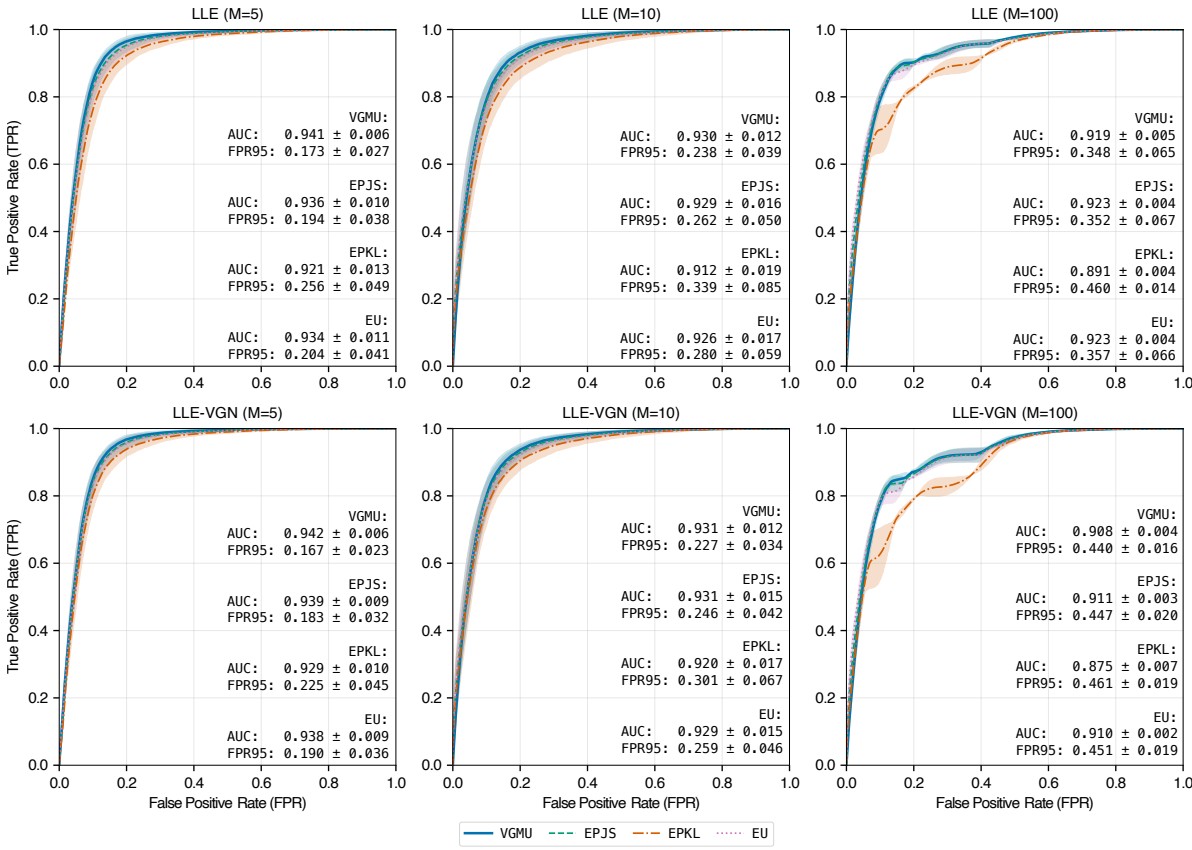

Figure 8: ROC curves for OOD detection (SVHN, ID→ CIFAR-10, OOD) across LLE (top) and LLE-VGN (bottom) with $M \in \{5, 10, 100\}$. At small ensemble sizes ($M = 5$), VGMU shows a visible separation from EPKL in both AUC and FPR@95, while differences with EPJS and EU fall within observed variability. As $M$ increases, FPR@95 degrades for all methods. VGN provides consistent but modest FPR@95 improvements at $M = 5$ and $M = 10$, with degradation at $M = 100$.

**MCD-LLE: Effect of ensemble configuration.**   To examine how the source of ensemble diversity affects OOD detection, we evaluate MCD-LLE under varying head ($H$) and sampling ($S$) configurations (SI S4.7). VGMU remains remarkably stable across all configurations, maintaining AUC = $0.934 \pm 0.006$ and FPR@95 $\approx 0.216$ regardless of the head–sampling configurations. In contrast, information-theoretic baselines improve as either $H$ or $S$ increases, with mean AUC rising from 0.911 ($M = 5$) to 0.916 ($M = 100$) for EPJS. The AUC gap between VGMU and the baselines ($\approx 0.018$–$0.023$) exceeds the observed run-to-run variability across all configurations, representing a consistent separation. This stability arises because VGMU depends only on the top-2 margin and its associated variance, both of which saturate quickly, even with modest ensemble sizes. From a practical perspective, this insensitivity to ensemble configuration is advantageous. VGMU can provide reliable OOD signals without rigorous tuning of MCD-LLE configurations.

**Summary.**   Across ensemble types and configurations, VGMU provides OOD detection performance that is comparable to or exceeds information-theoretic baselines. In the LLE setting, VGMU and EPJS/EU produce similar AUC values with overlapping variability, while VGMU consistently outperforms EPKL in FPR@95 by a margin exceeding run-to-run variation. In the MCD-LLE setting, the separation between VGMU and all baselines is more robust, with AUC differences that consistently exceed the observed standard deviations. VGN provides modest FPR@95 improvements at small ensemble sizes but does not enhance OOD detection at large $M$.

Table 8: Comparison of uncertainty decomposition frameworks. Each row corresponds to an axiom (A0–A5), and each column indicates whether a given framework satisfies an axiom[1].

| Axioms Wimmer et al. (2023) | Standard Entropy Decomposition Houlsby et al. (2011) | Expected Pairwise CE/KL Divergence Schweighofer et al. (2023) | Variance-Gated Normalization (proposed) |
|---|---|---|---|
| **A0**: TU, AU, EU $\geq 0$ (non-negativity) | ✓ | ✓ | ✓ |
| **A1**: EU $= 0$ for identical ensemble members. | ✓ | ✓ | ✓ |
| **A2**: EU, TU maximal when ensembles are uniform distributions. | ✗ | ◑ | ◑ |
| **A3**: EU$^{\uparrow}$, TU$^{\uparrow}$ with mean-preserving variance increases. | ✓ | ✓ | ◑ |
| **A4**: AU$^{\uparrow}$, TU$^{\uparrow}$ with addition of uniform noise. | ✗ | ◑ | ✓ |
| **A5**: EU invariant to variance-preserving location shifts. | ✓ | ✓ | ✓ |

[1] ✓ Satisfied, ✗ violated, ◑ partially satisfied.

## 6 Discussion

### 6.1 Axiomatic Analysis

We introduced VGE as a computationally efficient alternative to entropy- and divergence-based uncertainty decomposition, with sensitivity to epistemic disagreement. Table 8 positions VGN within the axiomatic framework of Wimmer et al. (2023). These axioms highlight where standard entropy-based decompositions fail under finite ensembles and where disagreement-based alternatives improve epistemic sensitivity (for additional examples and discussion, see SI S6; formal proofs and justifications of axiomatic compliance for each axiom A0–A5 are provided in SI S7). VGN satisfies non-negativity (A0) and vanishing epistemic uncertainty for identical ensemble members (A1). This behavior is reflected in stable uncertainty estimates on low-noise datasets such as MNIST and in confident predictions on CIFAR-10 when ensemble members agree, consistent with theoretical expectations and information-theoretic baselines. VGN satisfies the EU component of the mean-preserving variance axiom (A3), consistent with pairwise divergence methods. The TU component is not satisfied, as suppression of high-variance classes by the gate can reduce total predictive entropy even as epistemic uncertainty increases (see SI S7). The margin-variance geometry in Section 5.3 further supports this interpretation. Samples with increased ensemble spread are consistently assigned higher VGMU values, even when mean confidence remains high. Standard entropy-based decompositions violate axioms A2 and A4, relating to the behavior under uniform predictions and injected noise. VGN partially satisfies A2 (with maximal EU attained for vertex-spanning ensembles but not guaranteed for all uniform-mean configurations due to **k**-dependence) and fully satisfies A4 (AU and TU increase under center-shift). These improvements over entropy-based decompositions help explain the weaker alignment and reduced uncertainty mass concentration observed for mutual-information-based epistemic uncertainty in Section 5.2. VGN addresses these limitations by suppressing high-variance class probabilities prior to normalization, reshaping ensemble predictions using moment-based statistics rather than relying exclusively on entropy identities.

Invariance to variance-preserving location shifts (A5), satisfied by both pairwise divergence methods and VGN, explains why VGMU remains stable under changes in tail-class disagreement. As shown in Section 5.1, this invariance accounts for the divergence between VGMU and full-simplex disagreement measures

on CIFAR-100. When ensemble members agree on the most likely classes but differ in low-probability regions, pairwise divergences increase substantially, whereas VGMU remains low. VGMU is decision-focused, quantifying uncertainty through the margin between the top-ranked classes modulated by predictive variance. Disagreement that does not affect the decision boundary is intentionally de-emphasized, aligning uncertainty estimation with selective prediction and human-in-the-loop decision-making.

The axiomatic analysis and experimental results together explain why variance-gated ensembles retain the desirable properties of disagreement-based uncertainty measures while avoiding their principal computational drawbacks, operating at $O(MC)$ for uncertainty decomposition and $O(C)$ for VGMU evaluation. The learned sensitivity parameter $\mathbf{k}$ adapts to ensemble diversity and task difficulty, supporting the interpretation of variance-gating as a data-driven mechanism for epistemic control.

## 6.2 Limitations and Future Work

On CIFAR-100, VGMU shows weaker $\text{AUC}_c$ concentration and lower rank correlation with full-simplex measures compared to CIFAR-10. This reduction is a direct consequence of the top-2 margin design. On a 100-class problem, pairwise measures capture disagreement across all classes, while VGMU intentionally de-emphasizes disagreement about low-ranked classes. This represents an expected trade-off between decision-focused efficiency and full-simplex sensitivity, rather than a failure of the method. Applications requiring sensitivity to distributional disagreement beyond the decision boundary may benefit from hybrid approaches combining margin-based and distributional signals.

More precisely, the top-2 restriction is inappropriate whenever decision-relevant information resides outside the leading margin. Consider the following examples: First, many-class problems where the decision-relevant uncertainty lies in the tail. When ensemble members agree on the top-2 classes but disagree across the remaining classes, VGMU is low by design, even though a full-simplex measure would report high disagreement. Second, tasks whose target is the full predictive distribution rather than a top-1 decision, such as calibrating the entire class posterior or risk-based decisions where the penalty for an error depends on which class is involved. In these tasks, reducing the output to a scalar margin may discard the quantity of interest. Third, distributional-shift and out-of-distribution detection, where the shift appears mainly in tail-class mass. In each case, a hybrid score that augments the top-2 margin with a full-simplex term, applied only when the margin is ambiguous, is an extension that would preserve the $O(C)$ common case while restoring full-distribution sensitivity where it is needed.

The proposed framework suggests several additional directions for future work. First, the current formulation relies on first- and second-order ensemble moments. While this enables linear-time computation, higher-order statistics or alternative measures may capture additional structure in multimodal predictive distributions. Second, the gating function itself is defined using a specific exponential form. This choice was motivated by smoothness, monotonicity with respect to confidence and variance, and well-behaved gradients for end-to-end optimization. Alternative gating functions or parameterizations may provide different trade-offs between sensitivity and saturation, and exploring such designs remains an open direction. Finally, extending variance-gating to prediction tasks with severe class imbalance remains an open problem. In these settings, decision margins may be ill-defined and ensemble variance may conflate epistemic uncertainty with class-frequency effects. Exploring interactions between VGN and other epistemic-aware training objectives, such as diversity-promoting loss functions, is another promising direction, as these objectives may interact non-trivially through shared gradient pathways.

## 7 Conclusion

We introduced VGE, an epistemic-aware uncertainty framework that leverages ensemble disagreement through a signal-to-noise gating principle. VGE unifies two complementary components operating at different stages of the learning pipeline: VGMU, a lightweight decision-based uncertainty score applicable at inference without retraining, and VGN, a differentiable normalization layer that modulates high-variance predictions during training. We derived closed-form vector–Jacobian products that enable end-to-end optimization of VGN through ensemble sample means and variances, allowing uncertainty sensitivity to be

learned directly from data. Empirically, across MNIST, SVHN, CIFAR-10, and CIFAR-100, VGMU exhibited strong alignment with information-theoretic uncertainty measures, while revealing a distinct behavior on CIFAR-100. In particular, where pairwise divergence measures emphasize full-simplex disagreement, VGMU prioritizes the decision-relevant margin between top-ranked classes. Despite this difference, VGMU achieved comparable uncertainty ranking and out-of-distribution detection performance at a fraction of the computational cost, enabling real-time uncertainty estimation. Incorporating VGN during training further suppressed high-variance predictions and reshaped ensemble member distributions, with learned sensitivity parameters adapting to ensemble diversity across models. Overall, VGE provides a computationally efficient approach to epistemic uncertainty estimation, supporting post hoc evaluation, training-time distribution shaping, and end-to-end integration. These results suggest that decision-focused margin structure offers a practical alternative to pairwise divergence-based uncertainty, particularly in large-class settings. We emphasize that these gains are not uniform across settings. VGE matches information-theoretic baselines at a fraction of their cost rather than uniformly improving on them, and is best positioned as a complementary, decision-focused alternative rather than a replacement for full-simplex measures. These results suggest that decision-focused margin structure offers a practical alternative to pairwise divergence-based uncertainty, particularly when computational cost is the constraint in large-ensemble or many-class settings.

### Broader Impact Statement

Reliable uncertainty estimation is important for deploying machine learning systems in risk-sensitive settings. This work introduces a computationally efficient framework for epistemic-aware uncertainty estimation in ensemble models, enabling real-time uncertainty assessment in large ensembles and many-class problems. As with all uncertainty estimates, the outputs should be used as supporting signals rather than as standalone decision criteria. Overall, this work aims to advance the safe, efficient, and responsible deployment of machine learning models by making epistemic uncertainty estimation more accessible and scalable.

### Author Contributions

**H. Martin Gillis:** Conceptualization – Ideas; Methodology – Development or design of methodology; creation of models; Software; Writing – original draft; **Isaac Xu:** Conceptualization – Ideas; Writing – review and editing; **Thomas Trappenberg:** Supervision; Writing – review and editing.

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

# Supporting Information

## Variance-Gated Ensembles: An Epistemic-Aware Framework for Uncertainty Estimation

**H. Martin Gillis, Isaac Xu, and Thomas Trappenberg**

*Faculty of Computer Science, Dalhousie University, 6050 University Avenue, Halifax, NS B3H 4R2, Canada*

*E-mail: tt@cs.dal.ca (Thomas Trappenberg)*

## Table of Contents

## S1 Symbols and Abbreviations

Table S1: Symbols and abbreviations used in the variance-gated normalization framework.

| Symbol | Domain or Type | Description |
|---|---|---|
| $C$ | $\mathbb{N}$ | Number of classes. |
| $M$ | $\mathbb{N}$ | Number of ensemble members. |
| $\Delta^{C-1}$ | simplex | Probability simplex $\{\mathbf{z} \in \mathbb{R}^C \geq 0 : \mathbf{1}^\top \mathbf{z} = 1\}$. |
| $\mathbf{x}$ | input | Input features. |
| $\mathbf{y}$ | categorical | Class label over $C$ classes. |
| $\mathbf{w}_m$ | parameters | Parameters of ensemble member $m \in \{1, \dots, M\}$. |
| $\mathbf{1}$ | $\mathbb{R}^C$ | All-ones vector. |
| $\mathbf{I}$ | $\mathbb{R}^{C \times C}$ | Identity matrix. |
| $\odot$ | operator | Hadamard (elementwise) product. |
| $\mathrm{Diag}(\cdot)$ | operator | Diagonal matrix with the given vector on the diagonal. |
| $\log(\cdot)$ | function | Base-2 logarithm (applied elementwise to vectors). |
| $\mathbf{p}_m$ | $\Delta^{C-1}$ | Member-$m$ predictive distribution $p(\mathbf{y} \mid \mathbf{x}, \mathbf{w}_m)$. |
| $p_m(c)$ | $[0, 1]$ | Probability for class $c$ from member $m$. |
| $\bar{\mathbf{z}}$ | $\Delta^{C-1}$ | Ensemble sample mean (per-class). |
| $\mathbf{S}$ | $\mathbb{R}^C_{\geq 0}$ | Per-class ensemble standard deviation (predictive spread). |
| $\varepsilon$ | $\mathbb{R}_{>0}$ | Small constant for numerical stability (*e.g.*, $1.0 \times 10^{-8}$). |
| $\mathbf{s}$ | $\mathbb{R}^C_{>0}$ | Numerically stabilized predictive spread. |
| $k_c$ | $\mathbb{R}_{>0}$ | Classwise sensitivity scalar for gating. |
| $\mathbf{k}$ | $\mathbb{R}^C_{>0}$ | Per-class sensitivity vector (learnable) for gating. |
| $\mathbf{\Gamma}$ | $[0, 1)^C$ | Variance gate with vector $\mathbf{k}$. |
| $Z_m$ | $\mathbb{R}_{>0}$ | Normalization constant for member $m$. |
| $\mathbf{q}_m$ | $\Delta^{C-1}$ | Normalized variance-gated distribution for member $m$. |
| $\bar{\mathbf{q}}$ | $\Delta^{C-1}$ | Variance-gated ensemble mixture (sample mean over members). |
| H | $[0, 1]$ | Normalized Shannon entropy of $\mathbf{z} \in \Delta^{C-1}$. |
| TU | $[0, 1]$ | Total (predictive) uncertainty. |
| AU | $[0, 1]$ | Aleatoric uncertainty (expected per-member entropy). |
| EU | $[0, 1]$ | Epistemic uncertainty (disagreement). |
| $\mathbf{m}_{ij}$ | $\Delta^{C-1}$ | Midpoint distribution for members $(i, j)$. |
| $\mathrm{D_{KL}}(\mathbf{z}_i \| \mathbf{z}_j)$ | $\mathbb{R}_{\geq 0}$ | Kullback-Leibler divergence. |
| $\mathrm{D_{JS}}(\mathbf{z}_i \| \mathbf{z}_j)$ | $[0, 1]$ | Jensen-Shannon divergence (normalized). |
| EPCE | $\mathbb{R}_{\geq 0}$ | Expected pairwise cross-entropy. |
| EPKL | $\mathbb{R}_{\geq 0}$ | Expected pairwise KL divergence (disagreement). |
| EPJS | $[0, 1]$ | Expected pairwise Jensen-Shannon divergence (disagreement). |
| $\hat{y}$ | class index or "abstain" | Predicted label under the margin rule. |
| SNR | $\mathbb{R}$ | Signal-to-noise margin between top-2 classes. |
| $\gamma$ | $[0, 1)$ | Top-2 variance gate with scalar $k$. |
| VGMU | $[0, 1]$ | Variance-gated margin uncertainty (lower is more confident). |
| $\mathcal{L}$ | scalar | Training objective (*e.g.*, NLL on mixture distribution $\bar{\mathbf{q}}$). |
| $\mathbf{u}$ | $\mathbb{R}^C$ | Upstream gradient at the mixture. |
| $\boldsymbol{\ell}$ | $\mathbb{R}^C$ | Unconstrained parameter with $\mathbf{k} = \mathrm{softplus}(\boldsymbol{\ell})$. |
| $\mathrm{softplus}(\boldsymbol{\ell})$ | function | Positive reparameterization for $\mathbf{k}$. |
| $\sigma(\boldsymbol{\ell})$ | function | Logistic function; appears in $\partial\mathcal{L}/\partial\boldsymbol{\ell} = (\partial\mathcal{L}/\partial\mathbf{k}) \odot \sigma(\boldsymbol{\ell})$. |
| $b$ | index | Batch/sample index in sums (*e.g.*, $\sum_b$). |

## S2 Analytical Derivations and Gradient Expressions

This section provides complete analytical derivations of all gradients required for end-to-end training using variance-gated normalization. The results in this section support the summary expressions presented in Section 3 of the main paper. In Table S2, we provide a concise reference for the derived gradients.

Table S2: Summary of analytical gradients for the variance-gated normalization framework. Each gradient describes the partial derivative of the loss $\mathcal{L}$, the gating function $\mathbf{\Gamma}$, or intermediate statistics $(\bar{\mathbf{p}}, \mathbf{s})$ and learnable parameter $\mathbf{k}$ (*via* $\boldsymbol{\ell}$) with respect to the quantities that affect variance-aware ensemble training.

| Gradient | Description |
|---|---|
| $\dfrac{\partial \bar{\mathbf{q}}}{\partial \mathbf{q}_m} = \dfrac{1}{M}\mathbf{I}$ | Mixture sensitivity to member distribution. |
| $\dfrac{\partial \mathcal{L}}{\partial \mathbf{q}_m} = \dfrac{1}{M}\mathbf{u}, \;\; \mathbf{u} = \dfrac{\partial \mathcal{L}}{\partial \bar{\mathbf{q}}}$ | Upstream gradient distributed equally to members. |
| $\dfrac{\partial \mathbf{q}_m}{\partial \mathbf{\Gamma}} = \dfrac{1}{Z_m}\left[\mathrm{Diag}(\mathbf{p}_m) - \mathbf{q}_m\mathbf{p}_m^\top\right]$ | Jacobian $\mathbf{J}_m$ of normalized gating with respect to the gate $\mathbf{\Gamma}$. |
| $\dfrac{\partial \mathcal{L}_m}{\partial \mathbf{\Gamma}} = \dfrac{1}{MZ_m}\left[\mathbf{p}_m \odot \mathbf{u} - \mathbf{p}_m(\mathbf{q}_m^\top \mathbf{u})\right]$ | Reverse-mode VJP to the gate (per member). |
| $\dfrac{\partial \mathcal{L}}{\partial \mathbf{\Gamma}} = \dfrac{1}{M}\displaystyle\sum_{m=1}^{M}\dfrac{1}{Z_m}\left[\mathbf{p}_m \odot \mathbf{u} - \mathbf{p}_m(\mathbf{q}_m^\top \mathbf{u})\right]$ | Total gradient to shared gate across members. |
| $\dfrac{\partial \mathbf{\Gamma}}{\partial \bar{\mathbf{p}}} = \dfrac{1 - \mathbf{\Gamma}}{\mathbf{ks}}$ | Gate sensitivity to mean confidence. |
| $\dfrac{\partial \mathbf{\Gamma}}{\partial \mathbf{s}} = -\dfrac{(1 - \mathbf{\Gamma})\,\bar{\mathbf{p}}}{\mathbf{ks}^2}$ | Gate sensitivity to predictive spread. |
| $\dfrac{\partial \mathbf{\Gamma}}{\partial \mathbf{k}} = -\dfrac{(1 - \mathbf{\Gamma})\,\bar{\mathbf{p}}}{\mathbf{k}^2\mathbf{s}}$ | Gate sensitivity scale $\mathbf{k}$. |
| $\dfrac{\partial \mathcal{L}}{\partial \bar{\mathbf{p}}} = \dfrac{\partial \mathcal{L}}{\partial \mathbf{\Gamma}} \odot \dfrac{1 - \mathbf{\Gamma}}{\mathbf{ks}}$ | Backpropagation through gate *via* mean. |
| $\dfrac{\partial \mathcal{L}}{\partial \mathbf{s}} = -\dfrac{\partial \mathcal{L}}{\partial \mathbf{\Gamma}} \odot \dfrac{(1 - \mathbf{\Gamma})\,\bar{\mathbf{p}}}{\mathbf{ks}^2}$ | Backpropagation through gate *via* spread. |
| $\dfrac{\partial \mathcal{L}}{\partial \mathbf{k}} = -\displaystyle\sum_b \left(\dfrac{\partial \mathcal{L}}{\partial \mathbf{\Gamma}} \odot \dfrac{(1 - \mathbf{\Gamma})\,\bar{\mathbf{p}}}{\mathbf{k}^2\mathbf{s}}\right)_b$ | Backpropagation to $\mathbf{k}$ (sum over classes $b$). |
| $\dfrac{\partial \mathcal{L}}{\partial \boldsymbol{\ell}} = \dfrac{\partial \mathcal{L}}{\partial \mathbf{k}} \odot \sigma(\boldsymbol{\ell}), \;\; \mathbf{k} = \mathrm{softplus}(\boldsymbol{\ell})$ | Through softplus reparameterization of $\mathbf{k}$. |
| *Total per-member gradient contributions:* | |
| $\dfrac{\partial \mathcal{L}}{\partial \mathbf{p}_m} = \dfrac{1}{M}\left(\left.\dfrac{\partial \mathcal{L}}{\partial \mathbf{p}_m}\right|_{\mathbf{\Gamma}} + \left.\dfrac{\partial \mathcal{L}}{\partial \mathbf{p}_m}\right|_{\bar{\mathbf{p}}} + \left.\dfrac{\partial \mathcal{L}}{\partial \mathbf{p}_m}\right|_{\mathbf{s}}\right)$ | Sums direct and two indirect paths. |
| $\left.\dfrac{\partial \mathcal{L}}{\partial \mathbf{p}_m}\right|_{\mathbf{\Gamma}} = \dfrac{1}{MZ_m}\left[\mathbf{\Gamma} \odot \mathbf{u} - \mathbf{\Gamma}(\mathbf{q}_m^\top \mathbf{u})\right]$ | Direct (local) path through normalization. |
| $\left.\dfrac{\partial \mathcal{L}}{\partial \mathbf{p}_m}\right|_{\bar{\mathbf{p}}} = \dfrac{\partial \mathcal{L}}{\partial \bar{\mathbf{p}}} \odot \dfrac{1}{M}$ | Indirect path *via* ensemble mean. |
| $\left.\dfrac{\partial \mathcal{L}}{\partial \mathbf{p}_m}\right|_{\mathbf{s}} = \dfrac{\mathbf{p}_m - \bar{\mathbf{p}}}{M\,\mathbf{s}}$ | Indirect path *via* spread. |

### S2.1 Gradients through Ensemble Mean

This subsection derives gradients that propagate through the ensemble sample mean, which couples the loss to all ensemble members through the shared mixture distribution.

**Proposition S2.1** (Through the mean). *Given $\bar{\mathbf{p}} = \frac{1}{M} \sum_{m=1}^{M} \mathbf{p}_m$, each member receives*

$$\frac{\partial \mathcal{L}}{\partial \mathbf{p}_m}\bigg|_{\bar{\mathbf{p}}} = \frac{\partial \mathcal{L}}{\partial \bar{\mathbf{p}}} \frac{\partial \bar{\mathbf{p}}}{\partial \mathbf{p}_m} = \frac{\partial \mathcal{L}}{\partial \bar{\mathbf{p}}} \frac{1}{M} \tag{S1}$$

*Proof.* Differentiating $\bar{\mathbf{p}}$ gives $d\bar{\mathbf{p}} = \frac{1}{M} d\mathbf{p}_m$; thus the Jacobian $\partial\bar{\mathbf{p}}/\partial\mathbf{p}_m = \frac{1}{M}\mathbf{I}$. $\square$

**Remark S2.1.1.** *Every ensemble member contributes equally to the gradient path through the mean.*

### S2.2 Gradients through Ensemble Variance

This subsection derives gradients that propagate through the ensemble predictive spread, enabling variance-aware updates that emphasize ensemble members deviating from the mean.

**Proposition S2.2** (Through the variance). *Let $\mathbf{S} = \sqrt{\frac{1}{M-1} \sum_{m=1}^{M} (\mathbf{p}_m - \bar{\mathbf{p}})^2}$ and $\mathbf{s} = \mathbf{S} + \varepsilon$ where $\varepsilon = 1.0 \times 10^{-8}$ (for numerical stability), then*

$$\frac{\partial \mathcal{L}}{\partial \mathbf{p}_m}\bigg|_{\mathbf{S}} = \frac{\mathbf{p}_m - \bar{\mathbf{p}}}{M\,\mathbf{S}} \implies \frac{\partial \mathcal{L}}{\partial \mathbf{p}_m}\bigg|_{\mathbf{s}} \approx \frac{\mathbf{p}_m - \bar{\mathbf{p}}}{M\,\mathbf{s}}. \tag{S2}$$

*Proof.* Define variance as $\mathbf{v} = \frac{1}{M-1} \sum_{m=1}^{M} (\mathbf{p}_m - \bar{\mathbf{p}})^2$ such that $\mathbf{S} = \sqrt{\mathbf{v}}$. Hence,

$$\frac{\partial \mathbf{S}}{\partial \mathbf{p}_m} = \frac{1}{2\sqrt{\mathbf{v}}} \frac{\partial \mathbf{v}}{\partial \mathbf{p}_m} = \frac{1}{2\mathbf{S}} \frac{\partial \mathbf{v}}{\partial \mathbf{p}_m}. \tag{S3}$$

Using the identity $\mathbf{v} = \mathbb{E}[\mathbf{p}_m^2] - \bar{\mathbf{p}}^2$, we have

$$\frac{\partial \mathbf{v}}{\partial \mathbf{p}_m} = \frac{1}{M} 2\mathbf{p}_m - 2\bar{\mathbf{p}} \frac{\partial \bar{\mathbf{p}}}{\partial \mathbf{p}_m}. \tag{S4}$$

Recall $\partial\bar{\mathbf{p}}/\partial\mathbf{p}_m = \frac{1}{M}\mathbf{I}$ therefore,

$$\frac{\partial \mathbf{v}}{\partial \mathbf{p}_m} = \frac{2}{M}(\mathbf{p}_m - \bar{\mathbf{p}}) \implies \frac{\partial \mathbf{S}}{\partial \mathbf{p}_m} = \frac{1}{2\mathbf{S}}\frac{2}{M}(\mathbf{p}_m - \bar{\mathbf{p}}) = \frac{\mathbf{p}_m - \bar{\mathbf{p}}}{M\mathbf{S}} \approx \frac{\mathbf{p}_m - \bar{\mathbf{p}}}{M\mathbf{s}} \tag{S5}$$

$\square$

**Remark S2.2.1.** *Members farther from the ensemble mean receive proportionally larger gradients, enabling variance awareness in the backward flow of gradients.*

### S2.3 Jacobians and Vector–Jacobian Products for Shared Gates

This subsection derives the local Jacobians and corresponding vector-Jacobian products for the variance-gated normalization layer when the gate is shared across ensemble members.

**Proposition S2.3.** *Let the normalized gated member probabilities be $\mathbf{q}_m = (\mathbf{p}_m \odot \mathbf{\Gamma})/Z_m$, $Z_m = \mathbf{p}_m^\top \mathbf{\Gamma}$. Then the Jacobian $\mathbf{J}_m = \partial \mathbf{q}_m / \partial \mathbf{\Gamma} = [\mathrm{Diag}(\mathbf{p}_m) - \mathbf{q}_m \mathbf{p}_m^\top]/Z_m$.*

*Proof.* With $\mathbf{p}_m$ fixed $\partial(\mathbf{p}_m \odot \mathbf{\Gamma}) = \mathbf{p}_m \odot \partial\mathbf{\Gamma}$ and $\partial(\mathbf{p}_m^\top \mathbf{\Gamma}) = \mathbf{p}_m^\top \partial\mathbf{\Gamma}$, we differentiate with respect to $\mathbf{\Gamma}$,

$$d\mathbf{q}_m = \frac{Z_m \partial(\mathbf{p}_m \odot \mathbf{\Gamma}) - (\mathbf{p}_m \odot \mathbf{\Gamma})\partial Z}{Z_m^2} = \frac{Z_m(\mathbf{p}_m \odot \partial\mathbf{\Gamma}) - (\mathbf{p}_m \odot \mathbf{\Gamma})\mathbf{p}_m^\top \partial\mathbf{\Gamma}}{Z_m^2} = \frac{\mathbf{p}_m \odot \partial\mathbf{\Gamma}}{Z_m} - \frac{\mathbf{p}_m \odot \mathbf{\Gamma}}{Z_m^2} \odot \mathbf{p}_m^\top \partial\mathbf{\Gamma} \tag{S6}$$

Substitute $\mathbf{q}_m = (\mathbf{p}_m \odot \boldsymbol{\Gamma})/\mathbf{p}_m^\top \boldsymbol{\Gamma}$ and $\mathrm{Diag}(\mathbf{p}_m) \odot \partial \boldsymbol{\Gamma} = \mathbf{p}_m \partial \odot \boldsymbol{\Gamma}$,

$$\partial \mathbf{q}_m = \frac{1}{Z_m}\Big[\mathbf{p}_m \odot \partial \boldsymbol{\Gamma} - \mathbf{q}_m(\mathbf{p}_m^\top \partial \boldsymbol{\Gamma})\Big] \implies \mathbf{J}_m = \frac{\partial \mathbf{q}_m}{\partial \boldsymbol{\Gamma}} = \frac{1}{Z_m}\Big[\mathrm{Diag}(\mathbf{p}_m) - \mathbf{q}_m \mathbf{p}_m^\top\Big] \in \mathbb{R}^{C \times C} \tag{S7}$$

$\square$

**Corollary S2.3.1** (Jacobian-vector product, forward-mode)**.** *For any direction vector $d\boldsymbol{\Gamma} \in \mathbb{R}^C$,*

$$d\mathbf{q}_m = \left(\frac{\partial \mathbf{q}_m}{\partial \boldsymbol{\Gamma}}\right) d\boldsymbol{\Gamma} = \frac{1}{Z_m}\Big[\mathbf{p}_m \odot d\boldsymbol{\Gamma} - \mathbf{q}_m(\mathbf{p}_m^\top d\boldsymbol{\Gamma})\Big]. \tag{S8}$$

*This calculates the directional derivative $d\mathbf{q}_m$ for $d\boldsymbol{\Gamma}$ in $O(C)$ time.*

**Corollary S2.3.2** (Vector–Jacobian product, reverse mode)**.** *Since the loss is evaluated on the ensemble mixture $\bar{\mathbf{q}}$, each ensemble member receives a scaled upstream gradient $\frac{1}{M}\mathbf{u}$, where $\mathbf{u} = \partial \mathcal{L}/\partial \bar{\mathbf{q}}$. The reverse-mode gradient of $\mathcal{L}_m$ with respect to the gate is therefore*

$$\frac{\partial \mathcal{L}_m}{\partial \boldsymbol{\Gamma}} = \frac{1}{M}\left(\frac{\partial \mathbf{q}_m}{\partial \boldsymbol{\Gamma}}\right)^\top \mathbf{u} = \frac{1}{MZ_m}\Big[\mathbf{p}_m \odot \mathbf{u} - \mathbf{p}_m(\mathbf{q}_m^\top \mathbf{u})\Big]. \tag{S9}$$

*This calculates the per-member reverse-mode gradient $\partial \mathcal{L}_m/\partial \boldsymbol{\Gamma}$ in $O(C)$ time through the variance-gated normalization layer.*

**Remark S2.3.1.** *The upstream gradient $\mathbf{u}$ is identical for all ensemble members. Each ensemble member receives a scaled contribution $\frac{1}{M}\mathbf{u}$ when gradients are backpropagated.*

**Proposition S2.4.** *Let the normalized gated member probabilities be $\mathbf{q}_m = (\mathbf{p}_m \odot \boldsymbol{\Gamma})/Z_m$, $Z_m = \mathbf{p}_m^\top \boldsymbol{\Gamma}$. Then the loss with respect to member $m$ probabilities is*

$$\frac{\partial \mathcal{L}}{\partial \mathbf{p}_m}\Big|_{\boldsymbol{\Gamma}} = \frac{1}{MZ_m}\Big[\boldsymbol{\Gamma} \odot \mathbf{u} - \boldsymbol{\Gamma}(\mathbf{q}_m^\top \mathbf{u})\Big]. \tag{S10}$$

*Proof.* We define the following, apply the quotient rule on $\mathbf{q}_m$, and substitute values

$$\mathbf{p}_m \odot \boldsymbol{\Gamma} = Z_m \mathbf{q}_m, \qquad d(\mathbf{p}_m \odot \boldsymbol{\Gamma}) = \mathrm{Diag}(\boldsymbol{\Gamma}) \, d\mathbf{p}_m, \qquad dZ_m = \boldsymbol{\Gamma}^\top d\mathbf{p}_m, \qquad \boldsymbol{\Gamma} \odot \mathbf{u} = \mathrm{Diag}(\boldsymbol{\Gamma}) \, \mathbf{u} \tag{S11}$$

$$d\mathbf{q}_m = \frac{Z_m \, d(\mathbf{p}_m \odot \boldsymbol{\Gamma}) - (\mathbf{p}_m \odot \boldsymbol{\Gamma}) \, dZ_m}{Z_m^2} \implies d\mathbf{q}_m = \frac{1}{Z_m}\Big[\mathrm{Diag}(\boldsymbol{\Gamma}) \, d\mathbf{p}_m - \mathbf{q}_m (\boldsymbol{\Gamma}^\top d\mathbf{p}_m)\Big]. \tag{S12}$$

Recall $\partial \mathcal{L}/\partial \mathbf{q}_m = \frac{1}{M}\mathbf{u}$, therefore

$$d\mathcal{L} = \left(\frac{\partial \mathcal{L}}{\partial \mathbf{q}_m}\right)^\top d\mathbf{q}_m = \left(\frac{1}{M}\mathbf{u}\right)^\top d\mathbf{q}_m \implies d\mathcal{L} = \frac{1}{MZ_m}\mathbf{u}^\top\Big[\mathrm{Diag}(\boldsymbol{\Gamma}) - \mathbf{q}_m\boldsymbol{\Gamma}^\top\Big]d\mathbf{p}_m, \tag{S13}$$

$$\frac{\partial \mathcal{L}}{\partial \mathbf{p}_m}\Big|_{\boldsymbol{\Gamma}} = \frac{1}{MZ_m}\Big[\mathrm{Diag}(\boldsymbol{\Gamma}) - \boldsymbol{\Gamma}\mathbf{q}_m^\top\Big]\mathbf{u} \implies \frac{\partial \mathcal{L}}{\partial \mathbf{p}_m}\Big|_{\boldsymbol{\Gamma}} = \frac{1}{MZ_m}\Big[\boldsymbol{\Gamma} \odot \mathbf{u} - \boldsymbol{\Gamma}(\mathbf{q}_m^\top \mathbf{u})\Big]. \tag{S14}$$

$\square$

**Remark S2.4.1.** *There is no sum over members in $\partial \mathcal{L}/\partial \mathbf{p}_m|_{\boldsymbol{\Gamma}}$. Each member contributes independently through its own normalization $Z_m$.*

### S2.4 General Reverse-Mode Differentiation for Shared Ensemble Gates

This subsection combines the individual gradient paths into a reverse-mode differentiation rule for ensemble layers with shared gating mechanisms.

**Proposition S2.5.** *When the ensemble shares a common gate* $\mathbf{\Gamma}$, *the total gradient accumulates over all members. Let* $\mathbf{J}_m = \partial\mathbf{q}_m/\partial\mathbf{\Gamma}$ *and* $\mathbf{u} = \partial\mathcal{L}/\partial\bar{\mathbf{q}}$, *where* $\bar{\mathbf{q}} = \frac{1}{M}\sum_{m=1}^{M}\mathbf{q}_m$ *is the ensemble mixture distribution. Then the total gradient of the loss with respect to* $\mathbf{\Gamma}$ *is*

$$\frac{\partial\mathcal{L}}{\partial\mathbf{\Gamma}} = \frac{1}{M}\sum_{m=1}^{M}\mathbf{J}_m^\top\mathbf{u} = \frac{1}{M}\sum_{m=1}^{M}\frac{1}{Z_m}\Big[\mathbf{p}_m\odot\mathbf{u} - \mathbf{p}_m(\mathbf{q}_m^\top\mathbf{u})\Big], \qquad Z_m = \mathbf{p}_m^\top\mathbf{\Gamma}. \tag{S15}$$

*Proof.* For a single member $\mathbf{q}_m = (\mathbf{p}_m\odot\mathbf{\Gamma})/Z_m$, the differential is

$$d\mathbf{q}_m = \frac{1}{Z_m}\Big[\mathbf{p}_m\odot d\mathbf{\Gamma} - \mathbf{q}_m(\mathbf{p}_m^\top d\mathbf{\Gamma})\Big]. \tag{S16}$$

With the upstream gradient $\mathbf{u}$, the change in the loss satisfies

$$d\mathcal{L}_m = \frac{1}{M}\mathbf{u}^\top d\mathbf{q}_m = \frac{1}{MZ_m}\Big[\mathbf{u}^\top(\mathbf{p}_m\odot d\mathbf{\Gamma}) - \mathbf{u}^\top\big[\mathbf{q}_m(\mathbf{p}_m^\top d\mathbf{\Gamma})\big]\Big] \tag{S17}$$

Since $\mathbf{p}_m\odot d\mathbf{\Gamma} = \mathrm{Diag}(\mathbf{p}_m)\,d\mathbf{\Gamma}$, $\mathbf{u}^\top(\mathbf{p}_m\odot d\mathbf{\Gamma}) = \mathbf{u}^\top\mathrm{Diag}(\mathbf{p}_m)\,d\mathbf{\Gamma}$. Using the product transpose of matrices property $a^\top Bx = (B^\top a)^\top x$, we have $\mathbf{u}^\top\mathrm{Diag}(\mathbf{p}_m)\,d\mathbf{\Gamma} = \big(\mathrm{Diag}(\mathbf{p}_m)^\top\mathbf{u}\big)^\top d\mathbf{\Gamma}$. Given a diagonal matrix is symmetric, $\mathrm{Diag}(\mathbf{p}_m)^\top = \mathrm{Diag}(\mathbf{p}_m) \implies \mathrm{Diag}(\mathbf{p}_m)\mathbf{u} = \mathbf{p}_m\odot\mathbf{u}$. Therefore,

$$d\mathcal{L}_m = \frac{1}{M}\mathbf{u}^\top d\mathbf{q}_m = \frac{1}{MZ_m}\Big[(\mathbf{p}_m\odot\mathbf{u})^\top d\mathbf{\Gamma}) - (\mathbf{q}_m^\top\mathbf{u})\mathbf{p}_m^\top d\mathbf{\Gamma}\Big]. \tag{S18}$$

By identification with the total differential $d\mathcal{L} = (\partial\mathcal{L}/\partial\mathbf{\Gamma})^\top d\mathbf{\Gamma}$, we obtain

$$\frac{\partial\mathcal{L}}{\partial\mathbf{\Gamma}} = \frac{1}{M}\sum_{m=1}^{M}\frac{1}{Z_m}\Big[\mathbf{p}_m\odot\mathbf{u} - \mathbf{p}_m(\mathbf{q}_m^\top\mathbf{u})\Big], \tag{S19}$$

$\square$

**Remark S2.5.1.** *Although the proof begins from the forward differential* $d\mathbf{q}_m = (\partial\mathbf{q}_m/\partial\mathbf{\Gamma})\,d\mathbf{\Gamma}$, *the final expression corresponds to the reverse-mode vector–Jacobian product used in gradient backpropagation, where* $\mathbf{u} = \partial\mathcal{L}/\partial\bar{\mathbf{q}}$.

**Remark S2.5.2.** *Since the gate* $\mathbf{\Gamma}$ *is shared across all ensemble members, the total gradient* $\partial\mathcal{L}/\partial\mathbf{\Gamma}$ *accumulates contributions from each member distribution* $\mathbf{q}_m$. *The factor* $\frac{1}{M}$ *arises from differentiating the loss with respect to the mixture distribution* $\bar{\mathbf{q}} = \frac{1}{M}\sum_m\mathbf{q}_m$.

### S2.5 Gradients with Respect to Learnable Sensitivity Parameters

This subsection derives gradients with respect to the learnable sensitivity parameters modulating the strength of variance-gating, including the softplus reparameterization used to enforce positivity.

**Proposition S2.6.** *For the variance-gating function* $\mathbf{\Gamma} = 1 - e^{-\bar{\mathbf{p}}/\mathbf{ks}}$, *the gating scalar* $\mathbf{k} > 0$ *is learned via a softplus reparameterization* $\mathbf{k} = \mathrm{softplus}(\boldsymbol{\ell})$, *where* $\boldsymbol{\ell} \in \mathbb{R}^C$ *are unconstrained learnable parameters. The gating function* $\mathbf{\Gamma} \in \mathbb{R}^{B\times C}$ *is defined per sample and class, whereas* $\mathbf{k}, \mathbf{s} \in \mathbb{R}^C$ *are classwise parameters shared across the batch. Then the gradients of the loss are*

$$\frac{\partial\mathcal{L}}{\partial\bar{\mathbf{p}}} = \frac{\partial\mathcal{L}}{\partial\mathbf{\Gamma}}\frac{1-\mathbf{\Gamma}}{\mathbf{ks}}, \qquad \frac{\partial\mathcal{L}}{\partial\mathbf{s}} = -\frac{\partial\mathcal{L}}{\partial\mathbf{\Gamma}}\frac{(1-\mathbf{\Gamma})\bar{\mathbf{p}}}{\mathbf{ks}^2}, \qquad \frac{\partial\mathcal{L}}{\partial\mathbf{k}} = -\sum_b\left(\frac{\partial\mathcal{L}}{\partial\mathbf{\Gamma}}\frac{(1-\mathbf{\Gamma})\bar{\mathbf{p}}}{\mathbf{k}^2\mathbf{s}}\right)_b. \tag{S20}$$

*Proof.* Recall the partial derivatives

$$\frac{\partial \mathbf{\Gamma}}{\partial \bar{\mathbf{p}}} = \frac{1 - \mathbf{\Gamma}}{\mathbf{ks}}, \qquad\qquad \frac{\partial \mathbf{\Gamma}}{\partial \mathbf{s}} = -\frac{(1 - \mathbf{\Gamma})\bar{\mathbf{p}}}{\mathbf{ks}^2}, \qquad\qquad \frac{\partial \mathbf{\Gamma}}{\partial \mathbf{k}} = -\frac{(1 - \mathbf{\Gamma})\bar{\mathbf{p}}}{\mathbf{k}^2\mathbf{s}}. \qquad \text{(S21)}$$

Hence, after applying the chain rule, we obtain

$$\frac{\partial \mathcal{L}}{\partial \bar{\mathbf{p}}} = \frac{\partial \mathcal{L}}{\partial \mathbf{\Gamma}}\frac{\partial \mathbf{\Gamma}}{\partial \bar{\mathbf{p}}}, \qquad \frac{\partial \mathcal{L}}{\partial \mathbf{s}} = \frac{\partial \mathcal{L}}{\partial \mathbf{\Gamma}}\frac{\partial \mathbf{\Gamma}}{\partial \mathbf{s}}, \qquad \frac{\partial \mathcal{L}}{\partial \mathbf{k}} = \sum_b \left(\frac{\partial \mathcal{L}}{\partial \mathbf{\Gamma}}\frac{\partial \mathbf{\Gamma}}{\partial \mathbf{k}}\right)_b, \qquad \frac{\partial \mathcal{L}}{\partial \boldsymbol{\ell}} = \frac{\partial \mathcal{L}}{\partial \mathbf{k}}\frac{\partial \mathbf{k}}{\partial \boldsymbol{\ell}} = \frac{\partial \mathcal{L}}{\partial \mathbf{k}}\sigma(\boldsymbol{\ell}), \qquad \text{(S22)}$$

where $\partial \mathcal{L}/\partial \boldsymbol{\ell}$ is the gradient propagated through softplus reparameterization, which enforces positivity of $\mathbf{k}$ and modulates the gradient magnitude by the logistic factor $\sigma(\boldsymbol{\ell}) = 1/(1 + e^{-\boldsymbol{\ell}})$. $\qquad\qquad \square$

**Remark S2.6.1.** *The partial derivatives reveal a complementary effect of the gating parameters. Increasing $\bar{\mathbf{p}}$ results with a corresponding increase in $\mathbf{\Gamma}$, reinforcing confident predictions, whereas larger $\mathbf{s}$ or $\mathbf{k}$ reduce $\mathbf{\Gamma}$, suppressing highly variable predictions. Through this mechanism, the variance-gating function adaptively balances aleatoric and epistemic contributions.*

## S3 Implementation Details

**Reporting and Interpretation**   All results are presented as mean $\pm$ standard deviation over three independent training runs with different random seeds. These statistics are intended to reflect training variability rather than to establish formal statistical significance. We interpret differences conservatively, focusing on (i) consistency of trends across datasets and ensembles, and (ii) differences whose magnitude exceeds the typical run-to-run variability observed across random seeds.

### S3.1 Model Architectures

**MNIST**   For the MNIST dataset, we used a LeNet-5 style convolutional neural network (CNN) as the base architecture for DE, MCD, LLE, and MCD–LLE models. The network consists of a shared feature extractor with two convolutional blocks using $5 \times 5$ kernels, mapping from 1 to 6 channels in the first block and from 6 to 16 channels in the second. Each block applies a Tanh activation, followed by $2 \times 2$ average pooling and spatial dropout ($p = 0.1$). The convolutional blocks ends with an adaptive average pooling layer, producing a fixed 16-dimensional representation that is subsequently flattened. This representation was passed to a multilayer perceptron (MLP) consisting of a linear layer (16 to 120 units), a Tanh activation, and a dropout ($p = 0.1$). Classification is performed by a task-specific head implemented as a two-layer MLP with a $120 \rightarrow 84 \rightarrow 10$ structure, using Tanh activations and dropout between layers. For LLE variants, the classifier head $H$ is replaced by a list of 100 independent classifier heads, while all preceding layers are shared.

**SVHN, CIFAR-10, and CIFAR-100**   For SVHN, CIFAR-10, and CIFAR-100, we employ a ResNet-18 and WideResNet-28-10 (Zagoruyko & Komodakis, 2017) as the base model for DE, MCD, LLE, and MCD-LLE configurations. The network follows the standard BasicBlock for both the WideResNet networks, with spatial dropout applied within convolutional blocks. The networks applies batch normalization, ReLU, and global average pooling. The pooled representation is passed through an additional fully connected block consisting of a linear layer with batch normalization, ReLU activation, and dropout, prior to classification. Dropout is applied with probability $p = 0.1$ for SVHN and $p = 0.3$ for CIFAR-10/100. For last-layer ensemble variants, the classifier module is instantiated as a list of independent classifier heads $H$, while all preceding layers are shared, analogous to the MNIST setup.

### S3.2 Training Protocols

Models were trained using the Adam optimizer with learning rate $1.0 \times 10^{-3}$ for SVHN, CIFAR-10/100 and $1.0 \times 10^{-4}$ for MNIST with a batch size 128 and the cross-entropy loss objective. For multihead architectures, losses were averaged across heads prior to backpropagation. Training was done with early stopping based on the validation loss using a minimum delta of $1.0 \times 10^{-4}$ and a patience of 5 epochs. Each dataset was normalized using its respective mean and standard deviation prior to training. Training was performed in triplicate using different random seeds under fully deterministic settings, including cuDNN and cuBLAS routines. Experiments were conducted with ensemble sizes of 5, 10, or 100.

### S3.3 Evaluation and Uncertainty Metrics

We compare the proposed variance-gated ensemble framework against standard entropy-based uncertainty decompositions and recent information-theoretic approaches (Schweighofer et al., 2023). We report both variance-gated and non-gated variants of entropy- and divergence-based uncertainty measures, together with the proposed variance-gated margin uncertainty. We compare VGMU against: (i) standard entropy-based epistemic uncertainty computed as mutual information Houlsby et al. (2011); (ii) Expected Pairwise KL divergence Schweighofer et al. (2023); and (iii) Expected Pairwise Jensen-Shannon divergence (this work). For variance-gated variants, we report results using learned per-class **k**.

We include a diversity measure $\mathbb{E}_{i,c}[\mathrm{Var}_M]$, defined as the ensemble variance averaged across samples and classes. In addition to uncertainty and diversity metrics, we evaluate predictive performance and calibration using accuracy, F1-score, and expected calibration error.

We assess rank consistency *via* Spearman's rank correlation $\rho$ and Kendall's $\tau$. Spearman's $\rho$ measures whether samples ranked higher under one measure tend to also be ranked higher under another, capturing overall agreement between rankings. Kendall's $\tau$ quantifies pairwise ordering agreement by measuring the fraction of sample pairs whose relative ordering is preserved between two rankings, making it sensitive to local rank inversions. Together, these metrics provide complementary views of global and fine-grained ranking stability across uncertainty scores.

In addition to rank correlation, we report the cumulative area under the curve (AUCc), which summarizes how rapidly scores accumulate when samples are ordered from highest to lowest. AUCc is high when a small number of top-ranked samples account for a large fraction of the total score mass, and low when scores are more evenly distributed. This provides a complementary perspective by characterizing the concentration and sharpness of uncertainty estimates beyond ranking agreement alone.

Monte Carlo Dropout (i.e, MCD and MCD-LLE) inference sampling ($S$) was done using a dropout rate of $p = 0.1$ with $H = 1$ and $S \in \{5, 10, 100\}$ or $H \in \{1, 10\}$ and $S = 10$. For $H > 1$, feature representations were re-used to improve compute efficiency.

### S3.4 Variance-Gated Normalization Implementation

The VGN module is applied on top of ensemble predictions during training. Given an ensemble of $M$ per-member probability distributions $\mathbf{P} \in \mathbb{R}^{B \times M \times C}$, the VGN computes outputs $\mathbf{Q}$ *via* the variance-based gating mechanism. The gated output is computed as $\mathbf{Q} = (\mathbf{P} \odot \mathbf{\Gamma})/\mathbf{Z}$, where $\mathbf{Z}$ normalizes each member's distribution. The final prediction is obtained by averaging the gated distributions: $\bar{\mathbf{Q}} = \frac{1}{M} \sum_{m=1}^{M} \mathbf{Q}_m$. Gradients flow through the VGN *via* a custom backward pass that accounts for contributions through the normalization constant $\mathbf{Z}$, the mean $\bar{\mathbf{P}}$, and variance $\mathbf{s}$ pathways. The per-class gate parameters $\mathbf{k}$ are re-parameterized as $\mathbf{k} = \text{softplus}(\boldsymbol{\ell}) + \epsilon$ where $\boldsymbol{\ell}$ is initialized to 0, resulting with $\mathbf{k} \approx 0.693$ at initialization. This ensures strictly positive gate parameters and non-saturating initialization. For sensitivity of VGN to fixed *vs.* learned hyperparameter $k$, see SI S4.8

### S3.5 Numerical Stability

All operations susceptible to numerical instability include a small constant $\varepsilon$ (*e.g.*, $1.0 \times 10^{-8}$) to prevent underflow, overflow, and division-by-zero errors. Ensemble standard deviations were stabilized as $\mathbf{s} = \mathbf{S} + \varepsilon$. The variance-gating function $\mathbf{\Gamma} = 1 - e^{-\bar{\mathbf{p}}/\mathbf{ks}}$ is clamped below by $\varepsilon$ to ensure finite gradients. The per-class scaling parameter $\mathbf{k}$ is learned using a softplus reparameterization and clamped to a minimum value of $1.0 \times 10^{-3}$ to prevent gate saturation (*i.e.*, excessively large signal-to-noise ratios) and to maintain stability as $\mathbf{k} \to 0$. Normalization constants $Z_m = \mathbf{p}_m^\top \mathbf{\Gamma}$ are likewise lower-bounded by $\varepsilon$; when this bound is active, gradients are passed through unchanged to avoid disrupting backpropagation.

# S4    Additional Experimental Results

## S4.1    Rank Consistency with Existing Measures

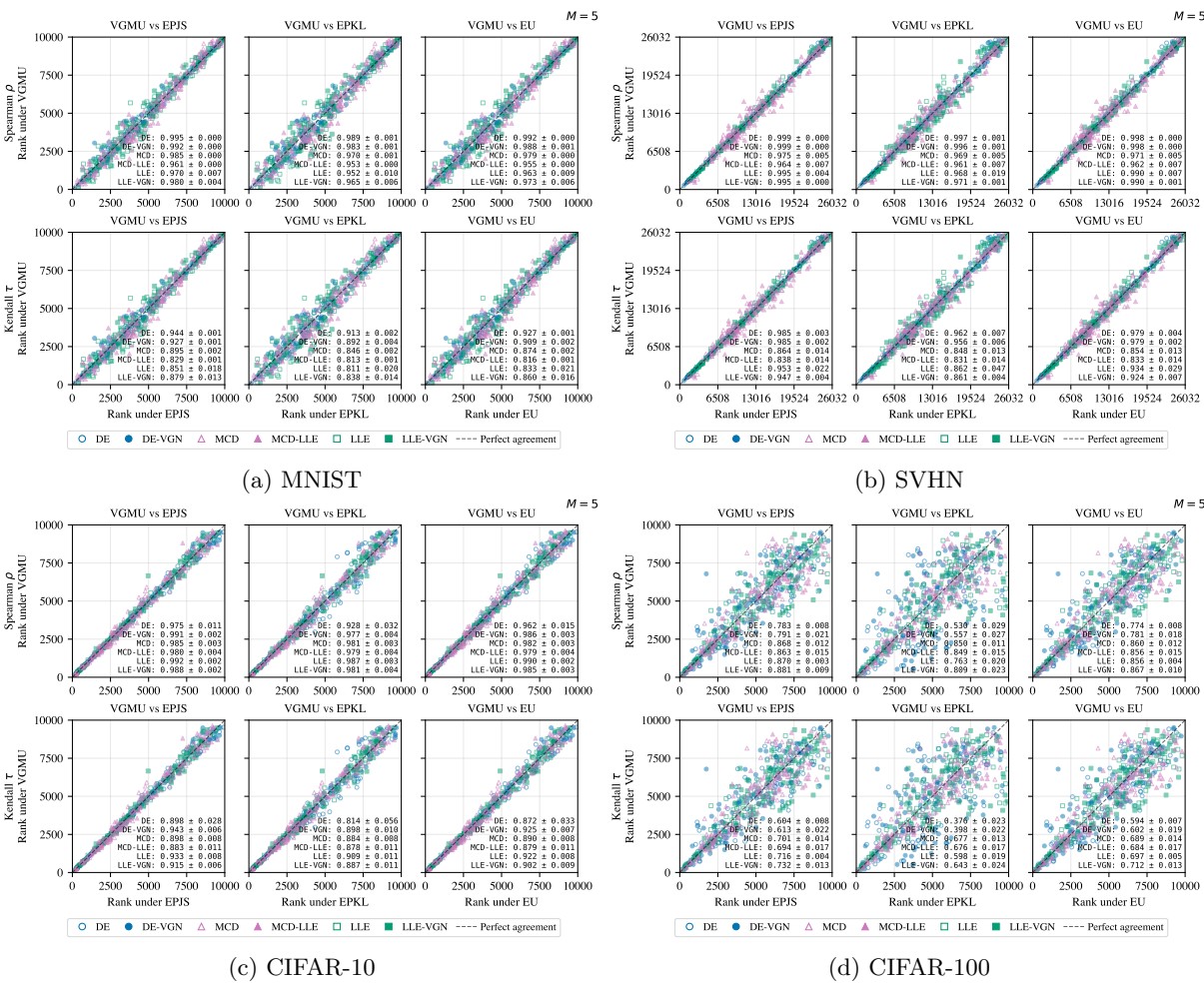

Figure S1: Spearman's $\rho$ and Kendall's $\tau$ rank correlations between VGMU and entropy- and divergence-based epistemic uncertainty measures (EPJS, EPKL, and mutual information). Results are reported across datasets and ensemble configurations, illustrating the degree of alignment between margin-based and full-simplex uncertainty rankings.

## S4.2 Uncertainty Mass Concentration

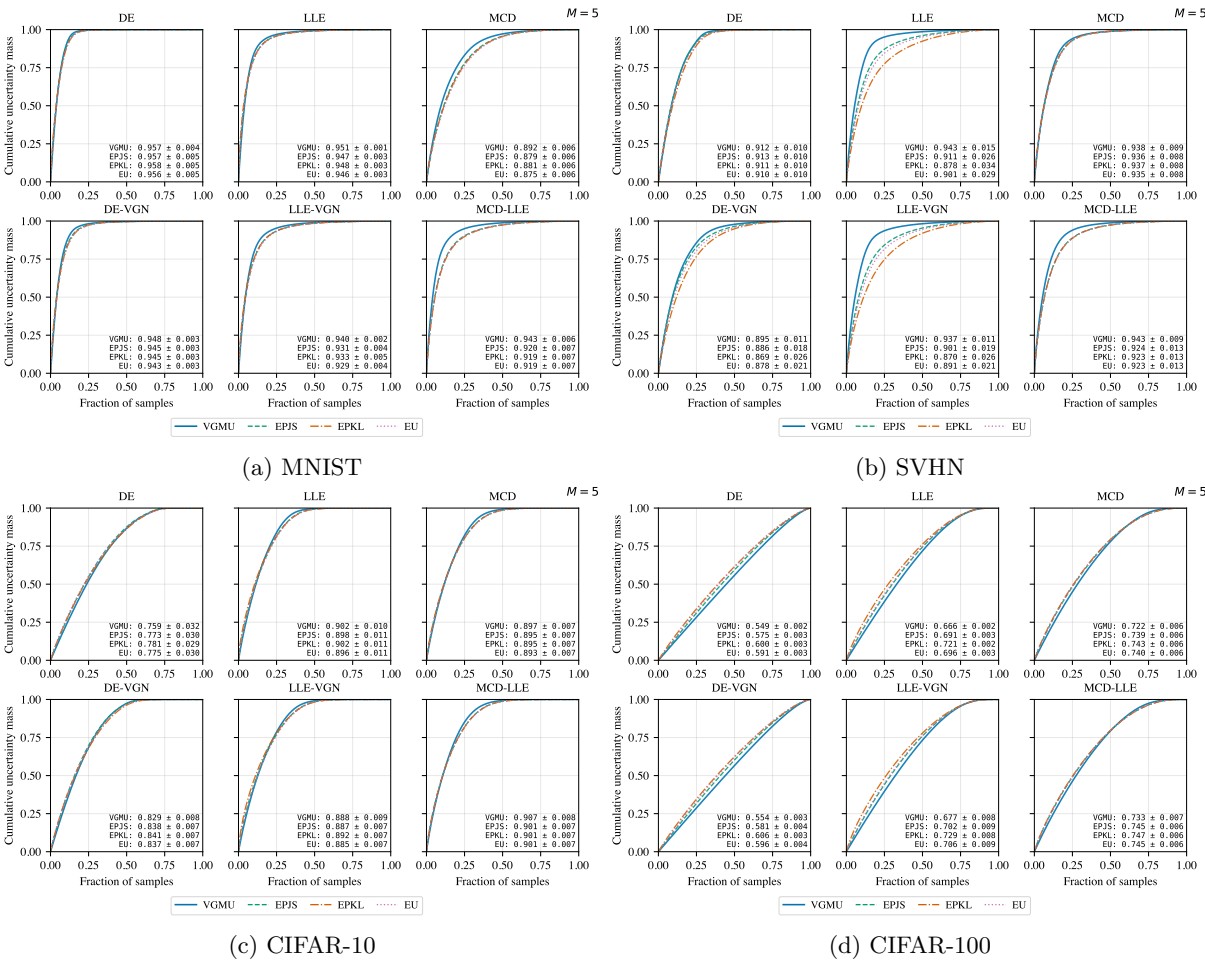

Figure S2: Cumulative uncertainty mass concentration curves and corresponding $\mathrm{AUC}_c$ values for VGMU and information-theoretic baselines. Samples are sorted by descending uncertainty, and higher $\mathrm{AUC}_c$ indicates stronger concentration of uncertainty on difficult samples. Results are shown across datasets and ensemble configurations.

## S4.3    Margin–Variance Geometry

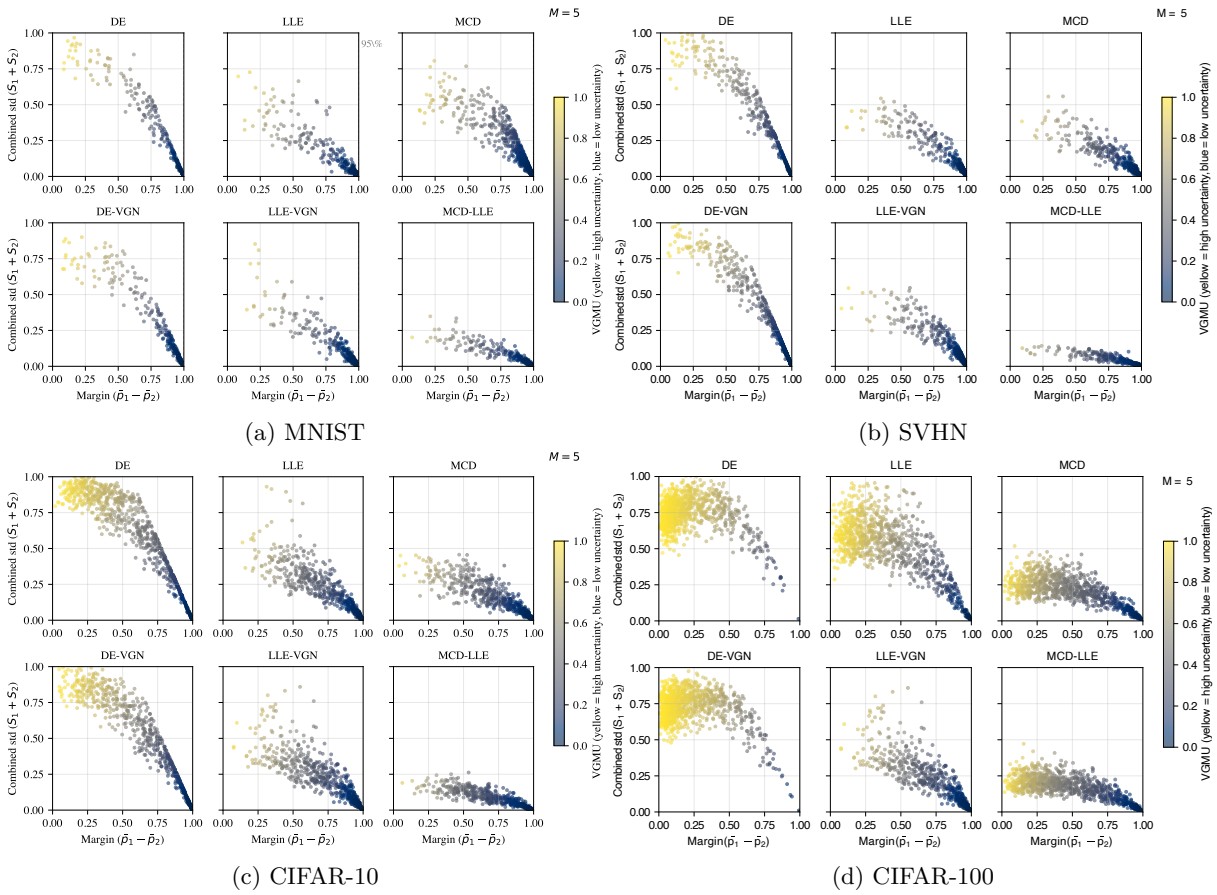

Figure S3: Visualization of predictive margin $(\bar{p}_1 - \bar{p}_2)$ and combined predictive spread $(S_1 + S_2)$ for different ensemble methods. Each point corresponds to a test sample and is colored by its VGMU value, illustrating how margin-based uncertainty couples class separability with epistemic disagreement.

## S4.4 Effects of Learned k

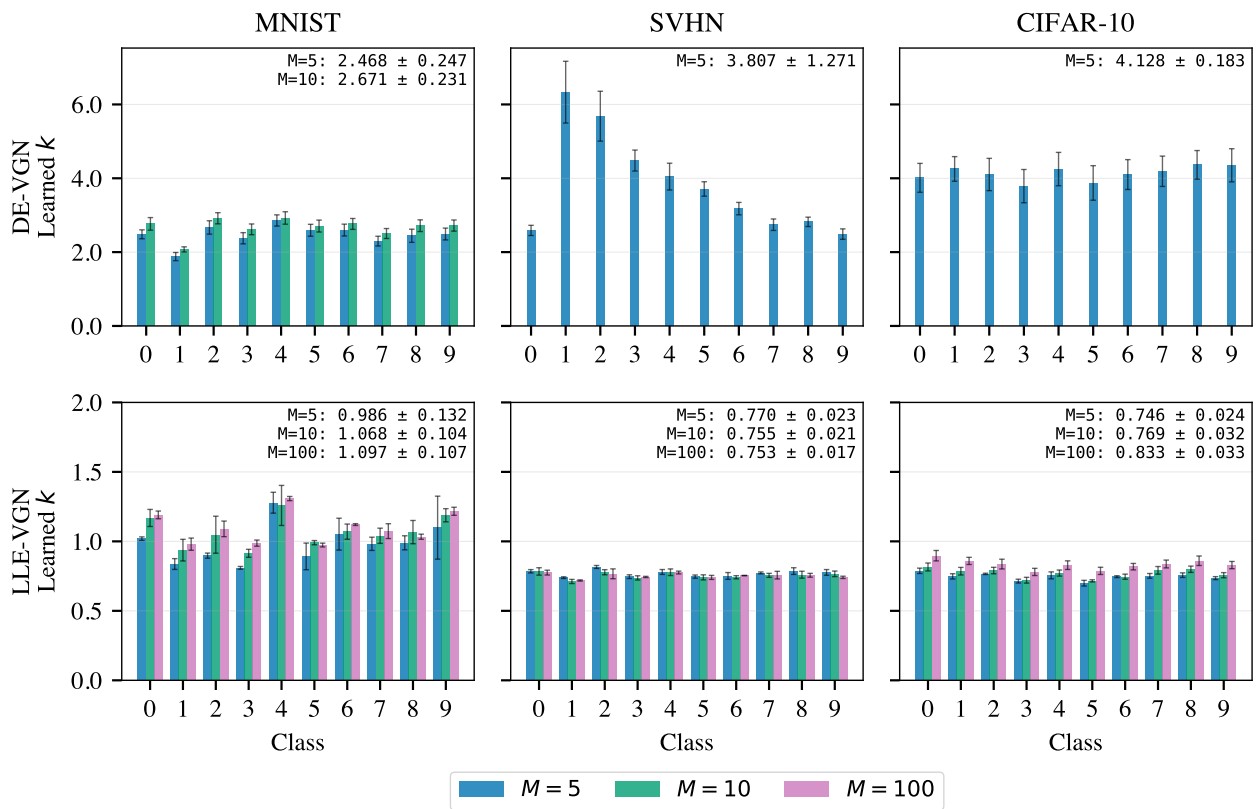

Figure S4: Learned **k**-values.

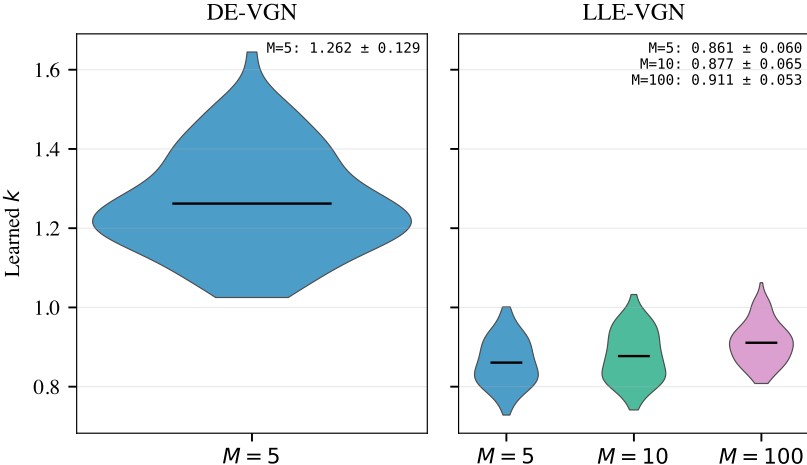

Figure S5: Learned **k**-values for CIFAR-100.

### S4.5 Performance and Calibration

Table S3: Performance and calibration metrics for ensemble models and the MNIST dataset.

| Network | Accuracy | F1-Score | ECE | Diversity |
|---|---|---|---|---|
| $M = 1$ | $0.973 \pm 0.001$ | $0.973 \pm 0.001$ | $0.004 \pm 0.001$ | — |
| MCD; $M = 5$ | $0.971 \pm 0.001$ | $0.970 \pm 0.001$ | $0.040 \pm 0.003$ | $2.7 \times 10^{-3} \pm 0.1$ |
| MCD; $M = 10$ | $0.971 \pm 0.001$ | $0.971 \pm 0.001$ | $0.041 \pm 0.005$ | $2.9 \times 10^{-3} \pm 0.1$ |
| MCD; $M = 100$ | $0.974 \pm 0.001$ | $0.973 \pm 0.001$ | $0.041 \pm 0.004$ | $3.0 \times 10^{-3} \pm 0.2$ |
| MCD-LLE; $M = 5$ | $0.973 \pm 0.001$ | $0.973 \pm 0.001$ | $0.006 \pm 0.002$ | $1.7 \times 10^{-4} \pm 0.0$ |
| MCD-LLE; $M = 10$ | $0.973 \pm 0.001$ | $0.973 \pm 0.001$ | $0.006 \pm 0.002$ | $1.9 \times 10^{-4} \pm 0.1$ |
| MCD-LLE; $M = 100$ | $0.973 \pm 0.001$ | $0.973 \pm 0.001$ | $0.006 \pm 0.002$ | $2.0 \times 10^{-4} \pm 0.1$ |
| MCD-LLE; $M = 1000$ | $0.973 \pm 0.001$ | $0.973 \pm 0.001$ | $0.006 \pm 0.002$ | $2.0 \times 10^{-4} \pm 0.1$ |
| DE; $M = 5$ | $0.981 \pm 0.001$ | $0.981 \pm 0.001$ | $0.012 \pm 0.002$ | $2.3 \times 10^{-3} \pm 0.2$ |
| DE-VGN; $M = 5$ | $0.982 \pm 0.001$ | $0.982 \pm 0.001$ | $0.007 \pm 0.002$ | $1.4 \times 10^{-3} \pm 0.1$ |
| LLE; $M = 5$ | $0.971 \pm 0.003$ | $0.971 \pm 0.003$ | $0.005 \pm 0.001$ | $6.0 \times 10^{-4} \pm 0.2$ |
| LLE; $M = 10$ | $0.972 \pm 0.001$ | $0.971 \pm 0.001$ | $0.005 \pm 0.001$ | $7.6 \times 10^{-4} \pm 0.9$ |
| LLE; $M = 100$ | $0.970 \pm 0.001$ | $0.970 \pm 0.001$ | $0.005 \pm 0.002$ | $1.0 \times 10^{-3} \pm 0.2$ |
| LLE-VGN; $M = 5$ | $0.970 \pm 0.002$ | $0.970 \pm 0.002$ | $0.005 \pm 0.001$ | $6.8 \times 10^{-4} \pm 0.4$ |
| LLE-VGN; $M = 10$ | $0.971 \pm 0.002$ | $0.971 \pm 0.002$ | $0.005 \pm 0.001$ | $8.9 \times 10^{-4} \pm 0.1$ |
| LLE-VGN; $M = 100$ | $0.970 \pm 0.003$ | $0.969 \pm 0.003$ | $0.005 \pm 0.001$ | $1.1 \times 10^{-3} \pm 0.1$ |

Table S4: Performance and calibration metrics for ensemble models and the SVHN dataset.

| Network | Accuracy | F1-Score | ECE | Diversity |
|---|---|---|---|---|
| $M = 1$ | $0.948 \pm 0.002$ | $0.943 \pm 0.003$ | $0.017 \pm 0.004$ | — |
| MCD; $M = 5$ | $0.946 \pm 0.003$ | $0.942 \pm 0.003$ | $0.010 \pm 0.004$ | $7.3 \times 10^{-4} \pm 0.8$ |
| MCD; $M = 10$ | $0.947 \pm 0.003$ | $0.942 \pm 0.003$ | $0.009 \pm 0.004$ | $8.1 \times 10^{-4} \pm 1.0$ |
| MCD; $M = 100$ | $0.947 \pm 0.003$ | $0.943 \pm 0.003$ | $0.008 \pm 0.003$ | $9.0 \times 10^{-4} \pm 1.1$ |
| MCD-LLE; $M = 5$ | $0.947 \pm 0.003$ | $0.943 \pm 0.003$ | $0.016 \pm 0.004$ | $7.3 \times 10^{-5} \pm 1.6$ |
| MCD-LLE; $M = 10$ | $0.947 \pm 0.003$ | $0.943 \pm 0.003$ | $0.016 \pm 0.004$ | $8.2 \times 10^{-5} \pm 1.9$ |
| MCD-LLE; $M = 100$ | $0.948 \pm 0.003$ | $0.943 \pm 0.003$ | $0.016 \pm 0.004$ | $8.5 \times 10^{-5} \pm 1.9$ |
| MCD-LLE; $M = 1000$ | $0.947 \pm 0.003$ | $0.943 \pm 0.003$ | $0.016 \pm 0.004$ | $8.5 \times 10^{-5} \pm 2.0$ |
| DE; $M = 5$ | $0.955 \pm 0.002$ | $0.952 \pm 0.002$ | $0.026 \pm 0.006$ | $5.8 \times 10^{-3} \pm 0.6$ |
| DE-VGN; $M = 5$ | $0.960 \pm 0.000$ | $0.957 \pm 0.000$ | $0.016 \pm 0.003$ | $4.0 \times 10^{-3} \pm 0.4$ |
| LLE; $M = 5$ | $0.950 \pm 0.002$ | $0.946 \pm 0.002$ | $0.013 \pm 0.006$ | $9.7 \times 10^{-4} \pm 1.0$ |
| LLE; $M = 10$ | $0.949 \pm 0.003$ | $0.945 \pm 0.003$ | $0.017 \pm 0.003$ | $1.3 \times 10^{-3} \pm 0.1$ |
| LLE; $M = 100$ | $0.951 \pm 0.004$ | $0.947 \pm 0.004$ | $0.012 \pm 0.002$ | $3.2 \times 10^{-3} \pm 0.3$ |
| LLE-VGN; $M = 5$ | $0.953 \pm 0.005$ | $0.949 \pm 0.006$ | $0.014 \pm 0.002$ | $8.1 \times 10^{-4} \pm 0.2$ |
| LLE-VGN; $M = 10$ | $0.952 \pm 0.002$ | $0.949 \pm 0.002$ | $0.012 \pm 0.001$ | $1.1 \times 10^{-3} \pm 0.0$ |
| LLE-VGN; $M = 100$ | $0.951 \pm 0.004$ | $0.947 \pm 0.005$ | $0.015 \pm 0.004$ | $2.5 \times 10^{-3} \pm 0.1$ |

Table S5: Performance and calibration metrics for ensemble models and the CIFAR-10 dataset.

| Network | Accuracy | F1-Score | ECE | Diversity |
|---|---|---|---|---|
| $M = 1$ | $0.853 \pm 0.008$ | $0.852 \pm 0.008$ | $0.070 \pm 0.005$ | — |
| MCD; $M = 5$ | $0.852 \pm 0.007$ | $0.851 \pm 0.007$ | $0.057 \pm 0.002$ | $1.4 \times 10^{-3} \pm 0.1$ |
| MCD; $M = 10$ | $0.852 \pm 0.007$ | $0.851 \pm 0.006$ | $0.055 \pm 0.003$ | $1.5 \times 10^{-3} \pm 0.1$ |
| MCD; $M = 100$ | $0.853 \pm 0.008$ | $0.853 \pm 0.007$ | $0.053 \pm 0.004$ | $1.7 \times 10^{-3} \pm 0.1$ |
| MCD-LLE; $M = 5$ | $0.852 \pm 0.009$ | $0.852 \pm 0.008$ | $0.067 \pm 0.004$ | $3.1 \times 10^{-4} \pm 0.1$ |
| MCD-LLE; $M = 10$ | $0.853 \pm 0.009$ | $0.852 \pm 0.008$ | $0.067 \pm 0.004$ | $3.5 \times 10^{-4} \pm 0.2$ |
| MCD-LLE; $M = 100$ | $0.853 \pm 0.008$ | $0.852 \pm 0.008$ | $0.067 \pm 0.004$ | $3.6 \times 10^{-4} \pm 0.2$ |
| MCD-LLE; $M = 1000$ | $0.853 \pm 0.008$ | $0.852 \pm 0.008$ | $0.067 \pm 0.004$ | $3.7 \times 10^{-4} \pm 0.2$ |
| DE; $M = 5$ | $0.836 \pm 0.012$ | $0.836 \pm 0.012$ | $0.049 \pm 0.014$ | $1.9 \times 10^{-2} \pm 0.2$ |
| DE-VGN; $M = 5$ | $0.875 \pm 0.005$ | $0.875 \pm 0.006$ | $0.023 \pm 0.003$ | $9.8 \times 10^{-3} \pm 0.5$ |
| LLE; $M = 5$ | $0.855 \pm 0.002$ | $0.855 \pm 0.002$ | $0.062 \pm 0.012$ | $2.4 \times 10^{-3} \pm 0.3$ |
| LLE; $M = 10$ | $0.848 \pm 0.003$ | $0.848 \pm 0.002$ | $0.058 \pm 0.010$ | $3.6 \times 10^{-3} \pm 0.5$ |
| LLE; $M = 100$ | $0.851 \pm 0.004$ | $0.850 \pm 0.005$ | $0.065 \pm 0.006$ | $5.9 \times 10^{-3} \pm 0.8$ |
| LLE-VGN; $M = 5$ | $0.846 \pm 0.008$ | $0.845 \pm 0.008$ | $0.067 \pm 0.005$ | $2.3 \times 10^{-3} \pm 0.5$ |
| LLE-VGN; $M = 10$ | $0.849 \pm 0.004$ | $0.849 \pm 0.003$ | $0.066 \pm 0.003$ | $3.2 \times 10^{-3} \pm 0.2$ |
| LLE-VGN; $M = 100$ | $0.847 \pm 0.005$ | $0.848 \pm 0.004$ | $0.071 \pm 0.006$ | $5.4 \times 10^{-3} \pm 0.8$ |

Table S6: Performance and calibration metrics for ensemble models and the CIFAR-100 dataset.

| Network | Accuracy | F1-Score | ECE | Diversity |
|---|---|---|---|---|
| $M = 1$ | $0.565 \pm 0.012$ | $0.563 \pm 0.012$ | $0.122 \pm 0.002$ | — |
| MCD; $M = 5$ | $0.564 \pm 0.012$ | $0.562 \pm 0.012$ | $0.092 \pm 0.001$ | $3.0 \times 10^{-4} \pm 0.2$ |
| MCD; $M = 10$ | $0.567 \pm 0.012$ | $0.565 \pm 0.012$ | $0.086 \pm 0.001$ | $3.4 \times 10^{-4} \pm 0.2$ |
| MCD; $M = 100$ | $0.568 \pm 0.012$ | $0.566 \pm 0.012$ | $0.082 \pm 0.002$ | $3.7 \times 10^{-4} \pm 0.2$ |
| MCD-LLE; $M = 5$ | $0.563 \pm 0.013$ | $0.562 \pm 0.013$ | $0.106 \pm 0.001$ | $1.9 \times 10^{-4} \pm 0.1$ |
| MCD-LLE; $M = 10$ | $0.564 \pm 0.014$ | $0.563 \pm 0.014$ | $0.103 \pm 0.003$ | $2.2 \times 10^{-4} \pm 0.1$ |
| MCD-LLE; $M = 100$ | $0.565 \pm 0.013$ | $0.563 \pm 0.013$ | $0.102 \pm 0.001$ | $2.3 \times 10^{-4} \pm 0.1$ |
| MCD-LLE; $M = 1000$ | $0.565 \pm 0.013$ | $0.564 \pm 0.013$ | $0.102 \pm 0.001$ | $2.3 \times 10^{-4} \pm 0.1$ |
| DE; $M = 5$ | $0.487 \pm 0.002$ | $0.481 \pm 0.003$ | $0.095 \pm 0.006$ | $3.5 \times 10^{-3} \pm 0.0$ |
| DE-VGN; $M = 5$ | $0.507 \pm 0.007$ | $0.504 \pm 0.007$ | $0.086 \pm 0.006$ | $3.1 \times 10^{-3} \pm 0.2$ |
| LLE; $M = 5$ | $0.545 \pm 0.002$ | $0.541 \pm 0.003$ | $0.074 \pm 0.002$ | $1.6 \times 10^{-3} \pm 0.0$ |
| LLE; $M = 10$ | $0.543 \pm 0.005$ | $0.538 \pm 0.005$ | $0.062 \pm 0.004$ | $2.4 \times 10^{-3} \pm 0.1$ |
| LLE; $M = 100$ | $0.537 \pm 0.010$ | $0.533 \pm 0.012$ | $0.059 \pm 0.019$ | $4.3 \times 10^{-3} \pm 0.2$ |
| LLE-VGN; $M = 5$ | $0.548 \pm 0.005$ | $0.545 \pm 0.007$ | $0.111 \pm 0.007$ | $1.3 \times 10^{-3} \pm 0.1$ |
| LLE-VGN; $M = 10$ | $0.545 \pm 0.014$ | $0.542 \pm 0.014$ | $0.095 \pm 0.026$ | $2.0 \times 10^{-3} \pm 0.2$ |
| LLE-VGN; $M = 100$ | $0.544 \pm 0.012$ | $0.542 \pm 0.011$ | $0.059 \pm 0.009$ | $4.2 \times 10^{-3} \pm 0.1$ |

### S4.6 Computational Efficiency: Full Scaling Results

Table S7: Wall-clock inference time ($\mu$s / sample) and VGMU speedup across all ensemble sizes $M$ and classes $C$.

| $M$ | $C$ | Time ($\mu$s / sample) | | | | VGMU Speedup | | |
|---|---|---|---|---|---|---|---|---|
| | | **VGMU** | **EPKL** | **EPJS** | **EU** | **EPKL** | **EPJS** | **EU** |
| 2 | 10 | 0.5 | 0.6 | 0.9 | 0.8 | 1.1 | 1.6 | 1.5 |
| 2 | 100 | 0.5 | 0.6 | 0.9 | 0.8 | 1.1 | 1.7 | 1.6 |
| 2 | 1000 | 0.5 | 0.6 | 0.9 | 0.8 | 1.2 | 1.8 | 1.6 |
| 5 | 10 | 0.5 | 0.6 | 0.8 | 0.8 | 1.1 | 1.7 | 1.6 |
| 5 | 100 | 0.5 | 0.6 | 0.8 | 0.8 | 1.1 | 1.7 | 1.6 |
| 5 | 1000 | 0.5 | 0.6 | 1.1 | 0.8 | 1.3 | 2.3 | 1.7 |
| 10 | 10 | 0.5 | 0.6 | 0.9 | 0.8 | 1.2 | 1.7 | 1.6 |
| 10 | 100 | 0.5 | 0.5 | 0.8 | 0.8 | 1.1 | 1.7 | 1.6 |
| 10 | 1000 | 0.5 | 1.6 | 6.1 | 0.8 | 3.3 | 13 | 1.7 |
| 20 | 10 | 0.5 | 0.6 | 0.9 | 0.8 | 1.1 | 1.7 | 1.6 |
| 20 | 100 | 0.5 | 0.8 | 1.5 | 0.8 | 1.5 | 2.9 | 1.6 |
| 20 | 1000 | 0.5 | 8.0 | 31 | 0.8 | 16 | 61 | 1.7 |
| 50 | 10 | 0.5 | 0.7 | 1.3 | 0.8 | 1.4 | 2.5 | 1.6 |
| 50 | 100 | 0.5 | 5.0 | 19 | 0.9 | 9.9 | 38 | 1.7 |
| 50 | 1000 | 0.5 | 45 | 186 | 1.0 | 87 | 360 | 2.0 |
| 100 | 10 | 0.5 | 1.8 | 6.8 | 0.8 | 3.4 | 13 | 1.6 |
| 100 | 100 | 0.5 | 19 | 75 | 0.9 | 38 | 150 | 1.7 |
| 100 | 1000 | 0.5 | 176 | 738 | 2.9 | 320 | 1342 | 5.2 |

## S4.7   Out-of-Distribution Detection

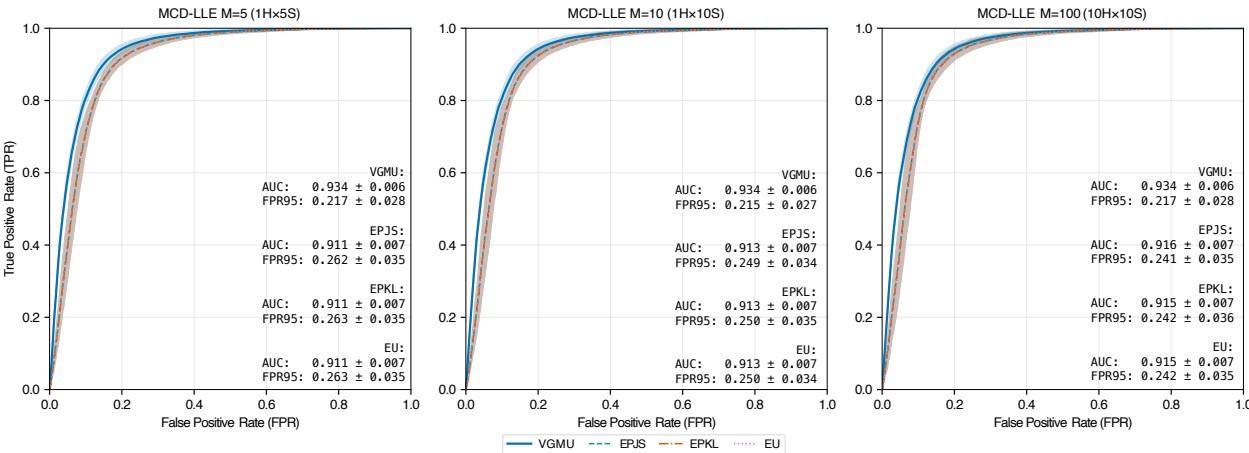

Figure S6: ROC curves for OOD detection (SVHN → CIFAR-10) with MCD-LLE under varying head ($H$) × sample ($S$) configurations with increasing stochastic samples: $1H \times 5S$ ($M = 5$), $1H \times 10S$ ($M = 10$), and $10H \times 10S$ ($M = 100$). VGMU curves are nearly identical across all configurations.

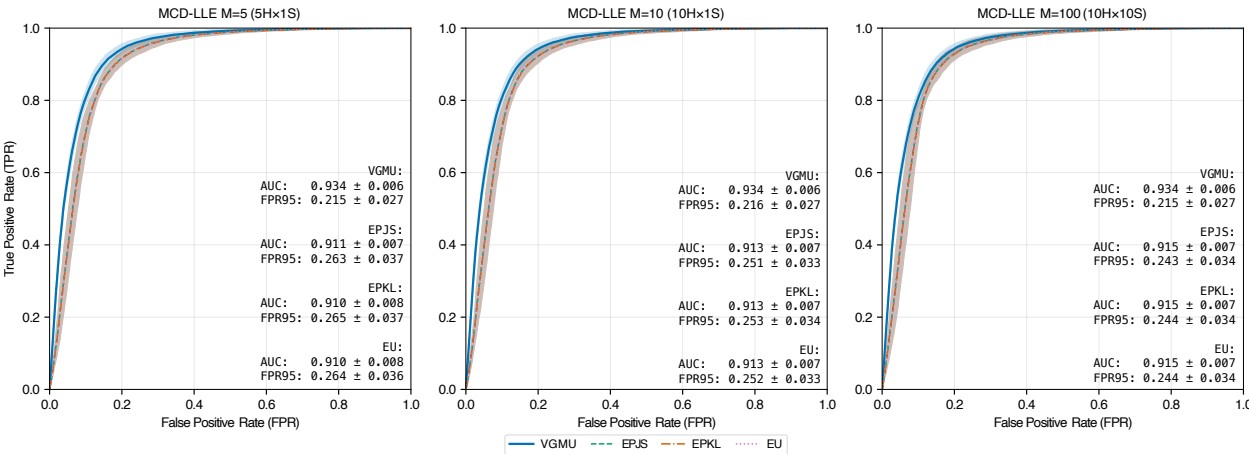

Figure S7: ROC curves for OOD detection (SVHN → CIFAR-10) with MCD-LLE under varying head ($H$) × sample ($S$) configurations with increasing classifier heads: $5H \times 1S$ ($M = 5$), $10H \times 1S$ ($M = 10$), and $10H \times 10S$ ($M = 100$). The rightmost panel ($M = 100$) is shared with Figure S6.

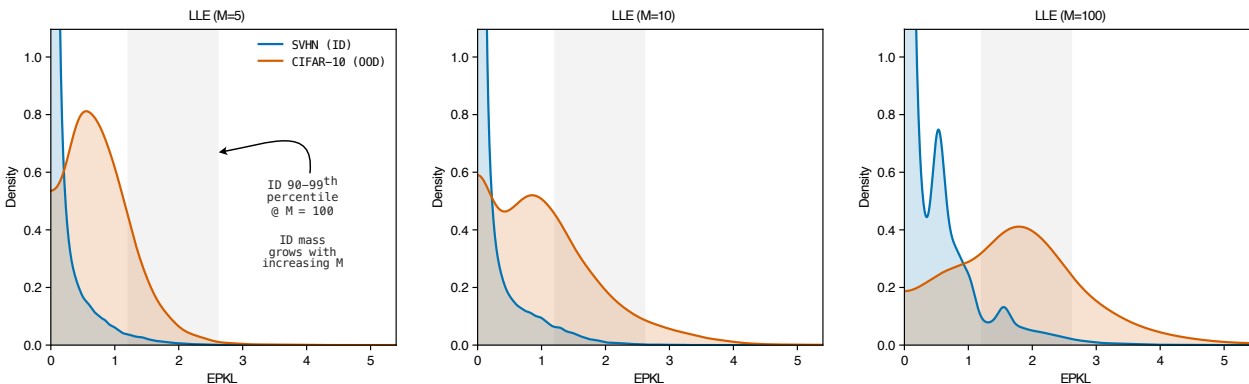

Figure S8: Per-sample EPKL density estimates on SVHN (ID) and CIFAR-10 (OOD) for LLE with $M \in \{5, 10, 100\}$. The shaded band identifies the ID 90–99$^{\text{th}}$ percentile range at $M = 100$, fixed across panels. At $M = 5$, the OOD distribution is unimodal at low EPKL with the ID concentrated near zero, resulting in sufficient separation. As $M$ increases, the ID distribution accumulates mass. The ID fraction grows from 1.7% ($M = 5$) to 3.0% ($M = 10$) to 9.0% ($M = 100$), while the OOD fraction grows from 17.9% to 32.5% to 52.2%. At $M = 100$, the ID distribution becomes visibly multimodal. The compression of the OOD-to-ID mass ratio from $10.9\times \to 5.8\times$, translates into the flat segment of the ROC at FPR $\in [0.15, 0.30]$ visible in the EPKL curve of Figure 8 (LLE, $M = 100$ panel).

### S4.8 Sensitivity of VGN to Hyperparameter k

In the main text, the per-class parameter $k$ in VGN is learned end-to-end during training. A natural question is whether this learnable parameterization is necessary, or whether a fixed $k$ suffices. To investigate, we train LLE-VGN on CIFAR-10 (WideResNet-28-10, $M$=10) with $k$ fixed at three values spanning an order of magnitude: $k \in \{0.127, 0.693, 4.018\}$, corresponding to softplus($\ell$) $\in \{-2.0, 0.0, 4.0\}$. All other hyperparameters, including the training protocol and early stopping criteria, are held constant. Results are averaged over three random seeds.

**Classification performance.** Table S8 reports accuracy, F1-score, ECE, and diversity. Classification accuracy is largely insensitive to the choice of $k$, with all configurations achieving 0.842–0.849 accuracy. The learnable-$k$ variant (0.849) and fixed $k = 0.127$ (0.849) perform comparably. ECE varies modestly across settings; the non-VGN baseline achieves the lowest ECE (0.058), while fixed $k = 4.018$ resulted with the highest (0.076).

Table S8: Classification performance for LLE and LLE-VGN on CIFAR-10 (WideResNet-28-10, $M$=10). LLE reports ensemble predictions ($P$); VGN variants report variance-gated predictions ($Q$).

| Method | $k$ | Accuracy | F1 | ECE | Diversity |
|---|---|---|---|---|---|
| LLE | – | 0.848 ± 0.003 | 0.848 ± 0.002 | 0.058 ± 0.010 | $3.6 \times 10^{-3}$ ±0.6 |
| LLE-VGN | 0.127 | 0.849 ± 0.006 | 0.850 ± 0.005 | 0.065 ± 0.002 | $3.3 \times 10^{-3}$ ±0.4 |
| LLE-VGN | 0.693 | 0.845 ± 0.005 | 0.845 ± 0.005 | 0.061 ± 0.004 | $3.6 \times 10^{-3}$ ±0.4 |
| LLE-VGN | 4.018 | 0.842 ± 0.004 | 0.843 ± 0.003 | 0.076 ± 0.009 | $3.5 \times 10^{-3}$ ±0.6 |
| LLE-VGN | learned | 0.849 ± 0.004 | 0.849 ± 0.003 | 0.066 ± 0.003 | $3.2 \times 10^{-3}$ ±0.2 |

**Uncertainty quality.** Table S9 reports AUCc scores, which measure the concentration of uncertainty mass on difficult samples. Here, the effect of $k$ is more pronounced. Fixed $k = 0.693$ matches the non-VGN baselines, indicating that this value of $k$ effectively removes any benefits of the variance gate. In contrast, fixed $k = 0.127$ and $k = 4.018$ both improve over the baseline, with $k = 4.018$ achieving the highest AUCc across all uncertainty metrics. The learnable-$k$ variants falls between the two best fixed values, indicating that it recovers near-optimal performance without requiring a hyperparameter search.

Table S9: Uncertainty mass concentration (AUCc) for LLE and LLE-VGN on CIFAR-10 (WideResNet-28-10, $M$=10). Higher values indicate sharper concentration on difficult samples.

| Method | $k$ | VGMU | EU | EPJS | EPKL |
|---|---|---|---|---|---|
| LLE | – | 0.885 ± 0.015 | 0.868 ± 0.016 | 0.873 ± 0.016 | 0.876 ± 0.015 |
| LLE-VGN | 0.127 | 0.894 ± 0.008 | 0.879 ± 0.009 | 0.884 ± 0.008 | 0.885 ± 0.009 |
| LLE-VGN | 0.693 | 0.885 ± 0.006 | 0.868 ± 0.007 | 0.873 ± 0.006 | 0.876 ± 0.006 |
| LLE-VGN | 4.018 | 0.899 ± 0.012 | 0.886 ± 0.012 | 0.891 ± 0.012 | 0.892 ± 0.010 |
| LLE-VGN | learned | 0.897 ± 0.009 | 0.881 ± 0.010 | 0.886 ± 0.010 | 0.886 ± 0.010 |

**Rank correlations.** Table S10 and Table S11 report Spearman and Kendall rank correlations between VGMU and epistemic uncertainty baselines. The pattern mirrors the AUCc results: fixed $k = 0.693$ matches the baseline, while $k = 4.018$ and learnable $k$ achieve the highest correlations. All VGN variants maintain strong rank agreement ($\rho > 0.986$, $\tau > 0.907$).

**Summary.** The sensitivity analysis reveals that VGN's benefit depends on the choice of $k$, with performance varying non-monotonically across the tested range. An unfavorable choice (*e.g.*, $k = 0.693$) can

Table S10: Spearman rank correlation ($\rho$) between VGMU and epistemic uncertainty baselines on CIFAR-10 (WideResNet-28-10, $M = 10$).

| Method | $k$ | VGMU *vs.* EU | VGMU *vs.* EPJS | VGMU *vs.* EPKL |
|---|---|---|---|---|
| LLE | – | $0.988 \pm 0.004$ | $0.992 \pm 0.002$ | $0.985 \pm 0.004$ |
| LLE-VGN | 0.127 | $0.991 \pm 0.002$ | $0.994 \pm 0.001$ | $0.988 \pm 0.003$ |
| LLE-VGN | 0.693 | $0.989 \pm 0.001$ | $0.993 \pm 0.001$ | $0.986 \pm 0.002$ |
| LLE-VGN | 4.018 | $0.993 \pm 0.002$ | $0.995 \pm 0.001$ | $0.990 \pm 0.002$ |
| LLE-VGN | learned | $0.992 \pm 0.002$ | $0.995 \pm 0.001$ | $0.990 \pm 0.002$ |

Table S11: Kendall rank correlation ($\tau$) between VGMU and epistemic uncertainty baselines on CIFAR-10 (WideResNet-28-10, $M = 10$).

| Method | $k$ | VGMU vs. EU | VGMU vs. EPJS | VGMU vs. EPKL |
|---|---|---|---|---|
| LLE | – | $0.918 \pm 0.011$ | $0.939 \pm 0.009$ | $0.905 \pm 0.012$ |
| LLE-VGN | 0.127 | $0.928 \pm 0.007$ | $0.946 \pm 0.005$ | $0.917 \pm 0.009$ |
| LLE-VGN | 0.693 | $0.920 \pm 0.003$ | $0.940 \pm 0.003$ | $0.907 \pm 0.003$ |
| LLE-VGN | 4.018 | $0.934 \pm 0.007$ | $0.951 \pm 0.006$ | $0.924 \pm 0.010$ |
| LLE-VGN | learned | $0.932 \pm 0.008$ | $0.948 \pm 0.007$ | $0.923 \pm 0.010$ |

remove the effect of variance gating, while a well-chosen fixed $k$ (*e.g.*, 4.018) can slightly outperform the learnable variant. However, the learnable parameterization consistently achieves near-optimal uncertainty calibration without requiring prior knowledge of the appropriate $k$ value, making it the recommended default for practical use.

## S5    Risk-Based Interpretation

The term **ks** represents a classwise tolerance scale describing the dispersion of ensemble member probabilities around the ensemble mean. Under mild distributional assumptions on ensemble dispersion, $\mathbf{k} \in \{1, 2, 3\}$ corresponds to *ca.* 68%, 95%, or 99.7% of members within one, two, or three standard deviations of the mean, respectively. When ensemble predictions are consistent ($\mathbf{ks} \ll \bar{\mathbf{p}}$), the gate satisfies $\boldsymbol{\Gamma} \approx 1$, leaving probabilities unchanged ($\mathbf{q}_m \approx \mathbf{p}_m$). When disagreement is significant ($\mathbf{ks} \gg \bar{\mathbf{p}}$), $\boldsymbol{\Gamma}$ decreases; however, after normalization, probability mass is redistributed such that high-variance (uncertain) classes are suppressed. The quadratic decay of sensitivity with respect to $\mathbf{s}$ and $\mathbf{k}$ ensures that further increases in disagreement produce diminishing changes to the gate. This variance-gating effect is not uniform suppression. The gating selectively down-weights inconsistent ensemble predictions while preserving those supported by ensemble agreement. Normalization amplifies the relative impact of smaller gates, ensuring that ensemble disagreement yields conservative, bounded, and well-calibrated predictive distributions. This behavior ensures that epistemic disagreement translates into conservative distributions, thereby enforcing the consistency and boundedness properties required by Wimmer's Axioms A0–A5.

## S6    Variance-Gated Behavior Across Axioms

Table S12 and Figure S9 provide illustrations of variance-gated ensemble behavior under the axiomatic framework introduced by Wimmer et al. (2023). Table S12 defines ensemble configurations designed to test each axiom (A2–A5), while Figure S9 visualizes the corresponding simplex geometry and uncertainty decompositions as the gating parameter $k$ varies.

**A2 (Maximal at Uniform Distribution).**  When ensemble members span the simplex vertices ($P_0$), the variance-gated decomposition correctly assigns maximal epistemic uncertainty (EU = 1.0), as members are maximally disagreeing despite the uniform ensemble mean. When members collapse to identical uniform distributions ($Q_0$), EU vanishes (EU = 0.0) while total and aleatoric uncertainty remain maximal (TU = AU = 1.0). This configuration is invariant to gating since $Q_0$ has zero variance. The results confirm that VGN distinguishes between distributional ambiguity (high AU) and ensemble disagreement (high EU), consistent with the partial satisfaction noted in Table 8.

**A3 (Mean-Preserving Spread).**  The transformation from $P_0$ (identical members with $\boldsymbol{\mu} = [0.70, 0.20, 0.10]$, $\boldsymbol{\sigma} = \mathbf{0}$) to $Q_0$ (spread members with preserved mean, $\boldsymbol{\sigma} = [0.17, 0.10, 0.10]$) demonstrates that VGN correctly increases EU when ensemble disagreement grows without changing the predictive mean. At $k = 0$, EU increases from 0.000 to 0.070. Figure S9 shows that increasing $k$ progressively attenuates this epistemic signal, with EU decreasing from 0.070 ($k = 0$) to 0.055 ($k = 1$) to 0.045 ($k = 2$), illustrating the saturation behavior described in Proposition 3.2. Correspondingly, the VGMU increases from 0.300 to 0.412 ($k = 0$), reflecting decision-relevant uncertainty under ensemble disagreement.

**A4 (Center-Shift).**  Shifting ensemble mass toward the simplex barycenter (from $\boldsymbol{\mu} = [0.70, 0.20, 0.10]$ in $P_0$ to $\boldsymbol{\mu} = [0.40, 0.35, 0.25]$ in $Q_0$) while preserving spread increases both TU ($0.730 \rightarrow 0.984$) and AU ($0.714 \rightarrow 0.971$). EU remains approximately stable across this transformation ($0.016 \rightarrow 0.013$ at $k = 0$), demonstrating that VGN correctly attributes the increased uncertainty to aleatoric rather than epistemic sources. The VGMU increases substantially ($0.325 \rightarrow 0.887$), reflecting the reduced class separability near the barycenter. This behavior persists across $k$ values, confirming that variance-gating preserves the distinction between location-induced ambiguity and spread-induced disagreement.

**A5 (Spread-Preserving Location Shift).**  Moving ensembles toward a vertex (from $\boldsymbol{\mu} = [0.70, 0.20, 0.10]$ in $P_0$ to $\boldsymbol{\mu} = [0.90, 0.05, 0.05]$ in $Q_0$) while preserving spread decreases TU ($0.730 \rightarrow 0.359$) and AU ($0.714 \rightarrow 0.314$). At $k = 0$, EU shows sensitivity to the location shift ($0.016 \rightarrow 0.045$), which represents a deviation from strict axiom compliance. However, as $k$ increases, the gap between $P$ and $Q$ epistemic uncertainty values narrows substantially: from 0.029 at $k = 0$, to 0.015 at $k = 1$, to 0.006 at $k = 2$. This progressive reduction suggests that variance-gating enforces approximate invariance to location shifts, with stronger gating (larger $k$) providing closer adherence to axiom A5. The corresponding VGMU values ($0.325 \rightarrow 0.103$ at $k = 0$) reflect the improved class separability near the vertex.

Table S12: Illustrative ensembles used to evaluate Wimmer's axiom (A2–A5). Each pair of ensembles $P_0$ and $Q_0$ displays a transformation corresponding to a specific axiom. These examples are used to test whether a given uncertainty measure respects the qualitative properties required by each axiom.

| Axioms | Ensemble $P_0$ | Ensemble $Q_0$ |
|---|---|---|
| A2: Maximal at Uniform Distribution | $P_0 = \begin{bmatrix} 1.00 & 0.00 & 0.00 \\ 0.00 & 1.00 & 0.00 \\ 0.00 & 0.00 & 1.00 \end{bmatrix}$ | $Q_0 = \begin{bmatrix} 1/3 & 1/3 & 1/3 \\ 1/3 & 1/3 & 1/3 \\ 1/3 & 1/3 & 1/3 \end{bmatrix}$ |
| | $\mu = \begin{bmatrix} 0.33 & 0.33 & 0.33 \end{bmatrix}$ $\sigma = \begin{bmatrix} 0.58 & 0.58 & 0.58 \end{bmatrix}$ | $\mu = \begin{bmatrix} 0.33 & 0.33 & 0.33 \end{bmatrix}$ $\sigma = \begin{bmatrix} 0.00 & 0.00 & 0.00 \end{bmatrix}$ |
| A3: Mean-Preserving Spread | $P_0 = \begin{bmatrix} 0.70 & 0.20 & 0.10 \\ 0.70 & 0.20 & 0.10 \\ 0.70 & 0.20 & 0.10 \end{bmatrix}$ | $Q_0 = \begin{bmatrix} 0.90 & 0.10 & 0.00 \\ 0.60 & 0.30 & 0.10 \\ 0.60 & 0.20 & 0.20 \end{bmatrix}$ |
| | $\mu = \begin{bmatrix} 0.70 & 0.20 & 0.10 \end{bmatrix}$ $\sigma = \begin{bmatrix} 0.00 & 0.00 & 0.00 \end{bmatrix}$ | $\mu = \begin{bmatrix} 0.70 & 0.20 & 0.10 \end{bmatrix}$ $\sigma = \begin{bmatrix} 0.17 & 0.10 & 0.10 \end{bmatrix}$ |
| A4: Center-Shift | $P_0 = \begin{bmatrix} 0.80 & 0.15 & 0.05 \\ 0.70 & 0.20 & 0.10 \\ 0.60 & 0.25 & 0.15 \end{bmatrix}$ | $Q_0 = \begin{bmatrix} 0.50 & 0.30 & 0.20 \\ 0.40 & 0.35 & 0.25 \\ 0.30 & 0.40 & 0.30 \end{bmatrix}$ |
| | $\mu = \begin{bmatrix} 0.70 & 0.20 & 0.10 \end{bmatrix}$ $\sigma = \begin{bmatrix} 0.10 & 0.05 & 0.05 \end{bmatrix}$ | $\mu = \begin{bmatrix} 0.40 & 0.35 & 0.25 \end{bmatrix}$ $\sigma = \begin{bmatrix} 0.10 & 0.05 & 0.05 \end{bmatrix}$ |
| A5: Spread-Preserving Location Shift | $P_0 = \begin{bmatrix} 0.80 & 0.15 & 0.05 \\ 0.70 & 0.20 & 0.10 \\ 0.60 & 0.25 & 0.15 \end{bmatrix}$ | $Q_0 = \begin{bmatrix} 1.00 & 0.00 & 0.00 \\ 0.90 & 0.05 & 0.05 \\ 0.80 & 0.10 & 0.10 \end{bmatrix}$ |
| | $\mu = \begin{bmatrix} 0.70 & 0.20 & 0.10 \end{bmatrix}$ $\sigma = \begin{bmatrix} 0.10 & 0.05 & 0.05 \end{bmatrix}$ | $\mu = \begin{bmatrix} 0.90 & 0.05 & 0.05 \end{bmatrix}$ $\sigma = \begin{bmatrix} 0.10 & 0.05 & 0.05 \end{bmatrix}$ |

These results provide examples of the axiomatic compliance summarized in Table 8 of the main text. The simplex visualizations in Figure S9 complement the geometric interpretation in Section 3.1.1, illustrating how the four ensemble simple regions (*i.e.*, confident–certain, ambiguous–certain, confident–uncertain, ambiguous–uncertain) manifest under controlled axiom-testing transformations. The progressive reduction in uncertainty measures with increasing $k$ demonstrates the risk-tolerance interpretation discussed in Section S5, where larger $k$ values accommodate greater ensemble disagreement before suppressing predictions. The partial deviations observed, particularly for A5 at low $k$, reflect the finite-ensemble and exponential gating effects inherent to the VGN formulation, while the convergence with increasing $k$ supports the properties claimed in the main text.

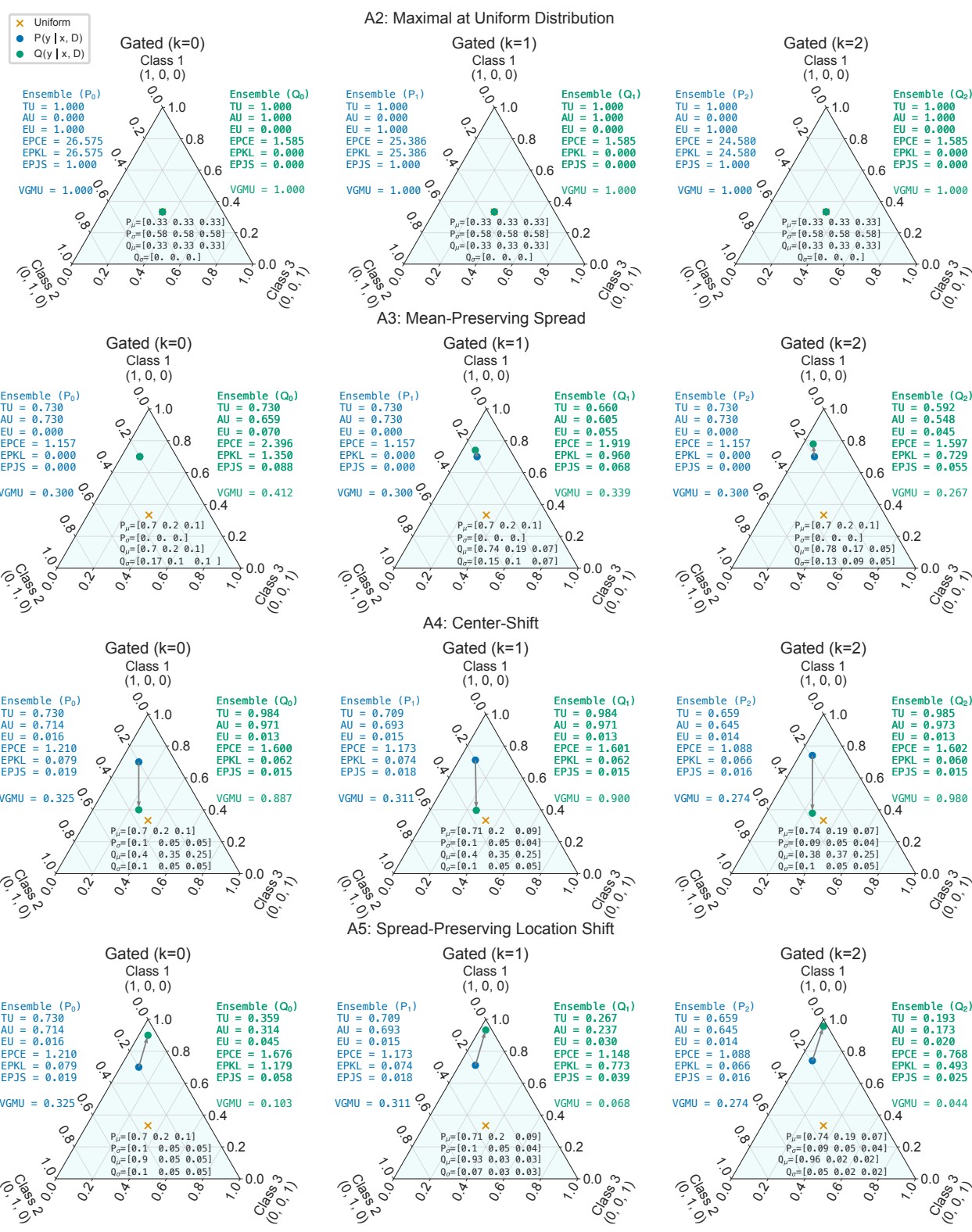

Figure S9: Variance-gated behavior across Wimmer's axioms (A2–A5). Annotations report TU, AU, and EU uncertainty, as well as divergence-based metrics (EPCE, EPKL, EPJS). Gating progressively reduces epistemic contributions, demonstrating compliance with the axioms under controlled variance attenuation. The orange cross denotes the simplex uniform distribution, while circles indicate distributions before (P, blue) and after (Q, green) gating.

## S7 Axiomatic Justification of Variance-Gated Normalization

This section provides formal proofs and justifications of the axiomatic properties claimed in Table 8 for the VGN framework. We adopt the axiomatic framework introduced by Wimmer et al. (2023) and evaluate each axiom (A0–A5) with respect to the VGN uncertainty decomposition.

Recall the relevant definitions from Section 3.3. Given an ensemble of $M$ members producing categorical distributions $\mathbf{p}_m \in \Delta^{C-1}$, the per-class ensemble sample mean and standard deviation are $\bar{\mathbf{p}}$ and $\mathbf{s}$, respectively, where $\mathbf{s}$ is stabilized by a small constant $\varepsilon > 0$ to ensure $s_c > 0$ for all $c$.

The variance gate is $\mathbf{\Gamma} = 1 - e^{-\bar{\mathbf{p}}/\mathbf{ks}}$, the gated member distribution is $\mathbf{q}_m = (\mathbf{p}_m \odot \mathbf{\Gamma})/Z_m$ with $Z_m = \mathbf{p}_m^\top \mathbf{\Gamma}$, and the gated mixture is $\bar{\mathbf{q}} = \frac{1}{M} \sum_{m=1}^M \mathbf{q}_m$. The VGN uncertainty decomposition is

$$\mathrm{TU} := H(\bar{\mathbf{q}}) = -\bar{\mathbf{q}}^\top \log \bar{\mathbf{q}}$$

$$\mathrm{AU} := \frac{1}{M} \sum_{m=1}^M H(\mathbf{q}_m)$$

$$\mathrm{EU} := \mathrm{TU} - \mathrm{AU} = \frac{1}{M} \sum_{m=1}^M D_{\mathrm{KL}}(\mathbf{q}_m \| \bar{\mathbf{q}}).$$

### S7.1 A0: Non-Negativity

**Proposition S7.1** (Non-negativity). *For any ensemble $\{\mathbf{p}_m\}_{m=1}^M$ with $\mathbf{p}_m \in \Delta^{C-1}$ and any $\mathbf{k} > 0$, the VGN decomposition satisfies* $\mathrm{TU} \geq 0$, $\mathrm{AU} \geq 0$, *and* $\mathrm{EU} \geq 0$.

*Proof.* We proceed by verifying that each gated distribution $\mathbf{q}_m$ lies in the probability simplex $\Delta^{C-1}$, from which non-negativity of TU, AU, and EU follows from standard information-theoretic inequalities.

**Step 1: $\mathbf{q}_m$ is a valid probability distribution.** Since $\mathbf{p}_m \in \Delta^{C-1}$, we have $p_m(c) \geq 0$ for all $c$ and $\sum_c p_m(c) = 1$. The gate satisfies $0 \leq \Gamma_c < 1$ for all $c$ (since the exponential is strictly positive). Therefore $p_m(c)\,\Gamma_c \geq 0$ for all $c$, which gives $q_m(c) = p_m(c)\,\Gamma_c/Z_m \geq 0$. The normalization constant $Z_m = \sum_c p_m(c)\,\Gamma_c > 0$ whenever at least one class $c$ satisfies $p_m(c) > 0$ and $\Gamma_c > 0$, which holds for any ensemble with $\bar{\mathbf{p}} \neq \mathbf{0}$ (ensured by $\varepsilon > 0$ in $\mathbf{s}$). By construction, $\sum_c q_m(c) = Z_m/Z_m = 1$. Hence $\mathbf{q}_m \in \Delta^{C-1}$.

**Step 2: $\bar{\mathbf{q}}$ is a valid probability distribution.** As a convex combination of distributions in $\Delta^{C-1}$, the mixture $\bar{\mathbf{q}} = \frac{1}{M} \sum_m \mathbf{q}_m$ satisfies $\bar{q}(c) \geq 0$ for all $c$ and $\sum_c \bar{q}(c) = 1$.

**Step 3: Non-negativity of TU, AU, EU.** Since $\bar{\mathbf{q}} \in \Delta^{C-1}$, Shannon entropy gives $\mathrm{TU} = H(\bar{\mathbf{q}}) \geq 0$, with equality if and only if $\bar{\mathbf{q}}$ is a point mass. Similarly, $\mathrm{AU} = \frac{1}{M} \sum_m H(\mathbf{q}_m) \geq 0$ as an average of non-negative entropies. Finally, since $-\log$ is convex, Jensen's inequality gives for each $m$:

$$D_{\mathrm{KL}}(\mathbf{q}_m \| \bar{\mathbf{q}}) = -\sum_c q_m(c) \log \frac{\bar{q}(c)}{q_m(c)} \geq -\log\left(\sum_c q_m(c) \frac{\bar{q}(c)}{q_m(c)}\right) = -\log(1) = 0, \qquad \text{(S23)}$$

with equality if and only if $\mathbf{q}_m = \bar{\mathbf{q}}$. Hence $\mathrm{EU} = \frac{1}{M} \sum_m D_{\mathrm{KL}}(\mathbf{q}_m \| \bar{\mathbf{q}}) \geq 0$. $\qquad\square$

### S7.2 A1: Vanishing Epistemic Uncertainty for Identical Members

**Proposition S7.2** (Vanishing EU under consensus). *If all ensemble members produce identical predictions,* $\mathbf{p}_m = \mathbf{p}$ *for all* $m \in \{1, \ldots, M\}$, *then* $\mathrm{EU} = 0$.

*Proof.* When $\mathbf{p}_m = \mathbf{p}$ for all $m$, the ensemble statistics are $\bar{\mathbf{p}} = \mathbf{p}$ and $\mathbf{s} = \boldsymbol{\varepsilon}$ (the stabilization vector). The gate becomes $\Gamma_c = 1 - e^{-p_c/k_c\varepsilon}$ for each class $c$. Since the gate depends only on shared statistics $(\bar{\mathbf{p}}, \mathbf{s})$, the vector $\mathbf{\Gamma}$ is identical for all members. Because all members share the same $\mathbf{p}$ and $\mathbf{\Gamma}$

$$\mathbf{q}_m = \frac{\mathbf{p} \odot \mathbf{\Gamma}}{\mathbf{p}^\top \mathbf{\Gamma}} = \mathbf{q}, \qquad \text{for all } m. \qquad \text{(S24)}$$

Consequently, $\bar{\mathbf{q}} = \frac{1}{M}\sum_m \mathbf{q} = \mathbf{q}$, and

$$\text{EU} = \frac{1}{M}\sum_{m=1}^{M} D_{\text{KL}}(\mathbf{q} \,\|\, \mathbf{q}) = 0. \tag{S25}$$

$\square$

**Remark S7.2.1.** *This result is independent of* $\mathbf{k}$, $\varepsilon$, *the specific form of* $\mathbf{p}$, *and the number of ensemble members* $M$. *The gate* $\mathbf{\Gamma}$ *is a shared function of ensemble statistics; when members agree, it applies an identical transformation to all members, preserving consensus.*

### S7.3 A2: Maximal EU and TU Under Maximally Disagreeing Uniform-Mean Ensembles

**Proposition S7.3** (Partial satisfaction of A2). *Let the ensemble consist of* $M = C$ *members, each placing all mass on a distinct class (i.e.,* $\mathbf{p}_m = \mathbf{e}_m$, *the* $m$-*th standard basis vector). Then the VGN decomposition gives* $\text{TU} = \log C$ *and* $\text{EU} = \log C$ *(i.e., both are maximal). However, the gate introduces a dependence on* $\mathbf{k}$ *for intermediate configurations, which prevents a global maximality guarantee under all uniform-mean ensembles.*

*Proof.*
**Case 1: Vertex-spanning ensemble ($\mathbf{p}_m = \mathbf{e}_m$).** The ensemble mean is $\bar{p}_c = 1/C$ for all $c$. The sample standard deviation satisfies $s_c > 0$ for all $c$ (since members alternate between 0 and 1 for each class). The gate is $\Gamma_c = 1 - e^{-\bar{p}_c/k_c s_c} > 0$ for all $c$. By symmetry across classes, $\Gamma_c = \Gamma$ for all $c$.

For member $m$ with $\mathbf{p}_m = \mathbf{e}_m$, only the $m$-th component is nonzero, so $q_m(c) = p_m(c)\,\Gamma_c/Z_m$. Since $p_m(c) = \delta_{mc}$ and $\Gamma_c = \Gamma$ for all $c$

$$q_m(c) = \frac{\delta_{mc}\Gamma}{\Gamma} = \delta_{mc}. \tag{S26}$$

Thus $\mathbf{q}_m = \mathbf{e}_m$ (the gated distributions remain one-hot). The mixture is $\bar{q}(c) = 1/C$ for all $c$. Therefore

$$\text{TU} = H(\bar{\mathbf{q}}) = \log C \qquad \text{AU} = \frac{1}{C}\sum_m H(\mathbf{e}_m) = 0 \qquad \text{EU} = \log C. \tag{S27}$$

These are the maximum possible values for entropy of a $C$-class distribution.

**Case 2: Identical uniform members ($\mathbf{p}_m = \mathbf{u} = [1/C, \dots, 1/C]^\top$).** By Proposition S7.2 (A1), $\text{EU} = 0$. Meanwhile $\text{TU} = H(\bar{\mathbf{q}})$. Since all members are identical and uniform, $\mathbf{s} = \boldsymbol{\varepsilon}$ and the gate saturates ($\Gamma_c \to 1$ as $\varepsilon \to 0$), resulting in $\mathbf{q}_m \approx \mathbf{u}$ and $\text{TU} \approx \log C$. Thus TU is maximal but EU is zero.

A2 requires that EU and TU are maximal when the ensemble mean is uniform. Case 1 satisfies this when members are maximally spread (vertex-spanning). However, for intermediate configurations where the mean is uniform but members are not at the vertices, the gate parameter $\mathbf{k}$ modulates the degree of suppression. In particular, larger $k$ values reduce the effective gate, attenuating disagreement signals and potentially decreasing EU below its non-gated maximum. This $\mathbf{k}$-dependence prevents a maximality guarantee across all uniform-mean ensemble configurations, resulting in partial satisfaction. $\square$

### S7.4 A3: Monotonicity Under Mean-Preserving Spread

**Proposition S7.4** (EU increases with mean-preserving spread). *Consider an ensemble* $P = \{\mathbf{p}_m\}$ *with non-identical members, mean* $\bar{\mathbf{p}}$, *and deviations* $\mathbf{d}_m = \mathbf{p}_m - \bar{\mathbf{p}}$. *Let* $P' = \{\bar{\mathbf{p}} + \alpha\,\mathbf{d}_m\}$ *for* $\alpha > 1$ *such that all members remain in* $\Delta^{C-1}$. *Then* $\text{EU}(P') > \text{EU}(P)$.

*Proof.*
**Case 1: $s_c = 0$ for all $c$.** Let $P$ have identical members, so $\text{EU}(P) = 0$ by A1. Since $P'$ has strictly greater variance on at least one class, $P'$ contains non-identical members. It remains to show that distinct $\mathbf{p}'_m$ yield distinct gated distributions $\mathbf{q}'_m$. For every class with $\bar{p}_c > 0$, the gate satisfies $\Gamma'_c = 1 - e^{-\bar{p}_c/k_c s'_c} > 0$;

classes with $\bar{p}_c = 0$ have $p'_m(c) = 0$ for all $m$ (non-negative terms summing to zero) and do not affect the gated distributions. Since $q'_m(c) = p'_m(c)\,\Gamma'_c/Z'_m$ with $\Gamma'_c > 0$ for all classes with $\bar{p}_c > 0$, the map $\mathbf{p}'_m \mapsto \mathbf{q}'_m$ is an elementwise rescaling by a shared positive vector followed by normalization. Such a map preserves distinctness, where the ratios between entries of distinct probability vectors are unchanged by positive rescaling, so $\mathbf{p}'_m \neq \mathbf{p}'_n \Longrightarrow \mathbf{q}'_m \neq \mathbf{q}'_n$. Therefore

$$\mathrm{EU}(P') = \frac{1}{M} \sum_m D_{\mathrm{KL}}(\mathbf{q}'_m \,\|\, \bar{\mathbf{q}}') > 0 = \mathrm{EU}(P). \tag{S28}$$

**Case 2:** $s_c > 0$ **in reference ensemble** $P$. Since $q_m(c) = p_m(c)\,\Gamma_c/Z_m$ and $\bar{q}(c) = \frac{1}{M}\sum_m q_m(c)$, the KL divergence in EU depends on the log-ratios $\log(q_m(c)/\bar{q}(c))$. A key structural property is that $\Gamma_c$ cancels from every log-ratio, since $\bar{q}(c) = \Gamma_c \cdot \frac{1}{M}\sum_n p_n(c)/Z_n$, so

$$\log \frac{q_m(c)}{\bar{q}(c)} = \log \frac{p_m(c)/Z_m}{\frac{1}{M}\sum_n p_n(c)/Z_n}\,. \tag{S29}$$

Scaling $\mathbf{d}_m$ by $\alpha > 1$ increases the spread of $\{p'_m(c)\}_m$ around the fixed mean $\bar{p}_c$, which directly amplifies these log-ratios. Meanwhile, the reduced gate $\mathbf{\Gamma}'$ suppresses high-variance classes, partially counteracting this effect. However, since $\Gamma_c$ cancels from the log-ratios, the spread of member values directly determines the KL divergences, while the gate suppression affects EU only indirectly through $q_m(c)$ and normalization constants $Z_m$. As a result, $\mathrm{EU}(P') > \mathrm{EU}(P)$. $\square$

**Remark S7.4.1.** *Note that A3 as stated by [Wimmer et al. (2023)](#) also requires TU monotonicity under mean-preserving spread. For VGN, increased spread reduces gate values, which can decrease AU by more than EU increases, resulting in lower TU (e.g., [Figure S9](#) shows TU decreasing from* $0.730$ *to* $0.592$ *at* $k=2$*). VGN therefore satisfies the EU component of A3 but not the TU component.*

### S7.5 A4: Monotonicity Under Center-Shift (uniform noise addition)

**Proposition S7.5** (AU and TU increase under center-shift). *Consider a transformation that shifts the ensemble mean* $\bar{\mathbf{p}}$ *toward the barycenter* $\mathbf{u} = [1/C, \ldots, 1/C]^\top$ *while preserving the per-class standard deviation* $\mathbf{s}$. *Then* AU *and* TU *both increase.*

*Proof.* Let $P = \{\mathbf{p}_m\}$ be the original ensemble and $P' = \{\mathbf{p}'_m\}$ the shifted ensemble, with $\bar{\mathbf{p}}' = (1-\alpha)\bar{\mathbf{p}} + \alpha\mathbf{u}$ for some $\alpha \in (0,1]$ and $\mathbf{s}' = \mathbf{s}$.

**Effect on the gate.** Since $\mathbf{s}' = \mathbf{s}$ and the mean shifts toward the barycenter: for classes $c$ where $\bar{p}_c > 1/C$ (high-probability classes), $\bar{p}'_c < \bar{p}_c$, so $\Gamma'_c < \Gamma_c$; for classes $c$ where $\bar{p}_c < 1/C$ (low-probability classes), $\bar{p}'_c > \bar{p}_c$, so $\Gamma'_c > \Gamma_c$. The gate values become more uniform across classes, suppressing all classes more equally.

**AU increases.** When the gate is class-independent (*i.e.*, $\Gamma_c = \Gamma$ for all $c$), it cancels in the normalization and $\mathbf{q}_m = \mathbf{p}_m$ exactly, since $q_m(c) = p_m(c)\,\Gamma/(\sum_j p_m(j)\,\Gamma) = p_m(c)$. In the limit $\bar{\mathbf{p}}' \to \mathbf{u}$, the gate satisfies $\Gamma'_c \to 1 - e^{-1/Ck_c s_c}$ uniformly across classes and $\mathbf{q}'_m \to \mathbf{p}'_m$. Since each $\mathbf{p}'_m$ is closer to the barycenter than $\mathbf{p}_m$ and has higher entropy, the more uniform gate preserves this higher entropy in $\mathbf{q}'_m$, giving $\mathrm{AU}(P') = \frac{1}{M}\sum_m H(\mathbf{q}'_m) \geq \mathrm{AU}(P)$.

**TU increases.** When the gate is class-independent, $\bar{\mathbf{q}} = \bar{\mathbf{p}}$ exactly, the uniform gate cancels during normalization and the mixture mean is unaffected. As center-shift makes the gate more uniform, $\bar{\mathbf{q}}'$ approximates $\bar{\mathbf{p}}'$ more closely than $\bar{\mathbf{q}}$ approximates $\bar{\mathbf{p}}$. Since $\bar{\mathbf{p}}' = (1-\alpha)\bar{\mathbf{p}} + \alpha\mathbf{u}$ is closer to the barycenter than $\bar{\mathbf{p}}$, the concavity of Shannon entropy gives

$$H(\bar{\mathbf{p}}') \geq (1-\alpha)\,H(\bar{\mathbf{p}}) + \alpha \log C \geq H(\bar{\mathbf{p}}), \tag{S30}$$

with strict inequality when $\bar{\mathbf{p}} \neq \mathbf{u}$. Since the gate becomes more uniform under center-shift, $\bar{\mathbf{q}}'$ closely approximates $\bar{\mathbf{p}}'$, giving $\mathrm{TU}(P') = H(\bar{\mathbf{q}}') \geq H(\bar{\mathbf{q}}) = \mathrm{TU}(P)$. $\square$

**Remark S7.5.1.** *This result distinguishes VGN from standard entropy decomposition, which violates A4. The key mechanism is that the variance gate depends on $\bar{\mathbf{p}}$, so it responds to center-shifts by suppressing all classes more equally. This allows the increased distributional ambiguity to manifest in both AU and TU. In standard entropy decomposition, the mixture $\bar{\mathbf{p}}$ is more uniform than the individual members, so $H(\bar{\mathbf{p}})$ may not increase under center-shift even as the uncertainty of individual members increases, causing TU to underestimate the increased aleatoric ambiguity.*

### S7.6 A5: Invariance of EU Under Spread-Preserving Location Shifts

**Proposition S7.6** (Approximate EU invariance under spread-preserving location shifts)**.** *Consider a transformation that shifts the ensemble mean $\bar{\mathbf{p}}$ to $\bar{\mathbf{p}}'$ while preserving the per-class standard deviation $\mathbf{s}' = \mathbf{s}$. The VGN epistemic uncertainty satisfies $\mathrm{EU}(P') = \mathrm{EU}(P)$ when the relative disagreement structure is preserved, and achieves controlled approximate invariance in the general case via the learnable parameter $\mathbf{k}$.*

*Proof.* **Exact invariance under proportional shifts.** Consider the special case where the location shift preserves the per-class signal-to-noise ratio, *i.e.*, $\bar{p}'_c/s'_c = \bar{p}_c/s_c$ for all $c$. This occurs, for example, when both $\bar{\mathbf{p}}$ and $\mathbf{s}$ are scaled by the same factor, but since $\mathbf{s}' = \mathbf{s}$, this case is trivial and corresponds to no shift. In the general case, a location shift with preserved $\mathbf{s}$ changes $\bar{p}_c/(k_c s_c)$ and therefore $\Gamma_c$, so exact invariance does not hold.

**Approximate invariance mechanism.** Despite the absence of exact invariance, VGN achieves controlled approximate invariance through two mechanisms:

(i) *Saturation of the gate.* For classes with $\bar{p}_c/(k_c s_c) \gg 1$, the gate satisfies $\Gamma_c \approx 1$ regardless of the specific value of $\bar{p}_c$. In this region, the gated distributions $\mathbf{q}_m \approx \mathbf{p}_m/(\mathbf{p}_m^\top \mathbf{1}) = \mathbf{p}_m$, and the decomposition reduces to the ungated case. Since the ungated entropy decomposition satisfies A5 exactly (Table 8), VGN inherits this invariance in the saturated region.

(ii) *Controlled sensitivity via $\mathbf{k}$.* The derivative of $\Gamma_c$ with respect to $\bar{p}_c$ is $\partial\Gamma_c/\partial\bar{p}_c = (1 - \Gamma_c)/(k_c s_c)$. As $k_c$ increases, this sensitivity decreases, making the gate less responsive to location shifts. In the limit $k_c \to \infty$, $\Gamma_c \to 0$ uniformly for all classes, and $\mathbf{q}_m \to \mathbf{p}_m$ after renormalization with a gate that is approximately proportional across classes.

For any $\eta > 0$, there exists a $k^*$ such that for all $k_c \geq k^*$

$$|\mathrm{EU}(P') - \mathrm{EU}(P)| < \eta. \tag{S31}$$

This follows from the continuity of the KL divergence and the uniform convergence of $\mathbf{\Gamma}$ as $\mathbf{k}$ increases.

**Empirical confirmation.** The illustrative ensembles in Table S12 confirm this behavior. For the A5 test configuration, the EU gap between $P_0$ and $Q_0$ decreases from 0.029 at $k = 0$ to 0.006 at $k = 2$ (Section S6), demonstrating progressive convergence toward exact invariance with increasing $k$. $\square$

**Remark S7.6.1.** *Recall A2, where no choice of $\mathbf{k}$ can guarantee maximality across all uniform-mean configurations, A5 admits a formal convergence guarantee. For any $\eta > 0$, there exists $k^*$ such that $|\mathrm{EU}(P') - \mathrm{EU}(P)| < \eta$ for all $k_c \geq k^*$. In addition, in the saturated gate region, VGN recovers the standard entropy decomposition, which satisfies A5 exactly. The full satisfaction mark (✓) for A5 in Table 8 reflects this distinction. VGN contains an A5-satisfying decomposition as a limiting case, with a $\mathbf{k}$-controllable bound on the deviation from exact invariance. In practice, the learned values of $\mathbf{k}$ (Section S4.4) place the gate in a region where approximate invariance holds, as confirmed by the progressive reduction in EU gap with increasing $k$ shown above.*

### S7.7 Summary

Table S13 summarizes the formal status of each axiom. Axioms A0, A1, A4, and A5 are satisfied (A5 with controlled approximation *via* $\mathbf{k}$). A2 and A3 are partially satisfied. A2 achieves maximality for the vertex-spanning ensemble but is $\mathbf{k}$-dependent for intermediate configurations; A3 satisfies EU monotonicity but TU

can decrease due to gate suppression of high-variance classes. These results are consistent with the claims in Table 8 and the empirical illustrations in Section S6.

Table S13: Summary of axiomatic proofs for variance-gated normalization.

| Axiom | Status | Proof Basis |
|---|---|---|
| A0: Non-negativity | ✓ | Entropy and KL divergence non-negativity |
| A1: Vanishing EU (identical members) | ✓ | Shared gate $\Rightarrow \mathbf{q}_m = \mathbf{q} \Rightarrow D_{\mathrm{KL}} = 0$ |
| A2: Maximal at uniform | ◐ | Exact for vertex-spanning; $\mathbf{k}$-dependent otherwise |
| A3: Mean-preserving spread | ◐ | EU proven; TU limited by gate suppression |
| A4: Center-shift | ✓ | Gate becomes uniform; concavity of $H$ |
| A5: Location-shift invariance | ✓ | Gate saturation and $\mathbf{k}$-controlled convergence |

