# OpenReview forum: "Variance-Gated Ensembles: An Epistemic-Aware Framework for Uncertainty Estimation"
_TMLR — Accepted by TMLR_

### Review · Reviewer_pDHd · 2026-03-30

**Summary Of Contributions:**

This paper proposes a framework called variance-gated ensembles that aims to provide an appropriate decomposition of aleatoric and epistemic uncertainty at low computational cost. The authors analyze the proposed framework via some properties of gradient values (Sec 3.1.1) and the axioms of uncertainty (Sec. 6). Furthermore, the authors provide the empirical evaluations for well-known datasets, such as MNIST and CIFAR.

**Audience:**

No

**Audience Explanation:**

For the reasons above, I believe most of the audience cannot follow the paper's claim.

**Claims And Evidence:**

No

**Claims Explanation:**

Overall, I could not follow the paper's claim, even though the authors describe the proposed framework as ``intuitive'' in the abstract.
In addition, I believe that the paper includes problems with the writing.

I list the points that I could not understand below:
- There is no description of the problem definition and, e.g., the model $p(y \mid x, w\_m)$. Therefore, I could not understand the problem setup mathematically.
- About the definition of Eqs (6--9), I could not follow why we should consider this transformation. Propositions on page 5 only claim the relative relationship (gradient), and Sec 3.1.1 also provides the extreme case only.
- The claim of propositions is uncertain. For example, Proposition 3.1 claims "scales inversely with predictive spread $s$". However, the proof ignores the dependence on $s$ in $\Gamma_c$.
- For Eq (20), what are $i$ and $S$ (does $S$ correspond to $s_c$ in Section 3.1?)? What does it mean by "uncertain"?
- For Eq. (21), why should we shold consider this equation? Although this equation only depends on $\bar{p}$, not on $\bar{q}$, why is this called "variance-gated"?
- For Eq. (22), there is no definition of $\bar{q}$.
- For Sec. 6, there is no rigorous justification of the content of Table 4. The authors must show mathematically whether or not the proposed method satisfies the axioms.


I list the other problems mainly for notations or writing below:
- Several abbreviations are defined only in the abstract. It is also recommended to define in the main paper.
- At lines 32--33, how to define the abbreviations AU and EU seems to be uncommon. It should be "aleatoric uncertainty (AU) and epistemic uncertainty (EU)."
- The abbreviation "OOD" is used for two different terms.
- I believe that the inequality or division for the vector around a line 149 is not common. The definition of these is at least required.
- The notation like $0 \leq X \leq 1 \in \mathbb{R}$ in Eqs (8) and (9) is also not common.
- How to use $\Rightarrow$ in Eq. (23) is uncommon.

**Requested Changes:**

I believe this paper requires substantial revision throughout to clarify its claim.

---

> ### Author Response · Authors · 2026-04-21
> **Response to Reviewer pDHd (Part 1/2)**
>
> We thank the reviewer for the careful reading. Technical concerns (Points 1–7) are addressed here; notation and writing issues (Points 8–13) follow in a separate comment. Margin annotations `NEW/FIX:pDHd::N` mark each change.
>
> ## Part I: Technical and Conceptual Concerns
>
> **Point 1: Problem Definition and Model $p(y \\mid \\mathbf{x}, \\mathbf{w}_m)$**
>
> > _"There is no description of the problem definition... I could not understand the problem setup mathematically."_
>
> The model was introduced in Eq. (1) via BMA but embedded in broader discussion rather than as a standalone definition.
>
> **Revision [pDHd::1]:** A formal **Problem setup** paragraph at the start of Section 3 defines the $C$-class task, the ensemble $\\text{\\{}\\mathbf{w}_m\\text{\\}}$, $m = 1, \\ldots, M$, and the predictive distribution $p(y \\mid \\mathbf{x}, \\mathbf{w}_m) \\in \\Delta^{C-1}$.
>
> **Point 2: Motivation for the Variance-Gated Transformation (Eqs. 6–9)**
>
> > _"I could not follow why we should consider this transformation."_
>
> The exponential gate $\\boldsymbol{\\Gamma} = 1 - e^{-\\bar{\\mathbf{p}}/\\mathbf{ks}}$ satisfies four properties: (i) smoothness for end-to-end training; (ii) monotonic increase with mean confidence, decrease with spread; (iii) bounded output in $[0,1)$; (iv) natural saturation preventing excessive attenuation.
>
> **Revision [pDHd::2]:** A **Design rationale** paragraph before Eq. (8) lists these four criteria and explains the choice of exponential form.
>
> **Point 3: Proposition 3.1 — Dependence on $s$ in $\\Gamma_c$**
>
> > _"Proposition 3.1 claims 'scales inversely with predictive spread,' but the proof ignores the dependence on $s$ in $\\Gamma_c$."_
>
> Technically valid. The derivative is $\\partial\\Gamma_c/\\partial\\bar{p}_c = (1-\\Gamma_c)/(k_c s_c)$, where $(1-\\Gamma_c) = e^{-\\bar{p}_c/k_c s_c}$ itself depends on $s_c$. The inverse-scaling claim is correct in a first-order sense via the prefactor, but the saturation term adds nonlinear $s_c$ dependence.
>
> **Revision [pDHd::3]:** Proposition 3.1 now states that $\\partial\\Gamma/\\partial\\bar{p} > 0$ is modulated by $s$ through both the inverse prefactor term $1/(k_c s_c)$ and the saturation term $(1-\\Gamma_c)$, which itself depends on $s_c$. The proof identifies both sources of dependence.
>
> **Point 4: Eq. (20) — Undefined $i$ and $S$**
>
> > _"What are $i$ and $S$? What does 'uncertain' mean?"_
>
> Originally, $i$ was the top-1 class index; $S_1, S_2$ were the top-1/top-2 standard deviations; the $s \\to S$ case change and "uncertain" were not explained.
>
> **Revision [pDHd::4]:** (a) $S_1, S_2 \\to s_1, s_2$ for notational consistency; (b) $i$ explicitly defined as the top-1 class index; (c) "uncertain" → "abstain" with an explanation.
>
> **Point 5: Eq. (21) — Why Is This Equation Variance-Gated?**
>
> > _"Although Eq. (21) depends only on $\\bar{p}$, not $\\bar{q}$, why is it called 'variance-gated'?"_
>
> VGMU is an inference-time metric for any ensemble prediction $p$ (not only variance-gated $q$), requiring only per-member means and standard deviations.
>
> **Revision [pDHd::5]:** An explanatory sentence after Eq. (21) clarifies the role of $s_1, s_2$ within the SNR-based gate $\\gamma$ and distinguishes VGMU from VGN.
>
> **Point 6: Eq. (22) — Missing Definition of $\\bar{\\mathbf{q}}$**
>
> > _"There is no definition of $\\bar{q}$."_
>
> Correct. $\\bar{\\mathbf{q}} = \\tfrac{1}{M}\\sum_{m=1}^{M}\\mathbf{q}_m$ was used in Eq. (22) but not formally introduced until the gradients section.
>
> **Revision [pDHd::6]:** Definition added immediately before Eq. (22), noting that $\\bar{\\mathbf{q}}$ is the ensemble mixture of variance-gated member distributions.
>
> **Point 7: Axiomatic Justification (Table 4)**
>
> > _"There is no rigorous justification of Table 4."_
>
> The original gave verbal arguments without formal justifications, and did not adequately acknowledge partial satisfaction in one axiom.
>
> **Revision [pDHd::7]:** A new Supporting Information section **Axiomatic Justification of Variance-Gated Normalization** provides proofs or constructive arguments for each axiom A0–A5 (Wimmer et al. 2023):
>
> - **A0** (non-negativity): entropy and KL non-negativity.
> - **A1** (vanishing EU, identical members): $\\mathbf{q}_m = \\bar{\\mathbf{q}}$ when all $\\mathbf{p}_m$ agree.
> - **A2** (partial): max at vertex-spanning ensembles; $\\mathbf{k}$-dependence prevents full satisfaction.
> - **A3** (mean-preserving spread, partial): EU monotonicity via gate response to $s$; TU fails due to gate suppression.
> - **A4** (center-shift): gate response to uniform noise injection.
> - **A5** (location-shift invariance): gate saturation and $\\mathbf{k}$-controlled convergence.
>
> The main text now references these justifications.
>
> _(Notation and writing issues in follow-up comment.)_

---

> ### Author Response · Authors · 2026-04-21
> **Response to Reviewer pDHd (Part 2/2)**
>
> _Continuing from the previous comment — this part addresses the notation and writing issues (Points 8–13)._
>
> ## Part II: Notation and Writing Issues
>
> **Point 8: Abbreviations Defined Only in the Abstract**
>
> > _"Several abbreviations are defined only in the abstract."_
>
> **Revision [pDHd::8]:** All abbreviations (VGE, VGMU, VGN, BMA, MCD, DE, LLE, BNN, TU, AU, EU, EPKL, EPJS, OOD, SNR) are now defined at first occurrence in the main body.
>
> **Point 9: Abbreviation Format for AU and EU (Lines 32–33)**
>
> > _"Should be 'aleatoric uncertainty (AU) and epistemic uncertainty (EU).'"_
>
> **Revision [pDHd::9]:** Corrected throughout to the standard convention.
>
> **Point 10: Dual Usage of "OOD" Abbreviation**
>
> > _"The abbreviation 'OOD' is used for two different terms."_
>
> **Revision [pDHd::10]:** All occurrences standardized to "out-of-distribution (OOD)"; the "out-of-domain" instance has been replaced.
>
> **Point 11: Element-Wise Inequality and Division for Vectors (Line 149)**
>
> > _"The inequality or division for vectors around line 149 is not common."_
>
> **Revision [pDHd::11]:** A **Notation Convention** statement at the start of Section 3 specifies that all vector operations ($\\ge$, $/$, $\\odot$) are element-wise unless noted.
>
> **Point 12: Notation $0 \\le X \\le 1 \\in \\mathbb{R}$ in Eqs. (8) and (9)**
>
> > _"Notation like $0 \\le X \\le 1$ in $\\mathbb{R}$ is not common."_
>
> **Revision [pDHd::12]:** Domain and bounds separated into distinct statements: $\\boldsymbol{\\Gamma} \\in \\mathbb{R}^C$ as domain, $0 \\le \\Gamma_c < 1$ for all $c$ as property. Same correction applied to $\\mathbf{q}_m$.
>
> **Point 13: Usage of $\\Rightarrow$ in Eq. (23)**
>
> > _"How $\\Rightarrow$ is used in Eq. (23) is uncommon."_
>
> **Revision [pDHd::13]:** All instances of $\\Rightarrow$ in Eqs. 22–24 replaced with $:=$.
>
> We are grateful for this feedback, and believe the revised manuscript is stronger as a result.

---

### Review · Reviewer_cddg · 2026-04-02

**Summary Of Contributions:**

The paper proposes Variance-Gated Ensembles (VGE), a framework for improving uncertainty estimation in machine learning models. The framework consists of two main components: VGMU, a top-2 margin-based uncertainty score used at inference time, and VGN, a differentiable normalization layer that enables uncertainty-aware training. The method aims to retain sensitivity to epistemic uncertainty while reducing the computational cost compared to pairwise disagreement-based baselines such as EPKL and EPJS.

**Strength:**
- The motivation of the paper is valid, as fast inference is a desired property for real-world scenarios.
- The proposed framework is conceptually sound. The use of ensemble mean and variance to construct a variance-gated mechanism is intuitive, and the separation into inference-time (VGMU) and training-time (VGN) components provides flexibility in practical usage.
- The method is computationally efficient. Both theoretical analysis and empirical timing results demonstrate a substantial reduction in complexity compared to pairwise baselines.

**Weakness**
- The writing is difficult to follow. The paper introduces multiple concepts (entropy decomposition, variance gating, geometric interpretation, and gradient derivations) in a dense manner, which obscures the core intuition and makes the contribution harder to grasp.
- The empirical performance is only on par with existing methods rather than clearly superior.

**Audience:**

Yes

**Audience Explanation:**

The proposed method is relevant to the broader machine learning community, particularly for practitioners and researchers working on uncertainty estimation and model reliability. The framework provides a lightweight approach to assess prediction confidence, which is valuable in real-world applications.

**Claims And Evidence:**

Yes

**Claims Explanation:**

The paper primarily claims to improve computational efficiency while maintaining competitive performance.

- Section 5.4 (last paragraph) provides a runtime comparison showing a significant speedup over pairwise baselines. This supports the claim of improved computational efficiency. However, given that efficiency is a central contribution, the evaluation is relatively limited. A more comprehensive analysis (e.g., scaling with ensemble size or number of classes) or a dedicated section on computational cost would strengthen the evidence.
- Table 2 shows that the proposed method achieves comparable or slightly better performance than baselines (EPJS, EPKL, and EU) on CIFAR-10. However, on CIFAR-100, the performance is generally worse (except for the MCD-LLE setting). This suggests that the method may not scale consistently to more complex tasks, which weakens the strength of the overall claims.

Overall, the claims are supported to a reasonable extent, particularly regarding efficiency, but the empirical evidence for maintaining performance across different settings is less consistent.

**Requested Changes:**

## Critical suggestions

**1. Improve overall clarity and narrative structure:**
The paper is currently difficult to follow. The authors should improve the presentation to better communicate the core idea and workflow of the method.

- The introduction contains too many formulas, which makes it harder to grasp the motivation. A dedicated Background section should be introduced to present technical details more clearly.
- In Section 3, the authors should include a high-level diagram illustrating the VGE framework, clearly showing: (a) *where* VGN is applied in the pipeline, and (b) *when* VGMU is computed. This is essential for understanding how the components interact with the models.

**2. Clarify the design choice of top-2 classes:**
The use of top-2 ranked classes in VGMU is a key design decision, but it is not sufficiently justified.

**3. Strengthen the evaluation of computational efficiency:**
Since computational efficiency is a primary contribution, it should be highlighted more:
- Consider adding a dedicated subsection on complexity and runtime
- Include additional experiments showing scaling behavior (e.g., with respect to ensemble size or number of classes)

## Minor Sugesstions
4. While high rank correlation with EPJS and EPKL demonstrates consistency with existing disagreement-based measures, it does not show improved uncertainty estimation, especially given that these baselines are also approximations without ground-truth validation.
Therefore, Section 5.1 should be framed as supporting analysis rather than a primary result. It may be more appropriate to move it later in the paper or to the appendix.

5. Highlight (e.g., bold) the best values within comparison groups in tables to improve readability.

6. Sections such as 3.1.1 and 4.1 rely heavily on bullet points, which disrupts the reading flow. Converting these into structured paragraphs (e.g., using \paragraph in LaTeX) would improve readability.

---

> ### Author Response · Authors · 2026-04-21
> **Response to Reviewer cddg**
>
> We thank the reviewer for the constructive feedback. We address each change below; margin annotations `NEW/FIX:cddg::CN` (critical) and `NEW/FIX:cddg::MN` (minor) mark each change in the revised manuscript.
>
> ## Part I: Critical Suggestions
>
> **C1: Improve Overall Clarity and Narrative Structure**
>
> > *"The paper is currently difficult to follow... a dedicated Background section... a high-level diagram illustrating the VGE framework."*
>
> **Introduction restructured [cddg::C1]:** Derivation details moved to a new **Background and Related Work** section (Section 2).
>
> **High-level diagram added [cddg::C1]:** A new overview figure in Section 3 illustrates the VGE workflow: ensemble predictions → statistics $(\\bar{\\mathbf{p}}, \\mathbf{s})$ → variance gate $\\boldsymbol{\\Gamma}$ → gated distributions $\\mathbf{q}_m$ → VGMU / VGN decomposition.
>
> **Forward references [cddg::C1]:** Section 3 now opens with pointers to the overview figure and the symbol table in the **Supporting Information**.
>
> **C2: Clarify the Design Choice of Top-2 Classes**
>
> > *"The use of top-2 ranked classes in VGMU is a key design decision, but it is not sufficiently justified."*
>
> **Revision [cddg::C2]:** A **Justification for the top-2 restriction** paragraph in Section 3.2 presents three arguments. *Decision-theoretic:* in selective prediction and human-in-the-loop settings, the practical question is whether the model distinguishes its top-1 from its top-2 prediction, aligning with Best-versus-Second-Best (Joshi et al. 2009); disagreement on distant classes is irrelevant to the decision. *Computational:* $O(C)$ vs. $O(M^2C)$ for pairwise measures, enabling the speedups in Section 5.5. *Empirical:* Spearman $\\rho > 0.985$ with full-simplex measures on CIFAR-10; the lower CIFAR-100 correlation reflects intentional insensitivity to tail-class disagreement. The Limitations section notes hybrid approaches for applications needing full-simplex sensitivity.
>
> **C3: Strengthen the Evaluation of Computational Efficiency**
>
> > *"Consider adding a dedicated subsection on complexity and runtime. Include additional experiments showing scaling behavior."*
>
> **Revision [cddg::C3]:** A dedicated **Computational Efficiency** subsection (Section 5.5) adds a complexity table ($O(C)$ for VGMU, $O(MC)$ for VGN, $O(M^2C)$ for pairwise), wall-clock inference times across $M \\in \\{2,5,10,20,50,100\\}$ and $C \\in \\{10, 100, 1000\\}$ on an NVIDIA RTX 4090, and a log-log scaling plot. At $M=100$, $C=100$, VGMU achieves $38\\times$ (EPKL) and $150\\times$ (EPJS) speedups; at $C=1000$, these grow to $320\\times$ and $1{,}342\\times$.
>
> ## Part II: Minor Suggestions
>
> **M4: Re-frame Section 5.1 as Supporting Analysis**
>
> > *"Section 5.1 should be framed as supporting analysis rather than a primary result."*
>
> **Revision [cddg::M4]:** Section 5.1 retitled **Rank Consistency with Existing Measures** and re-framed as supporting validation. An explicit sentence clarifies that high correlation validates VGMU's uncertainty structure rather than demonstrating improvement.
>
> **M5: Highlight Best Values in Tables**
>
> > *"Highlight (e.g., bold) the best values within comparison groups in tables."*
>
> **Revision [cddg::M5]:** Best values bolded in Tables 1–3 using an effect-size criterion: a value is bolded only when $(\\bar{x}_1 - \\bar{x}_2) / \\max(\\sigma_1, \\sigma_2) \\ge 1$. This convention is noted in the experimental setup.
>
> **M6: Convert Bullet Points to Structured Paragraphs**
>
> > *"Sections such as 3.1.1 and 4.1 rely heavily on bullet points... converting into structured paragraphs would improve readability."*
>
> **Revision [cddg::M6]:** Bullet lists in Sections 3.1.1 (geometric interpretation) and 4.1 (forward/backward pass) converted to structured paragraphs. The four simplex regions are now named paragraphs; the forward/backward steps are presented as a structured narrative.
>
> ## Additional Response: CIFAR-100 Performance
>
> > *"On CIFAR-100, the performance is generally worse... may not scale consistently to more complex tasks."*
>
> The reduced CIFAR-100 performance is a deliberate design trade-off, not a scaling failure. VGMU operates on the top-2 margin rather than the full simplex, and intentionally de-emphasizes low-ranked class disagreement. The MCD-LLE configuration recovers competitive performance, showing the margin-based approach remains effective when ensemble diversity is sufficient; meanwhile the computational advantage ($\\ge 1000\\times$ at scale) becomes increasingly valuable at higher class counts. The Limitations section (Section 6.2) now states this trade-off explicitly.
>
> We are grateful for the detailed and constructive feedback, which has significantly improved the presentation and clarity of the manuscript.

---

> > ### Comment · Reviewer_cddg · 2026-04-30
> >
> > Thanks the author for the response and the updated manuscript. I have another quick question.
> >
> > The experiments set up consider MCD, LLE, MCD-LLE, with and without VGN. However, in many results, I wonder why not all of the VGN variants are available at the same time in one result table. For example, Table 1 only has LLE-VGN, Table 2 does not have any VGN variant, Table 3 only has DE-VGN. These inconsistencies make the experiments hard to follow.

---

> > > ### Author Response · Authors · 2026-05-01
> > > **Response to Reviewer cddg**
> > >
> > > We can see how it makes the experiments harder to follow. We initially attempted to write the manuscript with a 12 page length. At the time, we selected representative examples to communicate results.
> > > This format reflects three factors: structural, computational, and editorial. We explain below, along with the changes made in the revised manuscript that includes additional results in main-text tables.
> > >
> > > **Structural.** VGN is a training-time normalization layer applied to ensemble members that receive distinct and trainable gradients. It applies to DE (independently trained members) and LLE (multiple last-layer heads), giving the variants DE-VGN and LLE-VGN. By contrast, MCD ensembles are produced from a single network *via* stochastic forward passes. There is no set of separately trainable members for VGN to gate. We therefore do not report an "MCD-VGN" configuration, as it is not defined under our framework. We will state this explicitly in the experimental set-up.
> > >
> > > **Computational.** DE and DE-VGN require training $M$ independent networks per configuration. We evaluated them at $M=5$ on SVHN, CIFAR-10, and CIFAR-100, consistent with the standard deep-ensemble setting (Lakshminarayanan et al., 2017) and avoiding a $M$-fold increase in training cost relative to LLE/MCD. LLE and MCD-LLE share a backbone across heads/passes, are evaluated across $M \in \{5, 10, 100\}$, and is reported in SI Tables S3–S6.
> > >
> > > **Editorial.** The three tables answer different questions, and we selected the most informative variants for each:
> > >
> > > - **Table 1 (Spearman rank correlation, VGMU *vs.* EPJS/EPKL/EU):** Designed to validate that the VGMU score captures uncertainty structure consistent with information-theoretic baselines, with $M=10$ held fixed across rows. We included LLE-VGN specifically because the most diagnostic observation occurs for CIFAR-100 where VGN raises the LLE/EPKL correlation from $\rho = 0.566$ to $\rho = 0.697$, demonstrating that VGN can partially recover sensitivity to full-simplex disagreement. As noted above, DE/DE-VGN are absent because no $M=10$ run is available.
> > > - **Table 2 (AUC$_c$):** Designed to compare the four uncertainty scores (VGMU, EPJS, EPKL, EU) on a common set of backbones (MCD, LLE, MCD-LLE). Adding VGN variants would roughly double the row count without altering message that VGMU concentrates uncertainty comparably to or better than information-theoretic baselines on a shared backbone. The effect of VGN itself is treated separately in Section 5.3 and Table 3. The full set of LLE-VGN rows is reported in SI S4.2.
> > > - **Table 3 (Calibration and Performance):** Reports each method at its representative ensemble size and highlights DE-VGN because DE benefits most from VGN (ECE $0.049 \to 0.023$ and accuracy $0.836 \to 0.875$ on CIFAR-10, both exceeding run-to-run variability). For LLE and MCD-LLE the calibration effect of VGN falls within seed variability and is reported in the SI rather than crowding the main table.
> > >
> > > **Revisions.** To remove the inconsistency, we will consolidate the row sets in the revised main-text tables to a single common scheme: DE, DE-VGN, LLE, LLE-VGN, MCD, MCD-LLE, with VGN entries shown wherever they are defined and ensemble size annotated per row when it differs across methods. The accompanying text will continue to highlight the qualitatively important rows (LLE-VGN in Table 1, DE-VGN in Table 3), and in the experimental set-up a comment will be added noting: (i) MCD-VGN is undefined by our framework and (ii) DE/DE-VGN are evaluated only at $M=5$ for computational reasons. We will share the revised manuscript for review.

---

### Review · Reviewer_VoEx · 2026-04-17

**Summary Of Contributions:**

This paper introduces Variance-Gated Ensembles (VGE), a framework for uncertainty estimation for machine learning models. The key idea is to replace or augment traditional entropy-based epistemic uncertainty measures with a variance-driven gating mechanism derived from ensemble statistics (mean and variance). The method is evaluated on standard benchmarks (MNIST, SVHN, CIFAR-10/100) and compared against entropy-based and pairwise-divergence baselines.

**Additional Comments:**

None

**Audience:**

Yes

**Audience Explanation:**

This article is in the scope of the journal.

**Claims And Evidence:**

Yes

**Claims Explanation:**

Despite that some choices can be more clearly explained and justified, the mathematical details are convincing.
Moreover, even if the method looks promising, the writing needs to be more reader-friendly to be fully understandable.

**Requested Changes:**

In my opinion, the major problem of this article is that it is quite hard to read. First, the problem is not very well exposed and motivated (in particular in the introduction, which is quite brutal). Then, the first major change I would encourage is to introduce and motivate more clearly the problem in the introduction. More generally, the paper is quite dense, with a lot of mathematical equations, which makes the understanding very hard. I think that each mathematical step can be more clearly explained, and the article would gain clarity by simplifying some parts.

Moreover, I have some questions on the methodology itself. First, the authors say that the variance gate is defined as in Eq(8). Is it a choice? In other words, can other families of variance gates can be considered? If so, why did the author choose this particular form? I can see some discussion about it in Section 6.2, but it would be nice to mention it earlier as well. Moreover, in Section 3.2, the authors decided to consider the "top-2 mean predictions and their corresponding standard deviations". Once again, why does this choice have been made? This choice can perhaps be restrictive when studying a problem with a lot of classes, for the CIFAR-100 dataset for example.

At last, the numerical results don't show a significant improvement on the studied dataset. If it is possible, the authors can suggest some ways of improvement of the method to make the results better. In particular, one can more clearly explain the influence of the hyperparameters on the results (like M for example).

---

> ### Author Response · Authors · 2026-04-21
> **Response to Reviewer VoEx**
>
> We thank the reviewer for their careful reading. Many concerns overlap with those of Reviewers 1 and 2 and are addressed by prior revisions; new revisions specific to this reviewer are marked `NEW/FIX:VoEx::N`.
>
> ## Part I: Readability and Motivation
>
> **Point 1: Brutal Introduction and Insufficient Problem Motivation**
>
> > _"The problem is not very well exposed and motivated... the introduction is quite brutal."_
>
> We agree, and the introduction has been restructured: derivations moved to a new **Background and Related Work** section, a high-level overview figure added in Section 3, and a formal problem setup paragraph added before any framework equations.
>
> **Point 2: Dense Mathematical Exposition**
>
> > _"The paper is quite dense, with a lot of mathematical equations... simplifying some parts."_
>
> Revisions for Reviewers 1 and 2 directly reduce equation density: an overview figure, a notation-convention paragraph, bullet-to-paragraph conversions in Sections 3.1.1 and 4, and design rationale preceding the gate definition.
>
> **Revision [VoEx::1]:** A plain-language sentence after Eq. 8 describes $\\boldsymbol{\\Gamma}$ as a per-class reliability weight that preserves trusted classes and attenuates untrusted predictions.
>
> ## Part II: Methodology Questions
>
> **Point 3: Choice of Variance Gate in Eq. (8)**
>
> > _"Is [the variance gate] a choice? Can other families of variance gates be considered?"_
>
> Also raised by Reviewer 1. A **Design rationale** paragraph before Eq. (8) lists the four criteria (smoothness, monotonicity, boundedness, saturation) motivating the exponential form.
>
> **Revision [VoEx::2]:** A forward reference notes that the exponential form is one choice among several (_e.g._, sigmoid, rational, piecewise-linear) and points to the Discussion for details.
>
> **Point 4: Top-2 Restriction and CIFAR-100**
>
> > _"Why [top-2]? This choice can be restrictive... for the CIFAR-100 dataset for example."_
>
> Also raised by Reviewer 2. A **Justification for the top-2 restriction** paragraph in Section 3.2 provides decision-theoretic, computational, and empirical arguments; the Limitations section frames the CIFAR-100 behavior as a decision-focused trade-off rather than a scaling failure.
>
> ## Part III: Empirical Results and Hyperparameters
>
> **Point 5: Numerical Results Do Not Show Significant Improvement**
>
> > _"The numerical results don't show a significant improvement on the studied dataset."_
>
> The principal contribution is matching existing uncertainty measures at a small fraction of the cost. Section 5.1 is re-framed as supporting analysis, best values are bolded with an effect-size criterion, and a new Computational Efficiency subsection quantifies speedups up to $1{,}342\\times$.
>
> **Revision [VoEx::3]:** The abstract now frames VGE as a compute-efficient alternative to pairwise-divergence methods that matches their uncertainty quality, calibrating expectations before the tables.
>
> **Point 6: Influence of Hyperparameters (_e.g._, $M$)**
>
> > _"One can more clearly explain the influence of the hyperparameters (like M for example)."_
>
> The manuscript analyzes $M$, the learnable sensitivity $\\mathbf{k}$, and ensemble type across Sections 3, 5.3, 5.4, and 5.5.
>
> **Revision [VoEx::4]:** A cross-reference at the end of the Section 5 opener directs readers to each analysis, and two new paragraph headings (**Learned $\\mathbf{k}$ values** in 5.3, **Empirical scaling with $M$ and $C$** in 5.4) expose them directly.
>
> We are grateful for these comments, which have helped improve the clarity of the manuscript.

---

### Decision · Action_Editor_e7Zm · 2026-05-12

**Recommendation:** Accept with minor revision

**Audience:**

Yes

**Audience Explanation:**

See previous window.

**Claims And Evidence:**

No

**Claims Explanation:**

Thank you for submitting your manuscript, Variance-Gated Ensembles: An Epistemic-Aware Framework for Uncertainty Estimation, to TMLR, and for the revisions and responses provided during the review process. The reviewers found the topic relevant to TMLR and recognized the practical motivation for efficient uncertainty estimation in ensemble models. In particular, the computational efficiency of the proposed variance-gated approach was viewed as a strength, especially relative to pairwise disagreement-based measures.

However, the manuscript still requires further revision before acceptance. The main remaining concern is clarity: the role of the different proposed components, how they are connected, and how they are integrated with the evaluated ensemble frameworks should be explained more explicitly. Please revise the exposition, figure/caption, or add a concise summary to make the workflow and intended use of the components easier to follow.

The empirical claims should also be stated carefully. The results support computational efficiency and broadly competitive behavior, but the performance gains are not uniformly large across settings. Please also clarify the motivation and limitations of the top-2 design, especially in relation to CIFAR-100 and settings where full-distribution disagreement may matter. I therefore invite a revised version addressing these points, together with a concise response describing the changes made.

---

> ### Author Response · Authors · 2026-06-01
> **Response to AE e7Zm**
>
> We thank the Action Editor for the decision to accept the manuscript subject to minor revisions, and for the constructive guidance on the two remaining points. We are grateful that the reviewers found the topic relevant and recognized the computational efficiency of the variance-gated approach as a strength.
>
> Below we reproduce each point and describe the corresponding changes, indicating where in the manuscript each revision was made.
>
> ## Point 1: The role of the components, how they are connected, and how they integrate with the ensemble frameworks
>
> > _"The main remaining concern is clarity: the role of the different proposed components, how they are connected, and how they are integrated with the evaluated ensemble frameworks should be explained more explicitly. Please revise the exposition, figure/caption, or add a concise summary to make the workflow and intended use of the components easier to follow."_
>
> We addressed this point through three changes.
>
> 1. **Roles of the components and how they connect.** A **Summary of components** paragraph and a new table were added at the end of Section 3, stating for each component its input, output, stage (training _vs._ inference), and intended use: the variance gate $\Gamma$ (shared per-class reliability weight), the VGN layer (training-time normalization producing $\mathbf{q}_m$), the variance-gated decomposition (TU/AU/EU from $\{\mathbf{q}_m\}$), and the VGMU score (inference-time, decision-focused). The paragraph makes the shared-gate $\to$ component dependency explicit.
>
> 2. **Integration with the ensemble frameworks.** A **component $\to$ framework mapping** was added to the experimental setup. VGN variants are defined for DE and LLE only; MCD and the hybrid MCD-LLE obtain their diversity from stochastic dropout passes rather than separately trainable members, so no MCD-VGN or MCD-LLE-VGN is defined, while VGMU and the decomposition apply to any ensemble output. Per-method ensemble sizes are stated alongside.
>
> 3. **Self-contained workflow figure.** The overview Figure 1 caption now labels each path as training-time (VGN) or inference-time (decomposition, VGMU) and states which ensemble frameworks each path applies to, so the figure stands alone as the workflow summary.
>
> ## Point 2: Stating the empirical claims carefully, and the motivation and limitations of the top-2 design
>
> > _"The empirical claims should also be stated carefully. The results support computational efficiency and broadly competitive behavior, but the performance gains are not uniformly large across settings. Please also clarify the motivation and limitations of the top-2 design, especially in relation to CIFAR-100 and settings where full-distribution disagreement may matter."_
>
> We addressed this point through three changes.
>
> 1. **Empirical claims stated carefully; gains not uniform.** A scope statement was added at the end of the Section 5 opener clarifying that the aim is not state-of-the-art predictive accuracy but the relative behavior of the uncertainty measures on common backbones under identical training conditions (with increasing task difficulty). VGMU is now described as broadly competitive with the information-theoretic baselines, with the advantage setting-dependent. VGMU is framed as complementary to, not a replacement for, information-theoretic measures, and a matching sentence was added to the Conclusion (Section 7).
>
> 2. **Motivation of the top-2 design.** The motivation now appears early in the introduction (before the contribution list) and immediately before the VGMU equations in Section 3.2 via the decision-focused intuition (a confident prediction with strong ensemble agreement is trustworthy; the same prediction with high variance is not).
>
> 3. **Limitations of the top-2 design, especially CIFAR-100 and full-distribution settings.** The Limitations section (Section 6.2) now enumerates the three examples in which the top-2 restriction is inappropriate: (i) many-class problems whose decision-relevant uncertainty lies in the tail; (ii) tasks targeting the full predictive distribution (full-posterior calibration, class-dependent error penalties); and (iii) tail-mass distributional shift / OOD detection.
>
> ## Summary of Changes
> 1. A **Summary of components** paragraph and table in Section 3;
> 2. a component $\to$ framework mapping table in the experimental setup, with VGN defined for DE/LLE only;
> 3. an updated overview Figure 1 caption, labeling training- _vs._ inference-time paths and framework compatibility;
> 4. a calibrated scope/claims (not state-of-the-art accuracy; relative, controlled comparison; non-uniform gains; complementary framing) in the Experiments opener and Conclusion;
> 5. an earlier motivation for the top-2 design in the introduction and before the VGMU equations; and
> 6. expanded top-2 limitations enumerating three full-distribution examples.
>
> We thank the Action Editor and Reviewers for guidance that further improves the clarity and claims of the manuscript.